# Architecture and autoinhibitory mechanism of the plasma membrane Na⁺/H⁺ antiporter SOS1 in *Arabidopsis*

Yuhang Wang[1,2,3], Chengcai Pan[4], Qihao Chen[1,2,3], Qing Xie[4], Yiwei Gao [1,2,3], Lingli He[1,2,3], Yue Li [1,2,3], Yanli Dong [1,2,3], Xingyu Jiang[4] ✉ & Yan Zhao [1,2,3] ✉

Salt-overly-sensitive 1 (SOS1) is a unique electroneutral Na⁺/H⁺ antiporter at the plasma membrane of higher plants and plays a central role in resisting salt stress. SOS1 is kept in a resting state with basal activity and activated upon phosphorylation. Here, we report the structures of SOS1. SOS1 forms a homodimer, with each monomer composed of transmembrane and intracellular domains. We find that SOS1 is locked in an occluded state by shifting of the lateral-gate TM5b toward the dimerization domain, thus shielding the Na⁺/H⁺ binding site. We speculate that the dimerization of the intracellular domain is crucial to stabilize the transporter in this specific conformation. Moreover, two discrete fragments and a residue W1013 are important to prevent the transition of SOS1 to an alternative conformational state, as validated by functional complementation assays. Our study enriches understanding of the alternate access model of eukaryotic Na⁺/H⁺ exchangers.

Sodium is one of the most abundant elements in Earth's crust. For most plants, low levels of Na⁺ are beneficial for many processes, such as maintaining osmotic pressure, partly eliminating potassium deficiency symptoms, and stimulating plant growth[1,2]. However, excessive accumulation of cytosolic Na⁺ disrupts several important physiological processes, including the cell cycle, enzymatic reactions, and photosynthesis, and thus leads to stunted growth and arrested cell development[3,4]. Therefore, Na⁺ extrusion from the cytosol is critical for the salt tolerance of plants, especially in saline soil[5]. SOS1, which is highly conserved in the phylae of the plant kingdom, is identified as the only Na⁺/H⁺ transporter that localizes at the plasma membrane and specifically exports Na⁺ by utilizing the electrochemical proton gradient generated by H⁺-ATPase[6,7]. Compared with the vacuolar Na⁺/H⁺ transporters, SOS1 is insensitive to K⁺ and plays a unique physiological role in cellular Na⁺ homeostasis[8,9]. Overexpression of SOS1 has been shown to improve plant salt tolerance[10], whereas disruption of SOS1 expression conversely results in the loss of salt resistance in

halophytes[11]. Thus, SOS1 has become an important target for the improvement of salinity resistance in agricultural biotechnology[12,13].

SOS1 is an electroneutral Na⁺/H⁺ antiporter that belongs to the NhaP/SOS1 subfamily of the monovalent cation proton antiporter family 1 (CPA1)[14,15]. The transmembrane portion of SOS1 exhibits significant sequence similarities with the animal, bacterial, and fungal plasma membrane Na⁺/H⁺ antiporters. The molecular mechanisms of Na⁺/H⁺ transport have been extensively studied over the past two decades, including in studies of electrogenic antiporters such as *Escherichia coli* (EcNhaA)[16] and *Thermus thermophilus* (TtNapA)[17], electroneutral transporters such as *Pyrococcus abyssi* (PaNhaP)[17], *Methanocaldococcus jannaschii* (MjNhaP)[18], the mammalian protein *Equus caballus* NHE9[19] and the human NHE1-CHP1 complex in inward- and cariporide-bound outward-facing conformations[20] and in the self-inhibited NHE3-CHP1 complex[21]. These studies showed that these transporters form homodimers, with each monomer consisting of 12 or 13 transmembrane (TM) helices that contain a dimerization domain

[1]National Laboratory of Biomacromolecules, CAS Center for Excellence in Biomacromolecules, Institute of Biophysics, Chinese Academy of Sciences, 100101 Beijing, China. [2]State Key Laboratory of Brain and Cognitive Science, Institute of Biophysics, Chinese Academy of Sciences, 15 Datun Road, 100101 Beijing, China. [3]College of Life Sciences, University of Chinese Academy of Sciences, 100049 Beijing, China. [4]National Center for Technology Innovation of Saline-Alkali tolerant Rice/College of Coastal Agricultural Sciences, Guangdong Ocean University, 524088 Zhanjiang, China. ✉e-mail: jiangxingyuhu@163.com; zhaoy@ibp.ac.cn

responsible for dimerization and an ion transport core domain. The ion binding cavity is located between these two domains and possesses highly conserved acidic residues, which are responsible for substrate ion binding and are alternately exposed to the intracellular or extracellular sides during the transport cycle in a model of an elevator mechanism[22,23]. In addition to the conserved transmembrane domain, SOS1 harbors a cytoplasmic C-terminal domain consisting of ~700 amino acids, making it the largest known cation/proton antiporter[24]. Previous studies have shown that the cytoplasmic domain contains an autoinhibitory region (residues: 998–1146) to keep the transporter in an inactive state[7]. Phosphorylation of SOS1 mediated by SOS3 (CBL4)-SOS2 (CIPK24) or CBL10-SOS2/CIPK8 protein kinase complexes is crucial for recovery of the Na$^+$ extrusion activity of SOS1[25–27]. However, the mechanism underlying the autoinhibition of SOS1 by its intracellular domain remains largely unknown.

In this study, we express and purify full-length wild-type SOS1 and resolve its structure by single-particle cryo-electron microscopy. SOS1 forms a homodimer and is composed of a TMD and an intracellular domain. Based on the structural comparison, SOS1 is stabilized in an occluded state, mediated by interactions between the TMD and intracellular domain. Moreover, we identify two segments that are able to regulate the activity of SOS1. Deletion or mutations within these two regions can significantly rescue SOS1 transport activity, as evidenced by functional complementation assays using salt-sensitive yeast.

## Results

### Architecture of *Arabidopsis thaliana* SOS1

To investigate the autoinhibitory mechanism of SOS1, we expressed full-length wild-type *Arabidopsis thaliana* SOS1 in HEK293F cells. The sharp and symmetric size-exclusion chromatography (SEC) profile showed that the purified SOS1 sample was homogeneous. The peak fractions were pooled and concentrated for cryo-EM sample preparation (Supplementary Fig. 1a, b). A total of 1552 micrographs were collected and subjected to further analysis with RELION-3.1 and cryoSPARC. Four classes (class I–IV) are generated from 3D classification without alignment (Supplementary Fig. 2a). We determined three cryo-EM maps from different classes (class I, II, and IV) at 3.5 Å, 2.8 Å, and 3.4 Å, respectively, according to the gold-standard Fourier shell correlation (GSFSC) criterion (Supplementary Fig. 2b–j and Table 1). The high-quality cryo-EM maps allow us to reliably build the SOS1 models (Supplementary Fig. 3). Next, we compared the SOS1-ClassI and SOS1-ClassIV with the 2.8-Å resolution SOS1-ClassII. We found that the structures of SOS1-ClassI and SOS1-ClassII are nearly identical, supported by RMSD of ~0.3 Å for 1856 Cα atoms. However, superimposition of SOS1-ClassIV structure to the SOS1-ClassII structure gives rise to RMSDs of 1.5 Å for overall structure, indicating that two different SOS1 structures have been resolved. We firstly focused on the 2.8-Å resolution SOS1-ClassII to display structural features of SOS1 and name SOS1-ClassII as SOS1 in short.

The resulting dimeric SOS1 is approximately 85 Å in length, 95 Å in width, and 120 Å in height (Fig. 1a, b). *Arabidopsis thaliana* SOS1 consists of 1146 residues. However, our SOS1 model is only composed of residues 32–972, 1007–1017, and 1030–1048 and the vast majority of the autoinhibitory region (998–1146) is not resolved due to high flexibility[28] (Fig. 1c). The TMD is composed of residues 32 to 450 and contains 13 TM helices with an extracellular N-terminus and an intracellular C-terminus. Similar to other antiporters in the CPA family[19–21,29], the arrangement of TM helices is conserved in SOS1, that is, TMs 4–6 and TMs 11–13 form the core domain, and TMs 1–3 and TMs 7–10 form the dimerization domain (Fig. 1d and Supplementary Figure 4a–e). TM5 and TM12 are broken at the cross position, yielding the four fragments TM5a, TM5b, TM12a, and TM12b. Intriguingly, in addition to the short amphipathic helix between TM3 and TM4 (AH$^{3-4}$) on the intracellular side[20,23] (Supplementary Fig. 5a, b), SOS1 has an amphipathic helix on the extracellular side between TM2 and TM3 (AH$^{2-3}$; Fig. 1d).

The intracellular domains extend ~70 Å into the cytoplasm. Based on the enrichment of secondary structural elements and structural homology, we divided the intracellular domain into four parts, including an α-helix-rich cytoplasmic domain (α-CTD, residues 458–722), a cyclic nucleotide-binding domain-like domain (CNBLD, residues 732–883), a β-sheet-rich cytoplasmic domain (β-CTD, residues 888–972) and a not fully resolved C-terminal autoinhibitory tail (Fig. 1b, c). The α-CTD contains eight α-helical segments (HC1–HC8), which are arranged into a U-shaped structure and located proximal to the inner surface of the membrane. Every two adjacent helices are folded in an antiparallel manner connected by short loops. The HC1-HC2 helices and the HC7-HC8 helices form the two arms of the U-shaped α-CTD (Fig. 1e). HC1 directly connects to TM13 via an 8-amino acid linker at the N-terminus, forming a 15° angle to the membrane plane. However, the other arm formed by HC7 and HC8 is positioned at opposite sides of the HC1-HC2 helices, in which HC8 is parallel to the membrane plane, participating in interactions with intracellular loops between TM helices (Supplementary Fig. 5a, c, d).

CNBLD consists of a bent α-helix (αA) followed by a six-strand antiparallel β-roll (β1–β6) containing a short α-helix (αP) between β4 and β5, followed finally by three additional α-helices (αB–αD), which are located beneath the α-CTD (Supplementary Fig. 5e). The surface of the β-roll forms extensive electrostatic interactions with the HC6 of the α-CTD and there is hydrophobic interaction between αC of CNBLD and β-CTD (Supplementary Fig. 5f, g). CNBLD is homologous to cyclic nucleotide-binding domains (CNBDs) and is generally considered to be a putative cyclic nucleotide-binding motif[30]. Superimposition of the CNBLD with the human hyperpolarization-activated cyclic nucleotide-gated ion channel HCN1[31] and plant hyperpolarization-activated phosphorylation-regulated AKT1[32,33] using the β-roll as a reference, the αA and αD helices at the N- and C-termini undergoes remarkable conformational change (Supplementary Fig. 6a). The root mean squared deviation (RMSD) for α carbons is 2.11 Å and 2.72 Å between CNBLD and the CNBD of the HCN1 and AKT1, respectively. Relative to the cAMP-unbound state, αB is more similarly positioned to the relative helix in the cAMP-bound form of HCN1, and αC moves close to the cavity formed by the β-roll that serves as the cyclic nucleotide-binding pocket, probably causing steric hindrance for cAMP binding (Supplementary Fig. 6b). Moreover, the residues that form specific electrostatic and hydrogen-bonding interactions with bound cyclic nucleotides within the substrate-binding pocket are not conserved in SOS1, especially the highly conserved arginine residue located in the loop between the αP and the β5 sheet (e.g., R549 in human HCN1; Supplementary Fig. 6c) is replaced by hydrophobic Y820 in the CNBLD. Furthermore, the sidechain of E813 is oriented into the binding cavity and takes the place of the cyclic phosphate moiety, suggesting that CNBLD cannot bind and be regulated by cyclic nucleotides (Supplementary Fig. 6c). A previous study indicated that G777D and G784E mutations lead to inactive transporters[7]. Based on our structure, G777 is located on the N-terminal of β1 and forms a "mortise-tenon" joint with F834 on the neighboring β6 (Supplementary Fig. 6d, e). Such an interaction has been proved to be important for aligning and locking the neighboring β-sheets and thus improve the folding speed and stability of β-sheet[34,35]. G784 is the most conserved glycine among eukaryotic and prokaryotic CNBD, which adopt special dihedral angles that is disallowed for the other amino acid[36]. Thus, we speculate that the mutation of G777 or G784 may affect folding and stability of the CNBD, and thus significantly decrease transport activity of the SOS1.

The β-CTD touches the CNBLD and is close to the twofold symmetric axis on the bottom of the structure (Supplementary Fig. 5e, g). The density of β-CTD is not well resolved, probably due to its high conformational flexibility. Therefore, model building for the β-CTD is

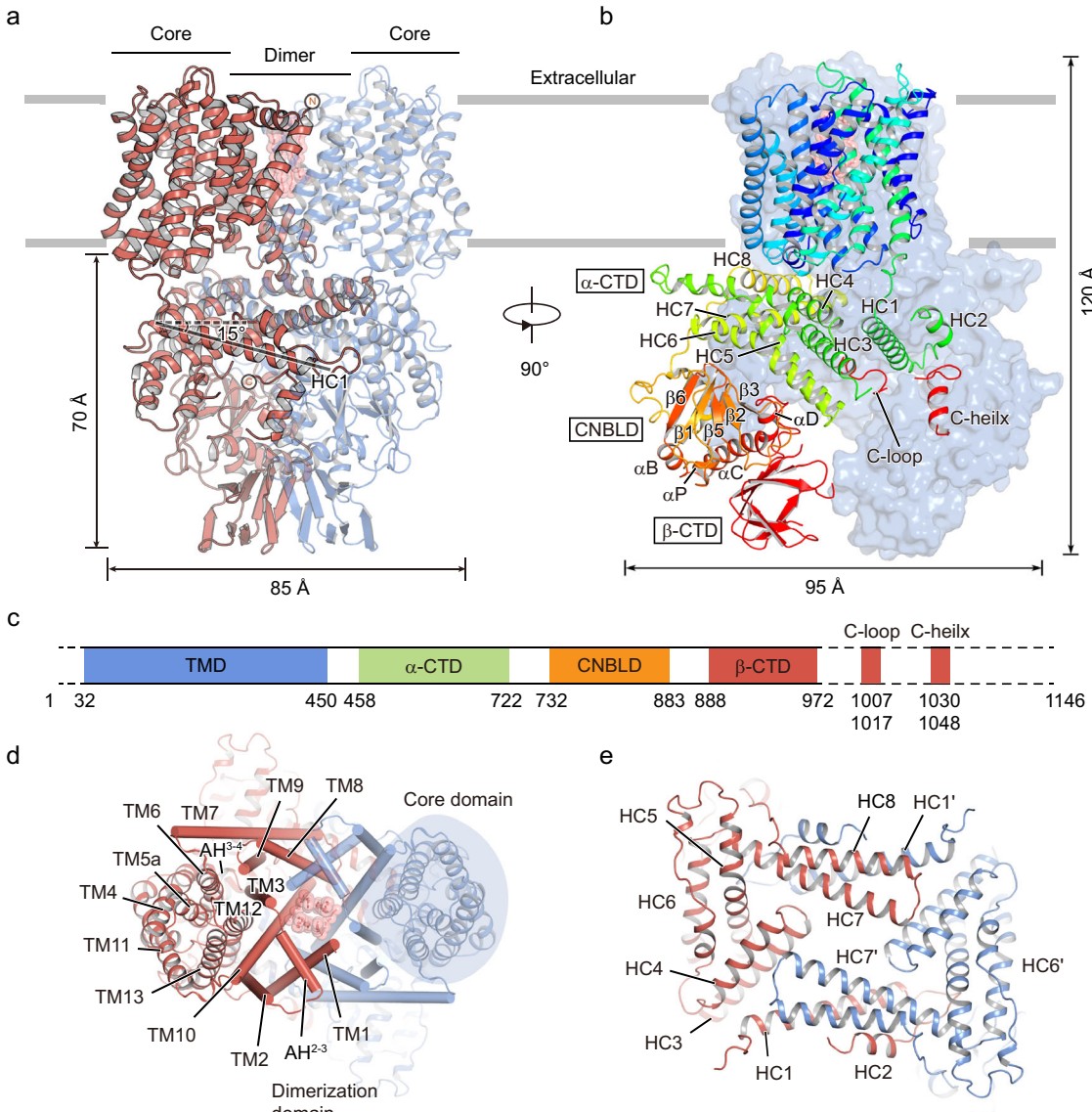

**Fig. 1 | Architecture of *Arabidopsis thaliana* SOS1. a, b** Overall structure of the dimeric model of SOS1 from two perpendicular views, parallel to the membrane plane. The two protomers in **a** are colored in deepsalmon and blue, respectively. Gray lines represent boundaries of the cell membrane. The N- and C-termini of the model are labeled. HC1 helix forms an angle of approximately 15 degrees with the plane of the membrane. The two protomers in **b** are shown as cartoon and surface representations. The protomer in the cartoon is colored in rainbow. The height and width of the structure are indicated. **c** Domain structure and organization of SOS1, numbers of amino acid defining the domains are indicated. Dashed line indicated the unresolved disorder regions in the structure. **d** Structure of the transmembrane domain viewed from the extracellular side. Core and dimerization domains are shown in cartoon and cylindrical representations, respectively. One core domain is shaded within a light blue oval. **e** The dimeric assembly of α-CTD viewed from the extracellular side.

initiated by docking the predicted structure by AlphaFold2 into the cryo-EM map as a rigid body. The last 174 residues comprised the autoinhibition domain of SOS1, which is largely not resolved in the cryo-EM map, in line with secondary structure prediction results suggesting that this segment is intrinsically disordered.

**Interactions mediate SOS1 dimerization**

Similar to other homologous transporters, SOS1 assembles as a homodimer. Many contacts, occurring within the TMD and CTD, contribute to the dimerization of SOS1[24]. Within the TMD, dimerization of SOS1 is primarily mediated by extensive van der Waals interactions between TMs 1, 8, and 10 (Fig. 2a). Most of the relevant residues from these helices are hydrophobic. Residues from protomer A, such as V34, V37, V38, V40, L44, V45, and I48 on TM1, interact with V259, I260, L264, T263, I266, A267, Y270, F271, Y274, and W279 on TM8′ from protomer B (the prime indicates its being from the symmetry mate)

(Supplementary Fig. 7a). At the site proximal to the intracellular surface, the N-terminus of TM10 is located near the 2-fold symmetry axis, and forms hydrophobic interactions with each other via residues F314, M317 and I321. The N-terminus of TM8 is also tightly patched with the intracellular helix of TM10, forming a hydrophobic cluster. In addition, at the periphery of the interface, there is a salt bridge between H313 and S310 at the C-terminus of TM10, providing further stabilization (Supplementary Fig. 7b). There is also a polar interaction between the N-terminal K85 of the extracellularly amphipathic helix AH²⁻³ and E278 of the C-terminal of TM8′, which probably contributes to the stabilization of the dimerization structure (Supplementary Fig. 7c). Moreover, a cavity in the central part of the dimer was visualized on the extracellular side, as previously observed in the NHE1 structures[20]. The cavity of SOS1 is smaller than that of NHE1 with four strip-shaped EM densities located within this cavity, which likely represent the hydrophobic tails of lipid molecules and form extensive hydrophobic

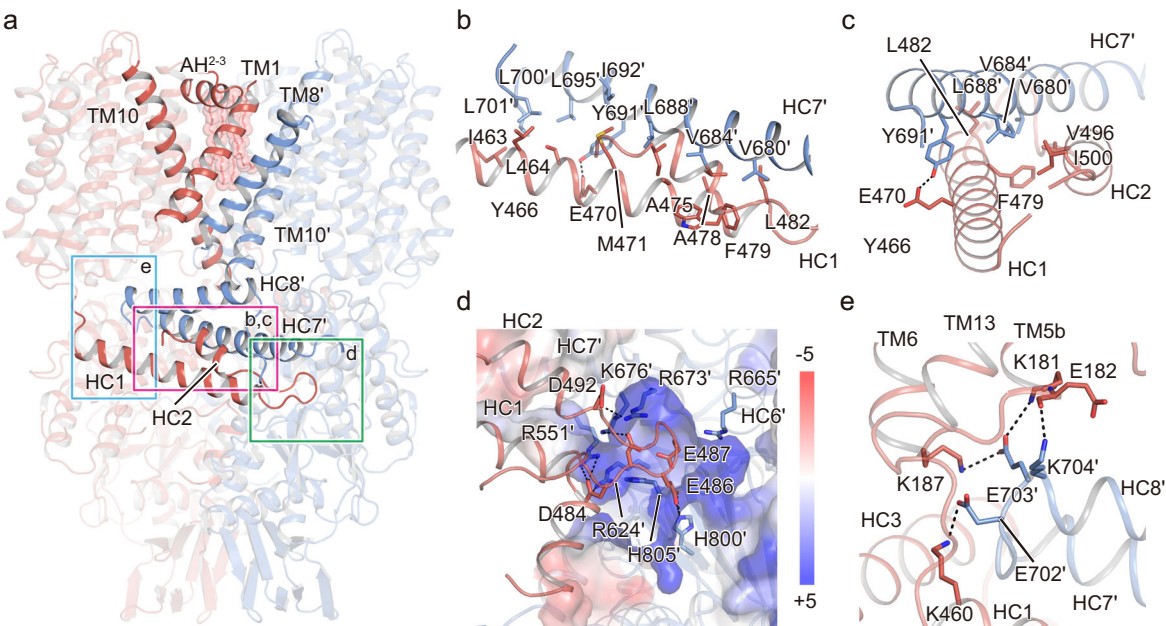

**Fig. 2 | CTD dimerization interface of SOS1. a** Overall structure of the SOS1 dimer model. Helices involved in the dimeric interactions are highlighted. Three regions in the dimeric CTD are boxed. **b, c** Interactions between HC1 and HC2 from one protomer and HC7′ from another protomer viewed from the front of the structure (**b**), and from the left (**c**). Sidechains of residues contributing to dimerization are shown as sticks. The hydrogen bond between E470 and Y691′ is shown as dashed lines. **d** Electrostatic complementary interactions between the HC1-HC2 loop and the pocket formed by α-CTD′ and CNBLD′ from another subunit. The positively charged pocket is shown according to its electrostatic potential calculated with the Adaptive Poisson-Boltzmann Solver (APBS) plugin in Pymol and contoured at ±5 kT/e. Residues that are harbored in the pocket and involved in the interactions are shown as sticks. **e** Electrostatic complementary interactions between the juxtamembrane helix HC8′ and TM5, TM6 of TMD and HC1 of α-CTD from the other subunit. Hydrogen and salt-bridge bonds are shown as dashed lines.

interactions with the hydrophobic residues W93[AH2-3], F105[TM3], P107[TM3], F314[TM10], and W315[TM10] (Supplementary Fig. 7d). Considering no additional lipids were supplied during sample preparation, we speculate that these lipid molecules observed within the cavity may be derived from HEK293 host and are probably involved in stabilizing the dimeric SOS1.

SOS1 harbors a large intracellular domain, which provides extensive interactions to stabilize the dimer, especially at the α-CTD (Fig. 2a). For instance, two arms of each U-shaped α-CTD are crossed to form a rectangle-shaped structure. Consequently, the HC1 and HC2 helices from one subunit interact closely with HC7 from another subunit by extensive hydrophobic and van der Waals interactions (Fig. 2b, c). E470 on HC1 and Y697 on HC7′ form hydrogen bonds to further stabilize the dimerization of α-CTD. Moreover, the conserved negatively charged loop between the HC1 and HC2 helices (HC1-HC2 loop, [482]LGDDEELGPAD[492]) of α-CTD inserts into a pocket formed by α-CTD′ and CNBLD′ from the other subunit and forms salt bridge and hydrogen bonding interactions with a cluster of highly conserved basic and polar residues, including R624[HC5′], R665[HC6′] R673[HC7′], K676[HC7′], H800[CNBD′], and H805[CNBD′] (Fig. 2d and Supplementary Fig. 8). Furthermore, some charged residues are distributed at the N-terminus of HC8′, including E702, E703, and K704 (Fig. 2e). They are located proximal to the cell membrane and are involved in electrostatic interactions with surrounding positively charged residues, such as K181[TM5b], K187[TM6], and K460[HC1], which may contribute to mediating the dimerization of SOS1.

The structural comparison indicates that the dimerized SOS1-ClassIV harbors a distinct conformation to the 2.8-Å resolution SOS1 that the TMD undergoes significant conformational changes and the intracellular domains are nearly identical (Fig. 3a–d), supported by the RMSD values between these two structures are 2.0 Å and 0.7 Å for TMD and CTD, respectively. Taking a close look, the extracellular ends of TM helices from two protomers expand in opposing directions, except that the TM8 and TM9 remain unaltered (Fig. 3b), while the

cytoplasmic ends of TM helices are superimposable (Fig. 3c). Therefore, we termed the SOS1-ClassIV structure as SOS1[expand]. We next compared structures of one protomer in these two conformations and found that they adopt a similar conformation, but have some subtle structural discrepancies at TM8 and TM9 helices (Fig. 3e), with RMSD of 0.6 Å. Within the hydrophobic central cavity between dimerization domains, we found that the lipid molecules differently interact with the SOS1 in these two conformations. In both structures, four lipid molecules are determined at similar positions proximal to the 2-fold symmetrical axis (Fig. 3f, g). However, in the SOS1[expand] structure, an extra lipid is inserted into space between TM1 and TM3′ helices at the dimerization interface. Meanwhile, the side chain of Y270 on TM8′ is extruded in the SOS1[expand] structure to accommodate this lipid molecule (Fig. 3h).

## Autoinhibitory mechanism of SOS1 mediated by the intracellular domain

Previous studies have shown that the C-terminus of SOS1 contains an autoinhibitory domain to keep the transporter in a resting state with basal activity[7]. The protein kinase complex formed by SOS3 (CBL4)-SOS2 (CIPK24) or CBL10-SOS2/CIPK8 in *Arabidopsis* can phosphorylate and activate SOS1 to relieve autoinhibition[25–27]. In addition, truncation of the C-terminal 148 amino acids (SOS1[Δ998]) is able to fully activate SOS1. Considering that our SOS1 sample was expressed in full length and in the absence of any overexpressed protein kinases, we speculate that the structure of SOS1 is trapped in an inhibited state. To investigate the autoinhibitory mechanism of the SOS1 transporter, we compared the TMD of SOS1 with the human homolog NHE1 in either the inward-facing or outward-facing state[20], yielding RMSD values of 2.4 Å and 3.2 Å, respectively, indicating that SOS1 is stabilized in an inward-facing conformation (Supplementary Fig. 4a, b). In line with previous observations, SOS1 harbors a negatively charged cytoplasmic cavity with a closed extracellular vestibule within each protomer. The depth of the cavity is 14 Å, which is clearly shallower than that seen in

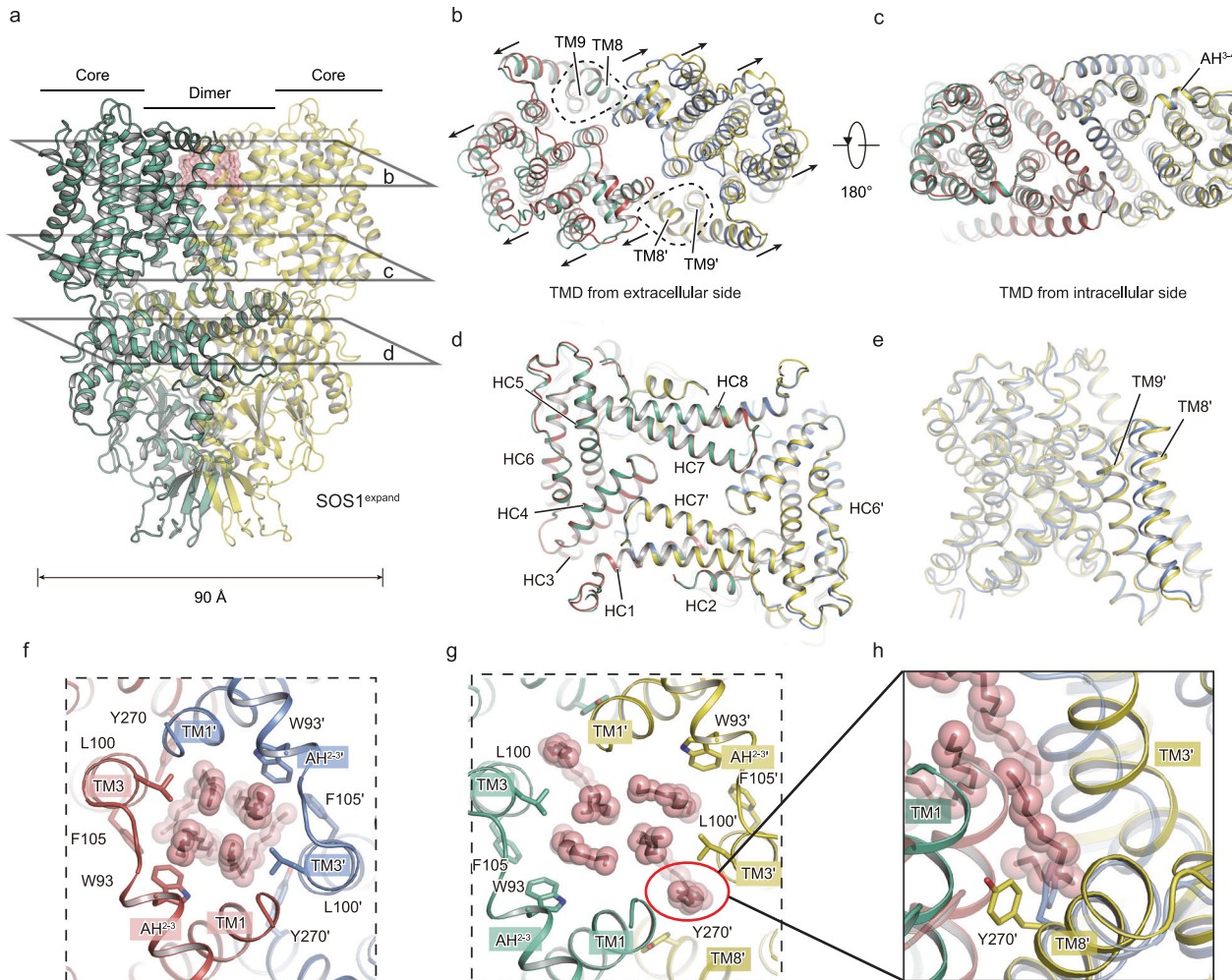

**Fig. 3 | Lipids induced rearrangement of dimerization interface. a** Overall structure of the SOS1$^{expand}$. Two monomers are colored in green and yellow, respectively. Lipid molecules are shown as red sticks and spheres. **b−d** Structural comparisons based on the whole dimeric structure between SOS1$^{expand}$ and SOS1, illustrated using three slice views **b–d** as indicated by labeled panes; namely, TMD from extracellular side (**b**), TMD from intracellular side (**c**), and α-CTD from extracellular side (**d**). Black arrows indicated the helix movements of SOS1$^{expand}$ compared to the SOS1. TM8 and TM9 are highlighted and labeled in **b**. **e** Structural comparison between the monomers of SOS1$^{expand}$ and SOS1, using monomer itself as the alignment reference. TM8 and TM9 are labeled. **f** Four lipid molecules embedded at the dimer interface of SOS1. **g, h** Six lipid molecules embedded at the dimer interface of SOS1$^{expand}$. The extra lipid in a SOS1$^{expand}$ monomer is highlighted in **g**, and its interactions with adjacent regions are shown in **h**. The flipped sidechain of Y270 was shown as sticks.

NHE1 (Fig. 4a). The conserved aspartate residue in "ND" motif on TM6 has been extensively characterized in CPA1 as the central ion binding site[37]. In the inward-facing and outward-facing NHE1 structures, residue D267$^{TM6}$ is exposed to the intracellular and extracellular sides for binding with and exchanging cations. Based on the sequence alignment and structural comparison, we identified the corresponding residues D201 that potentially responsible for Na$^+$ binding in SOS1 (Supplementary Fig. 9a). The complementation assay using salt-sensitive yeast indicates that the growth of AXT3K was marginally rescued upon SOS1 expression and notably augmented in the presence of SOS2-SOS3 protein kinase complex[35]. Nevertheless, the transport activity of SOS1 was eliminated by the D201A mutation, even in the presence of the SOS2-SOS3 complex, substantiating the significance of the D201$^{TM6}$ in cation association in both inward-facing and outward-facing conformations (Supplementary Fig. 9b). Strikingly, in the current SOS1 structure, D201$^{TM6}$ is approximately 5 Å away from the apex of the intracellular cavity and is not accessible by the solvent (Fig. 4a), indicating that the structure is stabilized in an occluded state, which prevents ion binding and thus inhibits the transporter.

To understand the molecular basis of the inaccessibility of D201$^{TM6}$, we compared the core domains of SOS1 and NHE1-CHP1 in the inward-facing conformational (NHE1-CHP1$^{IF}$) state, giving rise to an RMSD of 1.60 Å over 180 Cα atom pairs (Supplementary Fig. 9a). The arrangement of TM helices is nearly identical in the core domains of SOS1 and NHE1-CHP1$^{IF}$. However, TM5b exhibits a dramatic local conformational change, being displaced by 5 Å toward the dimerization domain compared with that of the NHE1-CHP1$^{IF}$ complex (Fig. 4b, c). Consequently, D201$^{TM6}$ is completely buried in a cavity composed of TM3, TM5b, and TM10 and cannot interact with ions from the intracellular side. The core domain of SOS1 is further compared with that of the NHE1-CHP1 complex in the outward-facing conformation (NHE1-CHP1$^{OF}$; Supplementary Fig. 9a). Strikingly, TM5b assumes a conformation similar to that of the NHE1-CHP1$^{OF}$ complex, indicating that SOS1 may be trapped in an occluded state, an intermediate state between inward- and outward-facing conformational states. Taking a closer look at the C-terminus of TM5b, we found that the intracellular portions of TM5b and TM6 are involved in extensive electrostatic interactions with the N-terminus of the juxtamembrane helix HC8 from another subunit (Fig. 4d). In particular, the salt-bridge bond between K187$^{TM5b}$ and E703$^{HC8}$ and a charge-dipole interaction between K704 and the C-cap of TM5b may participate in stabilizing this occluded conformation. Therefore, we mutated the charged residues ($^{702}$EEK$^{704}$)

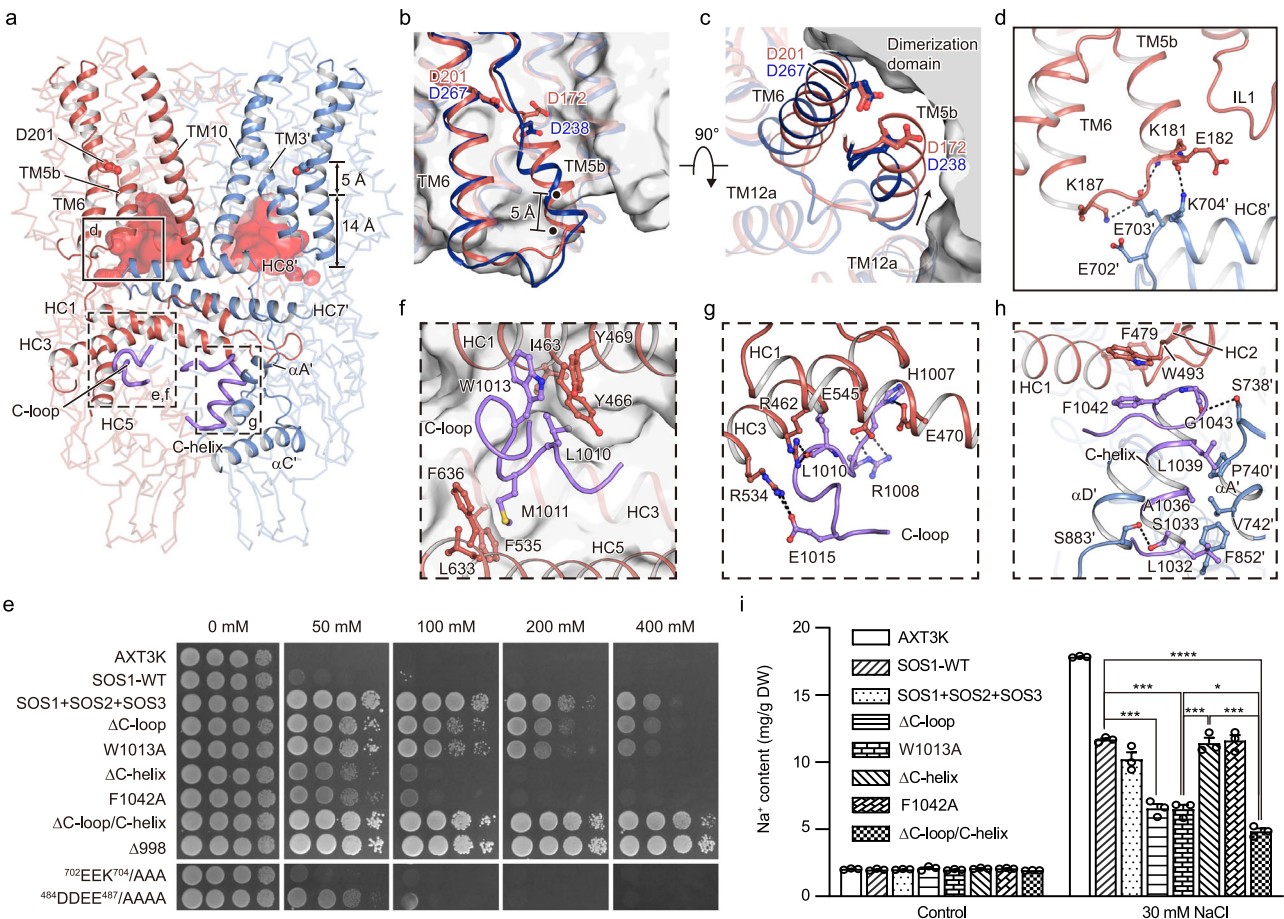

**Fig. 4 | C-loop and C-helix stabilize SOS1 in an occluded conformation.**
**a** Occluded conformation of SOS1, stabilized by the C-loop and the C-helix. The intracellular cavity between the dimerization domain and core domain is displayed with the electrostatic surface, and the depth of the cavity is labeled. The putative cation binding sites, D201[TM6], are shown as sticks. The C-loop and C-helix are displayed as tubes in purple. **b, c** Alignment of the core domain of inward-facing human NHE1-CHP1[IF] (deep blue) and SOS1 (orange). The core domain is shown in cartoon, and the dimerization domain of SOS1 is shown in a solid surface. The directions and distances of TM5b displacement are indicated. **d** Interactions among TM5b, TM6, and the N-terminus of HC8′ from the other protomer. Charged residues involved in the interactions are shown in sticks. The dashed line indicates the charge-dipole interaction between K704′ and the C-terminal of TM5b. **e** Salt tolerance test of AXT3K cells expressing solely the wild-type SOS1 (SOS1-WT), SOS1 together with SOS2-SOS3 (SOS1 + SOS2 + SOS3) and SOS1 mutants as indicated. Decimal dilutions of saturated cultures were plated in AP medium supplemented with 1 mM KCl and 0, 50, 100, 200, or 400 mM NaCl. The concentration of NaCl is labeled above the image. The growth of all transformants was indistinguishable in plates without NaCl. **f** Hydrophobic interactions between the C-loop and α-CTD. The hydrophobic cavity formed by HC1, HC3, and HC5 is shown as the surface. **g** Polar interactions between the C-loop and α-CTD. Hydrogen and salt-bridge bonds are shown as dashed lines. **h** C-helix interacts with HC1, HC2, and CNBLD′ from another subunit. **i** Intracellular Na⁺ content in the untransformed and transgenic yeast cells. Units are milligram of ion per gram dry weight of cell samples. DW: dry weight. Data are expressed as the means ± SEM (n = 3) and asterisks above the columns indicate significant differences. Each spot represents a single data value. Statistical significance was determined by two-side and unpaired t-test, without making any adjustments for multiple comparisons (*p < 0.05; **p < 0.01; ***p < 0.001; ***p < 0.0001). P value, SOS1-WT vs. ΔC-loop, 0.0001; SOS1-WT vs. W1013A, 0.0001; SOS1-WT vs. ΔC-loop/C-helix, <0.0001; W1013A vs. ΔC-helix, 0.0007; W1013A vs. ΔC-loop/C-helix, 0.0140; ΔC-helix vs. ΔC-loop/C-helix, 0.0001. Source data are provided as a Source Data file.

to alanine to disrupt these interactions, however, this mutation had little effect on the yeast salt-tolerant phenotype compared to wild-type SOS1 (Fig. 4e). We further speculate that in addition to these interactions, the restricted space between TMD and CTD may also impede the conformational change from an occluded state to an inward-facing state. We mutated the four consecutive acidic residues (484DDEE487) at the C-terminus of HC1 to alanine to rupture its interactions with the interface of α-CTD and CNBLD in the other subunit (Fig. 4e). The activity of this mutant was enhanced compared with that of the wild-type SOS1, indicating that this interaction is probably critical to inhibit SOS1 transport activity.

Previous studies indicated that the C-terminal tail plays key roles in inhibiting the transport activity of SOS1[7,38]. However, the majority of this tail is invisible in the cryo-EM map, presumably due to high flexibility. Interestingly, we identified two short segments composed of residues H1007–I1017 and L1030–V1048, which are termed the C-loop

and C-helix, respectively, according to their secondary structure (Fig. 4a and Supplementary Fig. 10a–c). Residues on the C-loop with large sidechain, such as M1011 and W1013 help to determine the peptide sequences and the sequence of C-helix was confirmed by secondary structure prediction and the large side chain residue of F1042. The C-loop is closely embedded in a cavity formed by the HC1, HC3, and HC5 helices from the α-CTD of the same subunit, mediated by hydrophobic and electrostatic interactions. In particular, residues L1010 and M1011 penetrate into two different hydrophobic pockets formed by I463[HC1]-Y466[HC1] and F535[HC3]-L633[HC5]-F636[HC5], respectively. The side chain of the highly conserved aromatic residue W1013 on the C-loop is close to residues Y466 and Y469 located on the N-terminus of HC1, forming a stack of π-π interactions with an edge-to-face geometry (Fig. 4f). Hydrogen bonds between the backbone carbonyl group of L1010 and the imidazole group of H1007 on the C-loop and side chains of R462 and E470 on HC1 further strengthen interactions between the

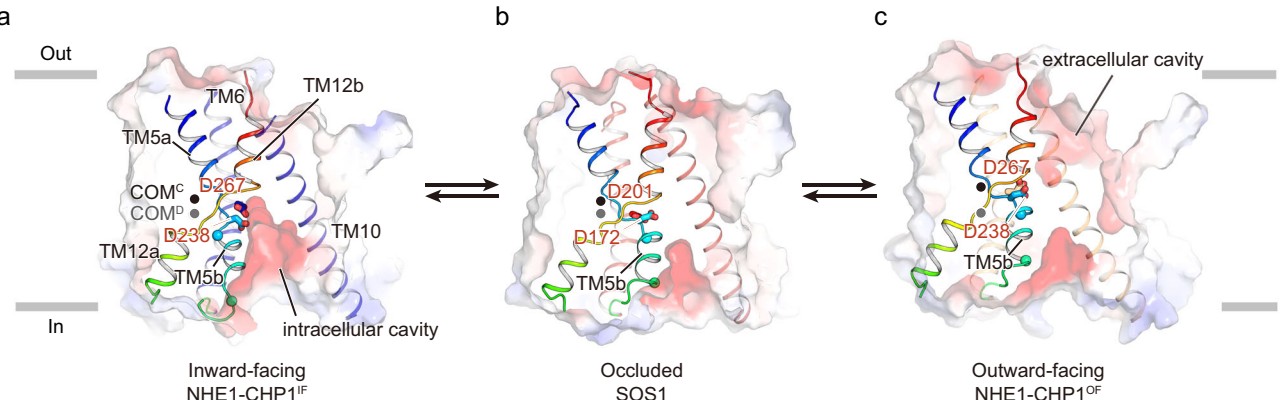

**Fig. 5 | Intracellular lateral gate in distinct conformations. a–c** Conformational transition of SOS1 and NHE1-CHP1 among the inward-facing state (**a**), the occluded state (**b**), and the outward-facing state (**c**), viewed in the membrane plane. The core domain is shown as surfaces and colored according to electrostatic potential. TM5, TM6, TM10, and TM12 are shown as cartoons. TM5 and TM12 are colored in rainbow. Asp172$^{SOS1}$ and Asp201$^{SOS1}$ (Asp238$^{NHE1}$ and Asp267$^{NHE1}$) are critical for ion binding and are labeled in red. Conformational change of TM5b is highlighted by showing the Cα of N-terminal (cyan, Pro173$^{SOS1}$ and Pro239$^{NHE1}$) and Cα of C-terminal (green, Lys181$^{SOS1}$ and Glu247$^{NHE1}$) residues in α-helical form as spheres. The centers of mass (COMs) of the core domain and dimerization domain are calculated using transmembrane helices and are shown as dots. COM$^C$, core domain, black; COM$^D$, dimerization domain, gray.

C-loop and α-CTD (Fig. 4g). Moreover, the salt bridges R1008-E545 and E1015-R534 mediate interactions of the C-loop and HC3 helix. The C-helix is located underneath the α-CTD and grips CNBLD from the same subunit and α-CTD from another subunit with extensive contact interfaces, thus stabilizing the dimerization of SOS1 and a specific geometry between α-CTD and CNBLD (Fig. 4h). Hydrophobic interactions are important to stabilize HC1 and HC2. For example, F1042 on the C-helix forms π-π stacking interactions with F479 and W493. Moreover, G1043 and S1033 of the C-helix also form hydrogen bonds with S738 and S883 of CNBLD. Taken together, the results show that C-loop and C-helix are involved in extensive intramolecular or intermolecular interactions with α-CTD and CNBLD and presumably stabilize the intracellular domain in a specific conformation to inhibit SOS1 activity. We speculate that the C-loop and C-helix may be important for the autoinhibitory regulation of the C-terminal tails.

To understand the functional roles of the C-loop and C-helix determined in the structure, we constructed internal deletions of C-loop (SOS1$^{ΔC-loop}$), C-helix (SOS1$^{ΔC-helix}$) or both (SOS1$^{ΔC-loop/C-helix}$) and performed functional complementation assays using the salt-sensitive yeast mutant strain AXT3K[11,39,40], which is unable to grow in nutrient medium supplementary with 50 mM NaCl (Fig. 4e, i). The expression level of these mutants is comparable to that of the wild-type (WT) SOS1, as evidenced by the Western blot analysis (Supplementary Fig. 11a, b). To further confirm that the salt tolerance of transgenic AXT3K is increased through enhanced Na⁺ efflux activity of SOS1, the intracellular Na⁺ content of cells expressing the SOS1 variants in AP medium with 30 mM NaCl was measured (Fig. 4i). Compared with WT SOS1, SOS1$^{ΔC-loop}$ upregulated SOS1 activity under salt stress conditions and significantly boosted the salt tolerance ability of transgenic yeast cells, allowing them to survive in medium containing a high concentration of 400 mM NaCl, which is similar to co-expression of the three SOS genes, while SOS1$^{ΔC-helix}$ only moderately enhanced transporter activity. However, the plaque growth of the combined deletion mutant SOS1$^{ΔC-loop/C-helix}$ was significantly better than that of the single mutant SOS1$^{ΔC-loop}$ or co-expression of the three SOS genes, which reached a high level similar to the hyperactivated truncation mutant at Q998[7]. Correspondingly, Na⁺ content in yeast cells transformed with SOS1$^{ΔC-loop/C-helix}$ was 26% lower than that in SOS1$^{ΔC-loop}$-expressing cells (Fig. 4i). Non-invasive micro-test technology (NMT) showed a significant enhancement of Na⁺ efflux in yeast expressing SOS1$^{ΔC-loop}$ and SOS1$^{ΔC-loop/C-helix}$ compared to WT SOS1 under salt conditions (Supplementary Fig. 12). Notably, the mean rate of Na⁺ efflux in the SOS1$^{ΔC-loop/C-helix}$ transformants exhibited 2-fold faster than the SOS1$^{ΔC-}$

$^{loop}$ transformants. These results suggest that these two fragments are cooperatively involved in the regulation of SOS1 activity, but the C-loop appears to play a more important regulatory role. Based on structural analysis, the W1013$^{C-loop}$ and F1042$^{C-helix}$ likely play a key role in locking the autoinhibitory conformation by stabilizing the N- and C-termini of HC1 through π-π stacking with surrounding aromatic residues, respectively. We substituted these two residues with alanine (SOS1$^{W1013A}$ and SOS1$^{F1042A}$). The SOS1$^{W1013A}$ and SOS1$^{F1042A}$ mutants improved the salt tolerance of transgenic yeast to a level equivalent to that achieved with SOS1$^{ΔC-loop}$ and SOS1$^{ΔC-helix}$, respectively, which is further supported by experiments measuring the relative intracellular Na⁺ content of these transformants (Fig. 4e, i).

In order to understand the conformational change of the transporter upon activation, we also carried out the structural study using constitutively activated C-terminal-truncated SOS1 (SOS1$^{Δ998}$) mutant[7]. The SOS1$^{Δ998}$ was recombinantly expressed and purified, which was subjected to cryo-EM analysis (Supplementary Fig. 13a–c). Despite the SEC profile and SDS PAGE analysis look decent, we could not resolve a reasonable cryo-EM map of the SOS1$^{Δ998}$. Specifically, the intracellular domains are completely blurred in the 2D classes and indistinguishable in the 3D map (Supplementary Fig. 13d), suggesting that the activated SOS1 is featured by highly flexible intracellular domains, instead of tightly packed domains found in the inhibited states. We thus speculate that the activation of the SOS1 through C-terminal phosphorylation, C-terminal truncation, and C-loop deletion may be due to disruption of the intramolecular and intermolecular interactions of intracellular domains.

## TM5b serves as an intracellular lateral gate

An elevator-like transport model has been proposed for Na⁺/H⁺ antiporters to carry Na⁺ ions from one side of the membrane to the other in exchange for protons based on the structural analyses of the inward- and outward-facing conformation structures. Recently, we reported atomic-resolution structures of the human NHE1-CHP1 complex in both inward-facing (NHE1-CHP1$^{IF}$) and cariporide-bound outward-facing (NHE1-CHP1$^{OF}$) conformations, in which the core domain undergoes significant conformational changes, including translation and rotation, relative to the dimerization domain (Fig. 5). TM5b of the core domain of the NHE1-CHP1$^{IF}$ complex forms a thinner 3₁₀ helix, and the hydrophobic middle part becomes fully stretched to significantly deepen and open the cavity to the intracellular side. Strikingly, TM5b is rearranged into an α-helix and lifted by a charge-dipole interaction between D238$^{TM5}$ and the N-cap of TM12b in the outward-facing NHE1-

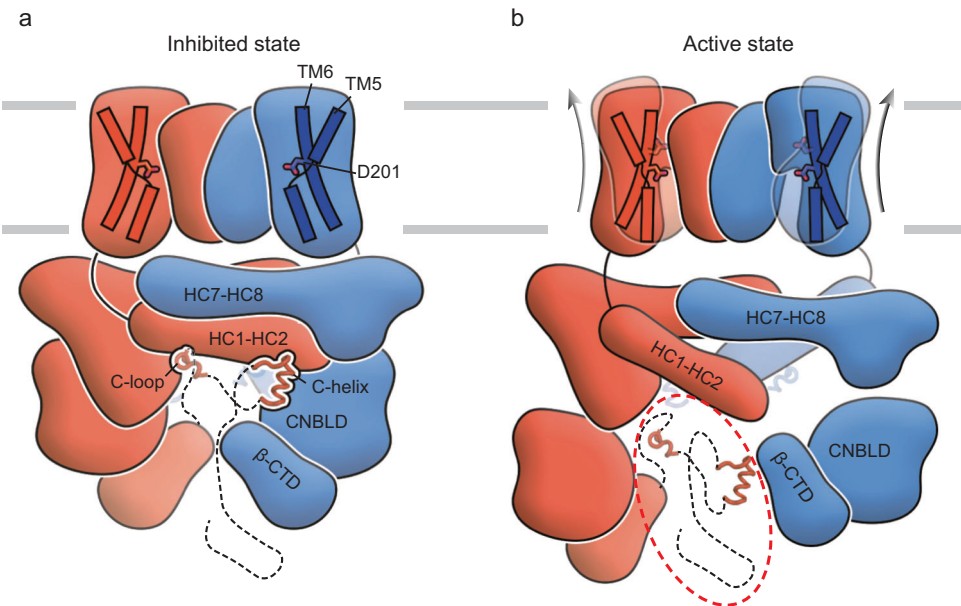

**Fig. 6 | Proposed model for SOS1 activation.** TM5 and TM6 helices from the core domains are illustrated and labeled. Side chains of the D201[TM6] are shown as sticks. Intracellular domains including the HCs, CNBLD, and β-CTD are represented according to their structure and labeled. The C-loops and C-helices are shown as tubes. The unresolved structure of the C-terminal tail is shown in dashed lines. **a** The inhibited state is shown where the intracellular domains are tightly packed together. The TM5b occludes D201[TM6] and makes it inaccessible from the

intracellular side. **b** The active state is demonstrated, where the intracellular domains are unpacked following the release of the C-loop and C-helices. The phosphorylated region is denoted by a red dotted circle. The core domain undergoes elevator-like movement between the inward-facing conformation and outward-facing conformation, depicted using opaque and transparent cartoons, respectively. Arrows indicate the movement of the core domain from the inward-facing state to the outward-facing state.

CHP1 complex. SOS1 harbors a negatively charged inward-facing cavity between the core and dimerization domains, and this intracellular cavity is shallower than that of the NHE1-CHP1[IF] complex. Interestingly, the structure of the core domain of SOS1, including TM5b, resembles that of the NHE1-CHP1[OF] complex rather than that of the NHE1-CHP1[IF] complex. In particular, TM5b forms an α-helix and moves close to the dimerization domains through the charge-dipole interaction between D172[TM5] and the N-terminus of TM12b. Consequently, the side chain of D201[TM6], equivalent to D267[TM6] in NHE1, is buried and inaccessible, indicating that the structure of SOS1 represents an occluded state. Based on these structural analyses, we proposed that TM5b acts as an intracellular lateral gate, governing the association or dissociation of ions and conformational change of the transporter.

## Discussion

SOS1 is the only known plasma membrane-localized Na[+]/H[+] antiporter identified to date in higher plants[41]. It is ubiquitously expressed in epidermal cells at the root tip and in parenchyma at the xylem-symplast boundary of roots, stems, and leaves and plays pivotal roles in maintaining ion homeostasis and controlling long-distance Na[+] transport via the xylem[42]. Soil salinity is becoming a severe environmental stress factor worldwide, and the major toxic cation present in saline soils is Na[+], which is seriously detrimental to plant growth and restricts agricultural production[43,44]. Therefore, SOS1 is emerging as a promising candidate to develop salt-tolerant transgenic crops. Despite there have been recent advances in the understanding functional roles of cytosolic C-terminal tail of eukaryotic Na[+]/H[+] antiporters (i.e., NHE1 and NHE3)[20,21], the autoinhibitory mechanism of SOS1 remains elusive.

Here, we determined high-resolution cryo-EM structures of the full-length wild-type *Arabidopsis thaliana* SOS1, revealing the architecture and subunit arrangement of this Na[+]/H[+] antiporter. Unlike the classic Na[+]/H[+] exchanger, SOS1 harbors a large intracellular domain consisting of a rectangle-shaped α-helical disk just below the membrane, a cyclic nucleotide-binding homology domain, and a C-terminal antiparallel β-sheet roll. We also determined SOS1 at two

conformations in which lipid molecules differentially insert into the extracellular cavity formed by two dimerization domains. In the SOS1[expand] structure, two extra lipids were determined to be buried in the dimerization interface, leading to extracellular expansion of the TMD. However, the differences in bound lipids did not result in significant conformational changes in the intracellular ends of TM helices and in the intracellular domains. According to the structural comparison, we found that SOS1 is locked in an occluded state by shifting of TM5b toward the dimerization domain, stabilized by interactions between TM5b and HC7-HC8 helices from the other subunit (Fig. 6). We proposed that TM5b may function as a lateral gate to control the accessibility of the ion binding pocket from the intracellular side. Moreover, HC1 directly connects TM13 from the core domain. As reported in a previous study, the core domain is supposed to undergo upward or downward movement relative to the dimerization domain during the transport cycle[23]. We speculate that the close packing between TMD and intracellular domain would prohibit the core domain from sliding across the membrane, thus inhibiting the activity of SOS1.

Two discrete fragments, C-loop (residues 1007–1017) and C-helix (residues 1030–1048), were determined to interact with intracellular domains from different subunits. We speculate that interactions of C-loop and C-loop with other domains might contribute to stabilize dimerization of the intracellular domains. Deletion of C-loop and/or C-helix (SOS1[ΔC-helix], SOS1[ΔC-helix]) and mutation of W1013 or F1042 significantly stimulate the transport activity of SOS1, demonstrating C-loop and C-helix serve as a crucial molecular switch to regulate the activity of this antiporter. To gain insights into the activation mechanism of SOS1, we also made a constitutively active construct by truncating C-terminal autoinhibitory tail (SOS1[Δ998]) and carried out cryo-EM study. It turns out that intracellular domains are completely blurred during 2D and 3D classification, preventing us from obtaining a high-resolution map and suggesting that intracellular domains may become highly mobile upon SOS1 activation. Interestingly, we found that the surface electrostatic potential of the intracellular domains is

negatively charged (Supplementary Fig. 14). Considering the critical roles of C-loop/C-helix and phosphorylation of C terminal tail in activation of SOS1, we further hypothesis that the phosphorylation of the C-terminal tail would introduce additional negative charge(s) that would repel negatively charged intracellular domains, reducing binding affinity of the C-loop and C-helix to the intracellular domain and destabilizing dimerization of the intracellular domain, thereby activating the SOS1. Previous studies have identified S1138 as the phosphorylation site and highlighted the role of S1136 in localizing SOS2 to the phosphorylation site[7]. However, we found that the serine-to-alanine mutant SOS1[S1136A/S1138A] conferred the same degree of salt tolerance in yeast as the wild-type SOS1 in the presence of the SOS2-SOS3 complex (Supplementary Fig. 15). This finding strongly suggests the existence of additional potential phosphorylation site(s) involved in SOS1 activation by the SOS2-SOS3 complex that have not yet been clearly identified. Further investigations are required to map the specific locations of these additional phosphorylation sites and gain valuable insights into the activation mechanism.

## Methods

### Cloning, expression, and purification of the *Arabidopsis* SOS1

The gene encoding *Arabidopsis. thaliana* SOS1 (UniProtKB accession: Q9LKW9) was amplified from a cDNA library of *A. thaliana* ecotype Columbia (Col-0) and subcloned into a pEG-BacMam expression vector. To facilitate detection and purification of the SOS1, a super-folder green fluorescent protein (sfGFP) and a Twin-Strep tag were introduced at the C-terminus of the SOS1, following by a PreScission Protease (PPase) recognition site. Recombinant proteins were expressed in HEK293F cells (Thermofisher, 11625019) using the Bac-to-Bac baculovirus expression system (Invitrogen, USA). Specifically, P2 viruses generated in *Sf*9 insect cells (Thermofisher, 10902096) were used to infect HEK293F at a density of $2.2–2.5\times10^6$ cells mL$^{-1}$ with 1% (v/v) fetal bovine serum (FBS). Transfected HEK293F cells were cultured in a shaker with 5% $CO_2$ at 37 °C. After 12 h, 10 mM sodium butyrate was added to the medium and the cells were incubated for another 48 h before harvesting. Harvested cells were snap-frozen in liquid nitrogen and resuspended in a lysis buffer (20 mM HEPES pH 7.5, 150 mM NaCl, 5 mM β-mercaptoethanol (β-ME), 2 μM leupeptin, 0.8 μM pepstatin, and 2 μM aprotinin (MedChemExpress)), followed by lysing with a Dounce homogenizer. The membrane fraction was isolated by ultracentrifugation at 45,000 g for 40 min and immediately resuspended into a solubilization buffer (20 mM HEPES pH 7.5, 150 mM NaCl, 1% (w/v) digitonin, 5 mM β-ME, 5 mM ATP, 5 mM $MgCl_2$, 2 μM leupeptin, 0.8 μM pepstatin, and 2 μM aprotinin) for 3 h with agitation. Insoluble debris was removed by ultracentrifugation at 45,000 g for 40 min, and the supernatant was subjected to affinity chromatography column with Streptactin Beads (Smart-Lifesciences, China) at a flow rate of 1 mL/min. The beads were washed with 10 column volumes of washing buffer (20 mM HEPES pH 7.5, 150 mM NaCl, 5 mM β-ME, 0.03% (w/v) glycol-diosgenin (GDN), 2 μM leupeptin, 0.8 μM pepstatin, and 2 μM aprotinin) to remove nonspecific bound protein and then the SOS1 protein was eluted with the washing buffer supplemented with 5 mM desthiobiotin. The eluate was incubated together with PPase for 3 h to improve the sample homogeneity, concentrated with 100-kDa cut-off Amicon Ultra centrifugal filter (Merck Millipore, Germany) to 1 mL and further purified by size exclusion chromatography (SEC) using a Superose 6 Increase 10/300 GL column (GE Healthcare, USA) with a running buffer containing 20 mM HEPES pH 7.5, 150 mM NaCl, 5 mM β-ME and 0.007% (w/v) glycodiosgenin (GDN). The peak fractions were collected, concentrated to ~6 mg/mL for cryo-EM grid preparation. The yield of protein was approximately 0.2 mg/L of HEK293F cell culture. All processes for protein purification were carried out at 4 °C. The purified protein was also analyzed using High Performance Liquid Chromatography (HPLC) and Coomassie-stained SDS-PAGE.

The truncated SOS1$^{\Delta998}$ construct was incorporated from existing full-length wild-type SOS1, with a PPase cleavage site, sfGFP and Twinstrep tags being fused to the C-terminus of the gene product. The above described method was followed for the truncated protein purification.

### Cryo-EM grid preparation and data collection

A droplet (2.5 μL) of SOS1 protein sample was applied to the holey carbon grids (Cu R1.2/1.3 300 mesh, Quantifoil), which were glow-discharged beforehand in $H_2$-$O_2$ condition for 1 min using a Solarus plasma cleaner (Gatan, USA). The grids were then blotted with a Vitrobot Mark IV (Thermo Fisher Scientific, USA) for 4.5 s under 4 °C and 100% humidity and flash-frozen in liquid ethane cooled by liquid nitrogen. Cryo-EM data were collected on a 300-kV FEI Titan Krios transmission electron microscope (Thermo Fisher Scientific, USA) equipped with a Gatan K2 Summit direct electron detector and a GIF quantum energy filter. The SerialEM[45] was used to collect movie stacks automatically at a magnification of 130,000×, which corresponds to a pixel size of 0.52 Å across a defocus range of −1.2 μm to −2.2 μm. Each movie stack was dose-fractionated in 32 frames with a total dose of 60 e⁻/Å$^2$. The dose rate was set to 9.0 e⁻/pixel/s.

### Data processing

A total of 1,552 movies were collected for the structural analysis of SOS1. The movies were subjected to beam-induced motion correction and dose-weighted using MotionCor2 with 5 × 5 patching[46]. Contrast transfer function (CTF) parameters for each micrograph were determined by Gctf[47]. Particle picking was conducted by Gautomatch and the integrated blob picker and template picker in cryoSPARC, generating a dataset of 1,254,798 particles. Rounds of reference-free 2D classification were performed in cryoSPARC to clean particles, followed by Ab-initio reconstruction to generate a reference 3D map. Reference-free 2D classification resolved different projections and clearly showed that SOS1 consists of a transmembrane domain (TMD) and a large cytoplasmic domain (CTD) with distinguishable structural features, including transmembrane and intracellular helices. Particles were then imported into RELION-3.1 and cleaned up by rounds of 3D classifications. The initial multi-reference 3D classification in RELION was performed on a 4 × 4 binned data set against a good map and four biased maps, generating 5 classes. Class 5 (40.3%), which was calculated using the good reference, displayed structural features of the transmembrane and intracellular domains. Particles from class 5 were then re-extracted with 2 × 2 binning and subjected to another round of multi-reference 3D classification in RELION with the same reference set. Particles from the best class accounted for 67.6% of total input and displayed well-resolved structural features, including continuous transmembrane helices and α-helices within the intracellular domains. After beam-tilt CTF refinement, Bayesian polishing, and 3D refinement, a high-resolution density map was reconstructed in RELION and reported at 3.5 Å according to the golden-standard *Fourier* shell correlation (GSFSC) criterion. These particles were subsequently submitted to another round of skip-align 3D classification in RELION, resulting in four reconstructions at overall resolutions of 5.5 Å, 3.1 Å, 7.5 Å, and 4.0 Å, respectively. Particles from these classes, namely, 290,003 particles (58.6%) from class I, 151,646 particles (21.5%) from class II, and 80,155 particles (12.4%) from class IV, were imported into cryoSPARC for further processing. Heterogeneous refinements were performed on the classes individually to improve map quality. The final maps were generated by non-uniform refinement in cryoSPARC, which generated a class I map at 3.5 Å, a class II map at 2.8 Å, and a class IV map at 3.4 Å, all based on the GSFSC criterion. A diagram of data processing is presented in Supplementary Figure 2.

For the truncated SOS1$^{\Delta998}$ protein, a total of 931,449 particles were initially picked from 1,312 micrographs, followed by particle

cleaning using 2D classification. In contrast to the full-length SOS1, the intracellular part of SOS1$^{\Delta 998}$ in 2D micrographs is completely blurred. The helical features of the transmembrane region are also ambiguous. Ab initio reconstruction was conducted to generate an initial map. 3D classification against the low-resolution map and four trash maps was carried out with the application of C2 symmetry, which yielded five classes. The most populated class was composed of 26% of total particles but the TM helices cannot be clearly displayed. Particles from this class were selected and submitted to further 3D classification focusing on TMD domain. However, none of the six classes displaying resolved transmembrane helices and we did not obtain a map with clear TMD density because the completely disordered cytoplasmic structure broken the alignment of the particles. All procedures of data processing were conducted in cryoSPARC.

## Model building

To build the atomic model of the SOS1, the transmembrane helices of a NHE1 (PDB ID: 7DSX) protomer was fitted into the density map of SOS1 using UCSF Chimera[48] as the initial model and then were manually inspected and rebuild in Coot[49]. The remaining parts of SOS1 were built de novo according to the cryo-EM density and secondary structure prediction. The two discrete cytoplasmic segments containing residues 1007–1017 (C-loop) and 1030–1048 (C-helix) was clearly determined with well-resolved densities for mainchain and bulky side chains, which enabled us to unambiguously build the atomic model. However, the chain ID of the C-loop and C-helix is not clearly determined based on the cryo-EM map, due to linkers connecting β-CTD, C-loop and C-helix are missing. As shown in Supplementary Figure 10, The distances between I972 from A subunit and H1007 from two C-loops are about 27 Å and 24 Å, respectively. Considering 35 amino acids missing between these two termini, we speculate that both two C-loop segments possibly belong to chain A. The C-loop interacting with HC1, HC3, and HC5 from A subunit was set as chain A. The distances between I1017 on C-loop from chain A and L1030 from two C-helix are 16 Å and 45 Å, respectively. Considering only 12 residues missing between C-loop and C-helix, we speculate that the C-helix interacting with CNBLD from chain B belongs to chain A. To generate a dimerized SOS1 model, the atomic model of a SOS1 protomer was copied and manually fit into the densities according to the symmetry axis. Manual refinement and real-space refinement in PHENIX[50] were performed iteratively to further improve the model. The hydrocarbon chain of lipid was added manually in Coot. The geometries of the structures were valuated using PHENIX. The model vs. map Fourier shell correlation curve was calculated using the comprehensive validation (cryo-EM) in PHENIX. Figures were prepared in PyMOL[51] and UCSF Chimera. Statistics for cryo-EM data collection and model refinement are summarized in Supplementary Table 1.

## Yeast complementary assay

To analyze the mutational effects of SOS1 variants on the antiport activity, colony growth, and Na$^+$ content determination were performed using the *Saccharomyces cerevisiae* mutant strain AXT3K (*4ena1::HIS3::4ena4, 4nha1::LEU2*, and *4nhx1::KanMX4*), which is deficient in the Na$^+$ efflux proteins ENA1−4 and NHA1, and the vacuolar Na$^+$/H$^+$ antiporter NHX1[52]. Full-length *Arabidopsis thaliana* SOS1 was cloned into the plasmid pYPGE15 with a C-terminal 6×His tag. Nine different constructs of SOS1 were made, containing five mutations (D201A, W1013A, F1042A, $^{702}$EEK$^{704}$/AAA, $^{484}$DDEE$^{487}$/AAAA), three internal deletions (SOS1$^{\Delta C\text{-loop}}$, SOS1$^{\Delta C\text{-helix}}$ and SOS1$^{\Delta C\text{-loop/C-helix}}$) and a truncation construction (SOS1$^{\Delta 998}$). All the mutations were generated by overlap extension PCR and sequenced prior to use. The forward primers and the reverse primers for the modifications are summarized in Supplementary Table 2. Plasmids pYPGE15 containing the variant gene were transformed into AXT3K using the PEG/LiAc method[53]. In both types of assays, co-transformed with pYPGE15-AtSOS1 and

p414-AtSOS2-AtSOS3 was used as the positive control, and the untransformed AXT3K was used as a negative control.

The NaCl-resistance colony assay were performed in AP medium (8 mM phosphoric acid, 10 mM arginine, 2% glucose, 2 mM MgSO$_4$, 1 mM KCl, 0.2 mM CaCl$_2$, plus trace elements and vitamins, adjusted to pH 6.5 with arginine). The transgenic and untransformed yeast cells were precultured in liquid YPD medium until OD$_{600}$ = 1.0 and diluted to $10^{-1}$, $10^{-2}$, $10^{-3}$, $10^{-4}$, $10^{-5}$ successively. 4.5 µL aliquots of each serial 10-fold dilutions of yeast cells were spotted onto AP plates supplemented with 0, 50, 100, 200 or 400 mM NaCl. After incubation for 3−5 days at 28 °C, the growth of yeast cells on plates was imaged and analyzed.

## Na$^+$ content determination

In the Na$^+$ content determination, yeast cells were grown either with or without 30 mM NaCl in 2.5 L of AP medium at 28 °C with shaking (200 rpm). The cells were harvested by centrifugation (3000 g, 5 min) when the OD$_{600}$ reached $0.25 \pm 0.01$. After treatment with ice-cold washing buffer (10 mM MgCl$_2$, 10 mM CaCl$_2$ and 1 mM HEPES)[38], the yeast cells were then oven-dried at 85 °C for 48 h and weighed, after which the samples were digested with 10 mL of HNO$_3$ and H$_2$O$_2$ overnight. And then the Na$^+$ content was determined using an atomic absorption spectrometer (Agilent, USA). All data are reported as the mean ± SEM. Data analyses were performed using Prism 9 (GraphPad, USA). The original data was shown in Supplementary Table 3. The functional data is not normalized by the expression level.

## Measurement of Na$^+$ efflux with non-invasive micro-test technology

The real-time Na$^+$ efflux from the yeast cells under salt stress was measured using NMT[54,55] (NMT Physiolyzer, YoungerUSA, MA, USA). Yeast cells were first cultivated in 5 mL AP medium (8 mM phosphoric acid, 2% glucose, 2 mM MgSO$_4$, 1 mM KCl, 0.2 mM CaCl$_2$, plus trace elements and vitamins, adjusted to pH 6.5 with arginine) at 28 °C for 24 h with shaking (200 rpm), and then transferred into 10 mL fresh medium with a final concentration of 100 mM NaCl and allowed to grow at 28 °C for 2 h with shaking (200 rpm), after which 0.5 mL nutrient solution with yeast cells were harvested by centrifuge (3000 × $g$, 5 min), the pellet was resuspended in 0.2 mL measuring buffer (100 mM mannitol, 1.0 mM NaCl, 0.2 mM MES, pH 6.3). Na$^+$-flux microsensor were positioned 5 µm away from the cells enriched by the conical filter membrane, and a continuous flux recording was taken for 5 min for each sample using imFluxes V3.0 software (Xuyue Company, Beijing, China). All data are reported as the mean ± SEM. Data analyses were performed using Prism 9 (GraphPad, USA). The experiment was repeated three times.

## Western blot analysis

For Western blot analyses, yeast cells were cultivated in 10 mL AP medium at 28 °C with shaking (200 rpm) and harvested by centrifugation at 1500 × $g$ after 24 h. The cell pellet was resuspended in ice-cold disruption buffer composed of 25 mM Tris, 150 mM NaCl (pH 8.0), supplemented with 2 µM leupeptin, 0.8 µM pepstatin, 2 µM aprotinin (MedChemExpress) and 1 mM PMSF (Sangon Biotech). Cell lysates were prepared by vigorous vertexing with glass beads in a mixer mill for 10 × 3 min at 4 °C, followed by centrifugation at 13,000 × $g$ for 30 min at 4 °C. The pellet was resuspended and incubated with 2×SDS sample loading buffer for 5 min prior to subjecting the denatured protein samples to 8% SDS-PAGE. Subsequently, the proteins were transferred to nitrocellulose blotting membranes (GE health, USA) for immunoblotting. The membranes were blocked with 5% nonfat skim milk for 1 h and then incubated with the specified primary mouse monoclonal antibodies: anti-His (1:6000 dilution, CW0286M, CoWin Biotech, Beijing, China) and anti-β-tubulin (1:6000 dilution, ab184970, Abcam, UK) in PBS containing 0.1% Tween 20,

followed by extensive washes and incubation with corresponding horseradish peroxidase–conjugated secondary antibodies goat anti-mouse (1:6000 dilution, CW0102S, CoWin Biotech, Beijing, China) and goat anti-rabbit (1:6000 dilution, CW0103S, CoWin Biotech, Beijing, China) for 1 h at room temperature. Immunoreactive bands were detected using Western Lightning Plus-ECL blotting detection reagents (Tanon, China).

## Reporting summary
Further information on research design is available in the Nature Portfolio Reporting Summary linked to this article.

## Data availability
The cryo-EM density map of the SOS1 and SOS1$^{expand}$ have been deposited in the Electron Microscopy Data Bank (EMDB) under the accession code EMD-33592 and EMD-35085, respectively. The coordinate for the SOS1 and SOS1$^{expand}$ have been deposited in the Protein Data Bank (PDB) under the PDB ID 7Y3E and 8HYA, respectively. The sequence of *Arabidopsis thaliana* SOS1 is available in the Universal Protein Resource (UniProt) databases under accession code Q9LKW9. Source data are provided with this paper.

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

## Acknowledgements

We thank Xiaojun Huang, Boling Zhu, Longlong Zhang, and other staff members at the Center for Biological Imaging (CBI), Core Facilities for Protein Science at the Institute of Biophysics, Chinese Academy of Science (IBP, CAS) for their support in cryo-EM data collection. We thank Yan Wu for his research assistance. This work is funded by the Chinese Academy of Sciences Strategic Priority Research Program (Grant No. XDB37 030304 to Y.Z.), National Key Research and Development Program of China (Grant No. 2021YFA1301501 to Y.Z.; 2018YFE0207203-2 and 2018YFD1000500 to X.J.), the National Natural Science Foundation of China (Grant No. 92157102 to Y.Z.; 31660253 to X.J.; 32200978 to L.H.), the Youth Innovation Promotion Association of the Chinese Academy of Sciences (Grant No. 2022089 to L.H.), and Chinese National Programs for Brain Science and Brain-like Intelligence Technology (Grant No. 2022ZD0205800 to Y.Z.).

## Author contributions

Y.Z. conceived the project and supervised the research. Y.W. and L.H. carried out the molecular cloning experiments. Y.W. expressed and purified protein samples. Y.D. prepared samples for cryo-EM study. D.Y., Y.W., and Y.G. carried out cryo-EM data collection. Y.W. and Q.C. processed the cryo-EM data. Y.W. and Y.L. built and refined the atomic model. Y.Z., Y.W., and Y.D. analyzed the structure. X.J., Q.X., and C.P. designed and conducted the yeast experiments. Y.W. wrote the original draft of the manuscript and prepared the figures. Y.Z., Y.W., Y.G., and Q.C. edited the manuscript with input from all authors.

## Competing interests

The authors declare no competing interests.
