## [Peer Review File · Nature Communications]

REVIEWER COMMENTS

Reviewer #1 (Remarks to the Author):

This is a nice report about the long-awaited structure of the Na⁺/H⁺ antiporter SOS1. This molecule is the major determinant of salt tolerance in plants.

I have enjoyed reading the article, the introduction is clear and informative, the experimental approaches are sound and the methods section is complete. However, the conclusions derived from the structural work do not adjust to the expectations. The work provides the basis for the self-inhibitory activity of SOS1, this is quite a lot, but the amount of new information is limited to the identification of the boundaries of the already reported self-inhibitory domain of the antiporter. In addition, there are several issues that there are vaguely discussed and that deserve more attention. Among them, are the activation mechanism of SOS1, the specificity of the channel (Na⁺ vs K⁺) and the role of the CNBD.

A.- Following the authors' arguments, one might think that the phosphorylation of the C-terminal end of the cytosolic domain would lead to the release of the delta C loop and delta C-helix and this, in turn, would destabilize the dimer structure allowing a reorganization of the TMD to a productive conformation (lines 324-326). This is too simplistic and far away from the experimental data showing that the truncation of the delta C-loop and delta C-helix yields a hyperactive antiporter. There is a complex regulation of SOS1 (and other NHX transporters) that involves different proteins in response to a variety of stimuli.

The work would benefit from additional structural work (even a bioinformatic model) about the transition from the inactive to the active conformations of the antiporter. Where are the hinge points allowing this transition? Data processing of the discarded classes might provide information on this aspect (see below).

B.- There is no mention of the important issue of the selectivity of SOS1 for Na⁺ vs K⁺. Some other plant NHX transporters display an enhanced preference for K⁺ or display both K⁺ and Na⁺ transport activity. The interplay between the activity of these transporters is central to the control of cell ion homeostasis under salt stress.

C.- The CNBD deserves more attention than a supplemental figure S4. Mutations G777E and G784D lead to an inactive transporter (Shi et al. PNAS 2000) which is the basis of such a role? The authors could compare the CNBD structure to that of the CNBD of plant AKT1 (Lu et al. Nat Com. 2022).

D.- There is care in the preparation of the figures but it is difficult to follow the nicely described structural features of the macromolecule.

The manuscript might be improved with a revision concerning the above-mentioned aspects and the following points.

1.- The purification protocol seems to yield a large quantity of homogeneous protein. Please provide details about the amount of protein in mg per volume of culture. Line 354, I understand that the GFP and the twin-Strep tag were introduced "following" (not "followed") the PreScission Protease (PPase) recognition site.

2- The data processing protocol is biased towards the description of the procedures to obtain the self-inhibited structure of SOS1. However, I wonder if the authors have explored enough the large conformational landscape of the observed particles. In particular, they discard around 80% of the particles (they used 151,649 particles out of 543439) that could contain valuable information about conformational changes that SOS1 might undergo to develop its function. Indeed, I have got the impression from the central panel of figure S2 showing four classes, that the discarded classes display different conformation at the cytosolic moiety of the molecule. The authors should process such particles and analyze the resulting structures or, alternatively, provide a convincing argument to discard them. In

addition, would it be possible to mask the transmembrane domain of the truncated protein version and produce a structural model?

3.- The authors employed a construct including a GFP protein at the C-terminal end of the truncated protein. I wonder if this might be the origin of the inhomogeneity observed for this preparation as, in contrast to the wild-type protein, the GFP might be too close to the beta-CTD.

4.- Lines 221 to 224. Is SOS1 trapped in an Inward facing conformation or in an intermediate state between inward and outward-facing conformation? Please clarify.

5.- Supplementary Fig. 4. Please color the sticks by element type so the reader can visualize the differences between SOS1 CNBD and a canonical one.

6.- Lines 288 to 293. The authors speculate about the influence on the phosphorylation-mediated activation mechanism of a negatively charged patch around the C-loop. They should use the salt tolerance test of AXT3K cells to prove this statement.

Other minor comments

7.- Line 416. "151,646 particles" not "151,646k particles"

8.- Fig. 1a (line 110) is called later than Fig. 1b. Supplementary fig 2 goes to supplementary fig 5d

9.- Line 215. "NHE-CHP1" not "NHE-CHP11IF"

10.- Please homogenize cation calls. Either Na⁺ or sodium, K⁺ or potassium

Reviewer #2 (Remarks to the Author):

The Na/H exchanger SOS1 is a critical component of the SOS pathway for salt tolerance that is ubiquitous in plants. Although the pore domain of SOS1 can be reliably modeled using as template known structures of phylogenetically similar proteins (e.g. animal NHEs) or by the AlphaFold software, the SOS1 protein is likely to possess regulatory roles that go beyond the mere ion exchange for sodium extrusion, and which may involve the unique C-terminal of this protein. Therefore, obtaining the refined cryo-EM structure of SOS1 could have been a major achievement in plant physiology. However, this report is insufficient to claim success, as it suffers from limitations that curtail the novelty and the potential impact. Below, I will try to explain the limitations of this work.

1. The main contribution of this work is a structural description of the mechanism underlying the autoinhibition of SOS1 by two segments of the cytosolic domain that repress the activity of SOS1 to keep the protein in a resting state. However, no biochemical analyses are performed to shed light on the differences between repressed, derepressed and activated proteins.

2. The SOS1 protein is 1146 residues long. The cryo-EM structure of SOS1 is only partial, comprising the protein stretches 32–972, 1007–1017 and 1030–1048. The structure of the phosphorylation domain, the interplay with the auto-inhibitory domain, and a functional description of how SOS1 becomes active are all missing.

3. The CNBLD of SOS1 has primary sequence homology with bona fide CNBD of the hHCN1 channel. Authors show that critical residues for cAMP binding in the CNBD pocket of hHCN1 are not really conserved in SOS1, and that some of these important residues are substituted

by larger ones imposing a steric hindrance. Authors conclude that SOS1 cannot possibly bind cAMP, but they do not make any attempt, not even present a speculative hypothesis, to provide a clue on what the agonist is, if any. The original report of the SOS1 gene sequence described mutations in the putative CNBLD that behaved as lost-of-function mutants (Shi et al., 2000, PNAS). Authors could try to provide a structural basis for these findings.

4. A large proportion of this report is devoted to give a structural framework to the mechanism of auto-inhibition that keeps SOS1 in a resting state. This is the only part in which authors took the effort to corroborate hypotheses by conducting functional studies. There are however several limitations here too. Firstly, the structure of the auto-inhibitory domain is based not on original cryo-EM data but on the docking of the AlphaFold2 prediction for this domain into the SOS1 structure. Secondly, the C-loop and C-helix domains conforming the auto-inhibitory domain reported here largely correspond to the bipartite auto-inhibitory domain previously identified by Quintero et al., 2011, PNAS, which detracts from novelty. Dissecting the two halves of the bipartite auto-inhibitory domain into what is called here C-loop and C-helix was also reported there. The real advance in this report is the identification of W1013 and F1042 as the critical residues in C-loop and C-helix, respectively.

5. Growth and sodium content data of yeast expressing mutated/truncated SOS1 proteins attributed to differential transport activity should be completed, at the very least, with the inspection of protein abundance, and preferably with direct measurements of transport activity. I could not find whether the mutant proteins named Δ C-loop and Δ C-helix corresponded to internal deletions or protein truncations; nothing in methods on how these mutations were constructed.

6. The purpose of the lengthy explanation of interactions between HC7-HC8 and TMs is unclear because, at the end, it only leads to the suggestion that these interactions are crucial to regulation SOS1 activity. However, no solid rationale or experimental evidence is given for this proposal.

7. No mutagenesis and functional analyses have been performed to substantiate some of the structure-based speculations on SOS1 activity (e.g., residues presumably interacting with lipid molecules, to which authors refer to as 'essential for the stability of the SOS1 dimer', or the role of D201 in substrate coordination).

8. Description of the residue interactions that authors consider important are sometimes hard to follow because figures do not represent with sufficient clarity (or not shown at all) what the text says (e.g. AH helices, see below).

Other minor issues are:

9. No description of how SOS1 was purified from HEK cells.

10. Ln 21. Change text to 'We report the PARTIAL structure of SOS1' to give a realistic view of the progress.

11. Ln 65-66. First report on CBL10 mediated regulation of SOS1 is Quan et al. (2007), Plant Cell 19: 1415-1431.

12. Ln 84-85. Cannot understand how authors concluded from 'mushroom-like' 2D pictures that SOS1 consisted of a transmembrane domain (TMD) and a large cytoplasmic domain (CTD).

13. Ln 101-102. Amphipathic helix AH3-4 is not shown neither in Fig 1d nor Suppl 5d.

14. Ln 117-118. HC7 is not shown in Suppl Fig 3a

15. Ln 131-132. Text refers to HC7-HC8 but Suppl. Fig 3e shows HC6

16. Ln 160-161. Revise broken sentence.

17. Ln 208-219 and Fig 3B. To what protein the grey background structure corresponds to, NHE1 or SOS1? Whereas the displacement of TM5b of SOS1 relative to that of NHE1 is evident, residues D201 of SOS1 and D267 of NHE1 occupy the same position in the inner cavity.

18. Ln 289-291 and Suppl Fig 7. The phosphorylation domain is missing in the SOS1 structure. Hence, the speculation regarding the effect of the phosphate charge on the interaction between the C-loop and cytoplasmic domain has no basis.

19. In Suppl Fig 7, the C-loop and C-helix domains are not labeled. The blue-color cartoon structure is hard to see against the electrostatic potential surface representation.

20. Ln 340-341. Revise sentence, NHE1-CHP1-OF named twice.

Reviewer #3 (Remarks to the Author):

In the manuscript of Wang et al. the authors describe the structure of salt-overly-sensitive 1 (SOS1) protein. This Na⁺/H⁺ antiporter belongs to the superfamily of cation proton antiporters (CPA) and is essential in higher plants to cope with salt stress. While several structures of Na⁺/H⁺ antiporter have been solved over the past decades describing their overall architecture and mode of transport, this study provides new and interesting insights into unique regulation mechanisms.

The manuscript is largely composed of a detailed description of the structure and single elements but lacks a better final proposition of a model and a broader comparison and perspective to the superfamily and the transporter field. The manuscript could be substantially strengthened if the authors could comment on/incorporate the following:

- An introduction and discussion on the composition of the cytoplasmic domain (CTD). Which Na⁺/H⁺ antiporter have such a CTD, including for example a CNBLD? The authors argue from the given structure why the CNBLD presumably cannot bind a cyclic-nucleotide, but has this been confirmed biochemically or structurally?**
- Are the exact residues known which are phosphorylated in the autoinhibitory region? This becomes largely important for the storytelling and conclusion drawn by the authors. If not known than the authors could have screened the few potential positions through their complementation assay.**
- If I understand the authors claims correctly, then the biggest determinant for the inhibition is the stabilization of the dimeric cytosolic domain, which would sterically hinder the elevator mechanism of the core domain. This in turn would make the interaction of the TM5 with HC8 less relevant, as also shown by the functional complementation assays. While I find the description of TM5b as an intracellular lateral gate interesting and important the current order of manuscript sections feels off. Better to finish with a short proper conclusion section at the end including all relevant elements.**
- Functional assays could be complemented by combination of 702EEK704 and 484DDEE487 alanine mutations, as the authors claim that both the detachment of TM5b from HC8 and the interactions of HC1 are required to activate the protein, while as single mutations not enough.**
- The description on the structural attempts for the truncated SOS1 construct is poor and not entirely convincing. First the SEC profile, shows a broader peak, with a potential right shoulder, which could also be indicative of presence of monomers (in particular considering**

it was done on a superose 6). More importantly however, while I agree with the authors that the cytosolic domain is not visible, I also do not see any evidence or clear density for the TMD in the 2D classes (there are also no top views shown). The latter makes one wonder if the entire protein might have been unstable or the data set too small for proper processing and thus relevant to draw any conclusions from it. Can the authors comment on how many images were acquired and particles processed? This needs to be properly revisited, as it represents the basis for an important statement: namely that the entire CTD is flexible, potentially even no longer dimerized, without the autoinhibitory region. Otherwise, most of the claims throughout the text (e.g. 324-326) also with regards to stabilization of the dimer interface need to be toned down.

- Another aspect why the structural attempt of the truncated SOS1 is unsatisfactory, is because even if the CTD are flexible and does not visible in the cryo-EM data, it would be highly interesting to see which state the transporter domain adopts under these conditions (and outward or inward facing instead of occluded state, with an open TM5b gate as expected?, or does it even remain as a dimer?)

- Figure 4 should be replaced with a schematic overview of the proposed regulation mechanism in SOS1. This is an example of how the reader is left with scattered detailed descriptions throughout the text, which could be perfectly summarized in a schematic final figure to accompany a discussion that then could include a broader perspective an comparison to regulation mechanism in other (Na⁺/H⁺) antiporters.

I would also recommend the authors to consider restructuring, shortening (some things can go into material and methods) and subdividing (rather long subsection on autoinhibition) some sections to better guide the reader and sell a more concise story. For example, are many structural elements described and their function of importance anticipated, while the required functional data is only presented later. Consequently, some sections are introduced without prove and later become repetitive. Example: Tm5b is mentioned and discussed at three different positions (i.e. around line 224, line 295 and line 327). Lastly, I would recommend toning down in general some of the strong statements done on stabilization of the dimerization, while in many cases the interactions clearly contribute to the dimerization, there is no mutagenesis data included to report the impact/contribution on dimerization for single positions.

Yet, it is clear that the CTD likely plays a major role on the stabilization of the dimer, which is different to other CTD in other Na⁺/H⁺ antiporters. Considering the increasing knowledge on the importance of the dimerization from other Na⁺/H⁺ antiporters, the authors could elaborate more on a boarder comparison to other known structures.

Other mostly minor:

- Intro and discussion lacks also a comparison to the other eukaryotic Na⁺/H⁺ antiporters like NHE9 and NHA2 instead of just NHE1
- Line 34: K=K+?
- Line 37: sodium = sodium ions
- Lines 51-52: lack of space between the organism names: *Pyrococcus abyssi* and *Methanocaldococcus jannaschii*
- Lines 70-91: can be largely shifted to material and methods
- Line 95: perhaps hint here that the vast majority of the autoinhibitory region (998-1146) is not resolved
- Line 101: amphipathic helix AH(3-4) is not shown/highlighted in any of the figures. On a similar note. Helix 3 is not labeled in Figure 1d
- e.g Line 119-121: Wouldn't it be easier for the descriptions of the loops and their interactions to use a similar nomenclature as for the amphipathic helix, so instead of IL4 use IL(TM7-8)
- Line 145-149: Hard to validate without map and model but how "bad" is the beta-CTD resolved? Is it appropriate to use AlphaFold at this point or better to exclude this region

from the final model?

- line 172-173: considering that no lipids were added during purification (correct?), it might be fair enough to mention it here and claim that the potential lipid densities observed here likely represent endogenously co-purified lipids from HEK host. A caveat here is that, no functional assays with the from HEK purified protein were performed, which would confirm that this state with the given lipids can be functional.
- Lines 186-188: based on what is the claim made that K181 and K187 are critical for dimerization of SOS1? Is it based on the inter-protomer interactions established? Still without actual mutagenesis studies you don't know if they are critical or might temporarily simply contribute to dimerization. Besides residues 181/187 need to move during transport cycle and can thus not be determinant for dimerization. Please rephrase.
- I agree with the authors that D201 most likely represent the determinant cation binding site. However, considering that the accessibility of this residue and its importance for the transport is a central point of the manuscript, the authors make a poor job in conveying this to the reader (in particular for a non-expert). Preferably, the authors could add a functional complementation assay (as this is not a very laborious experiment), showing that a mutation of this site renders the transporter inactive. The least the author should do, is to emphasize stronger (in introduction or in lines starting with 204) that this is a highly conserved motif of the CPA family, and has been extensively characterized and identified in all Na⁺/H⁺ antiporter studied so far as the central cation binding site.
- Lines 231-234: while I believe it I think the reader would benefit here from a short statement on how the authors can be sure to properly model (with correct register) a loose stretch of 10 and 18 aa within an otherwise large unresolved region. Also how sure can you be with the assignment of which element belongs to which polypeptide chain?
- Line 240: Fig 3f is cited before 3e
- Lines 250-254: tone down statement, e.g. "...and likely/presumably stabilize the intracellular domain into an inhibited state". In particular because this statement comes before your functional complementation assays
- Lines 312-315: tone down statement. The results indicate and hint but do not conclusive show your claim.
- Methods (data processing) considering the processing and the images shown the authors must have processed large amounts in Relion and not in CryoSparc as stated
- Methods: missing any info/details on the attempt for the structural determination of the truncated version
- Line 432: which lipids were added?
- Methods (complementation assay): might be good to refer here or even in the manuscript that the expression level or (if expressed at all) of the given constructs were not analyzed and thus the results nor normalized.
- Line 459: Would be nice to provide the original data of the atomic absorption spectrometer in the supplementary information, not only the final values in the main figure.
- Fig 1: Fig1a: I would call it "domain structure" instead of "linear scheme". Also display here the not fully resolved autoinhibitory tail". TM 3 and 4 are missing in description. If the authors want to state out the difference between dimerization and core domain, coloring might be more helpful, than the use of cylinders and helices
- Figure 3, the labelled boxed in panel a are written with capital, while panels are in small caps. Not all boxes and panels are labelled. (similar bug found in other figures as well)
- S4: labels or location of alpha-A,B,D partially missing in sequence alignment and structure
- S6: color code in legend wrong. Explain how structures were superimposed to better judge the RMSDs mentioned in text

Point-by-point response for

Architecture and autoinhibitory mechanism of the plasma membrane Na⁺/H⁺ antiporter SOS1 in
*Arabidopsis*

**Reviewer #1 (Remarks to the Author):**

This is a nice report about the long-awaited structure of the Na⁺/H⁺ antiporter SOS1. This molecule is the
major determinant of salt tolerance in plants.

I have enjoyed reading the article, the introduction is clear and informative, the experimental approaches
are sound and the methods section is complete. However, the conclusions derived from the structural work
do not adjust to the expectations. The work provides the basis for the self-inhibitory activity of SOS1, this
is quite a lot, but the amount of new information is limited to the identification of the boundaries of the
already reported self-inhibitory domain of the antiporter. In addition, there are several issues that there are
vaguely discussed and that deserve more attention. Among them, are the activation mechanism of SOS1,
the specificity of the channel (Na⁺ vs K⁺) and the role of the CNBD.

**Reply:** We appreciate very much the reviewer's positive comment and his/her suggestions for improving
our manuscript.

**A.-** Following the authors' arguments, one might think that the phosphorylation of the C-terminal end of the
cytosolic domain would lead to the release of the delta C loop and delta C-helix and this, in turn, would
destabilize the dimer structure allowing a reorganization of the TMD to a productive conformation (lines
324-326). This is too simplistic and far away from the experimental data showing that the truncation of the
delta C-loop and delta C-helix yields a hyperactive antiporter. There is a complex regulation of SOS1 (and
other NHX transporters) that involves different proteins in response to a variety of stimuli.

**Reply:** We appreciate reviewer's comment. We agree that the sodium transport capacity of SOS1
activated by phosphorylation via SOS2-SOS3 complex in yeast is lower than that of C-loop and C-helix
double deletion mutant as well as SOS^{Δ998} mutant. We speculate that the difference in transport activity
activated by C-terminal deletions and phosphorylation is due to the deletion mutations activate all
transporters on the cell surface, however, the phosphorylation of C-terminal tail may only reduce the
binding affinity of the inhibitory element to the transporter caused by electrostatic repulsion between
negatively charged phosphorylation site and intracellular protein surface. This repulsive force may not
completely disrupt interactions of C-loop and C-helix with intracellular domain, consequently the apparent
activity of phosphorylated SOS1 in yeast is lower than that of deletion mutations (SOS1^{ΔC-loop/C-helix} and
SOS1^{Δ998}).

The work would benefit from additional structural work (even a bioinformatic model) about the transition
from the inactive to the active conformations of the antiporter. Where are the hinge points allowing this
transition? Data processing of the discarded classes might provide information on this aspect (see below).

**Reply:** We appreciate the reviewer's comment and agree with reviewer in that the structure at active state
would help us gain more insight into activation mechanism. However, structural determination of activated
SOS1 is prohibited by high flexibility of the intracellular domains. Because we do not have the structure at
active conformation, so it is very hard to define the hinge point for conformational transition. But we
speculate that the C-loop and C-helix may serve as a switch to regulation conformational change of

intracellular domains.

We agree with reviewer to revisit the data processing. We did carry out more data processes. Please see
reply below.

**B.-** There is no mention of the important issue of the selectivity of SOS1 for Na⁺ vs K⁺. Some other plant
NHX transporters display an enhanced preference for K⁺ or display both K⁺ and Na⁺ transport activity.
The interplay between the activity of these transporters is central to the control of cell ion homeostasis
under salt stress.

**Reply:** We thanks reviewer for sharing his/her insights. We agree with the reviewer that studying the ion
selectivity of transporters is crucial. Previous research has shown that SOS1 exhibits Na⁺ specificity and
cannot transport K⁺ or Li⁺ [1]. However, as sodium ion was not identified in the electron density map, we
could not address this important issue in the manuscript. In the revised manuscript, we mentioned the ion
selectivity of the SOS1 in the introduction section. In the line 40-41, it reads “Compared with the vacuolar
Na⁺/H⁺ transporters, SOS1 is insensitive to K⁺ and plays a unique physiological role in cellular Na⁺
homeostasis^{8,9}.”.

Reference:

[1] Qiu, Quan-Sheng, et al. "Na⁺/H⁺ exchange activity in the plasma membrane of Arabidopsis." *Plant Physiology* 132.2
(2003): 1041-1052.

**C.-** The CNBD deserves more attention than a supplemental figure S4. Mutations G777E and G784D lead
to an inactive transporter (Shi et al. PNAS 2000) which is the basis of such a role? The authors could
compare the CNBLD structure to that of the CNBD of plant AKT1 (Lu et al. Nat Com. 2022).

**Reply:** We appreciate the reviewer’s comment. Previous evolutionary studies of CNBD have identified five
conserved glycine residues in the loops connecting the beta sheets in the β -roll [1]. Among the five
conserved glycine residues, G784, located in the β 1- β 2 loop, is the most conserved across diverse
eukaryotic and prokaryotic proteins [1, 2] and adopts a main-chain conformation (ϕ = 85.0; ψ = -176.5)
that is disallowed for other amino acids in the Ramachandran map [1]. We speculate that the mutation is
likely to affect the correct folding and thus impair the protein function.

G777 is located on the N-terminal of β 1, at the bottom of the β -roll domain. It forms a special glycine-
aromatic residue pair with F834 on the neighboring β 6 (Figure* 1a). The aromatic ring of F834 is located
over G777 on the neighboring sheet, the small size of which allows for F834 to position in a planar manner
directly over it. This conformation is called “mortise-tenon” joint in β -barrels containing outer-membrane
proteins and has been proved to improve the protein folding speed and stability by aligning and locking
the neighboring β -sheets in β -barrel [3][4]. G777D mutation leads to loss of function of SOS1 in previous
study and in our analysis. Moreover, we designed two new mutants (G777N and F834A) to interrupt this
interaction. It turns out both of mutants failed to improve tolerance to the high concentration of sodium,
even in the presence of the SOS2-SOS3, suggesting this interaction is important for SOS1 activity (Figure*
1b). Taken together, we speculate that the mutations of two glycine probably impair the folding and stability
of the CNBLD structure, thereby impair function of SOS1.

Figure*1. Loss-of-function analysis of *sos1-8* (G777D) and *sos1-9* (G784E) in SOS1. **a.** β -roll of the
 CNBLD. Residues G777 and G784 are shown in sticks and highlighted in yellow. F834, which is form
 “mortise-tenon” joint with G777 is highlighted in pink. **b.** Salt-tolerance test of AXT3K cells expressing the
 indicated SOS1 mutant proteins with and without the coexpression of the SOS2-SOS3 kinase complex.

In the revised manuscript, we have also included a brief discussion. In the line 144-152, it now reads “A
 previous study indicated that G777D and G784E mutations lead to an inactive transporter⁷. Based on our
 structure, G777 is located on the N-terminal of β 1 and forms a “mortise-tenon” joint with F834 on the
 neighboring β 6. Such an interaction has been proved to be important for aligning and locking the
 neighboring β -sheets and thus improve the folding speed and stability of β -sheet^{30,31}. G784 is the most
 conserved glycine among eukaryotic and prokaryotic CNBD, which adopt special dihedral angles that is
 disallowed for the other amino acid³². Thus, we speculate that the G777 and G784 residues are important
 for folding and stability of the CNBD, and thus mutations of these two residues significantly decrease
 transport activity of the SOS1.”.

We have also compared the CNBLD of SOS1 with CNBD structure of AKT1 [5], as reviewer suggested.
 They share similar overall folding, except that the two helices at the N- and C-termini (α A and α D). We
 have updated and adjusted the order of the Supplementary Figure 4 to panel Supplementary Figure 6
 according the flow in the manuscript, and included a brief description in the revised manuscript. In the line
 128-134, it reads “Superimposition of the CNBLD with the human hyperpolarization-activated cyclic
 nucleotide-gated ion channel HCN1³¹ and plant phosphorylation-activated AKT1^{32,33} using the β -roll as a
 reference, the α A and α D helices at the N- and C-termini undergoes remarkable conformational change
 (Supplementary Fig. 6a). The root mean squared deviation (RMSD) for α carbons is 2.11 Å and 2.72 Å
 between CNBLD and the CNBD of the HCN1 and AKT1, respectively.”.

We have attached revised Supplementary Figure 6 here for your convenience.

Reference:

[1] Kannan, Natarajan, et al. "Evolution of allostery in the cyclic nucleotide binding module." *Genome biology* 8.12 (2007):
1-13.

[2] Mohanty, Smita, et al. "Structural and evolutionary divergence of cyclic nucleotide binding domains in eukaryotic
pathogens: implications for drug design." *Biochimica et Biophysica Acta (BBA)-Proteins and Proteomics* 1854.10 (2015):
1575-1585.

[3] Leyton, Denisse L., et al. "A mortise–tenon joint in the transmembrane domain modulates autotransporter assembly into
bacterial outer membranes." *Nature communications* 5.1 (2014): 1-11.

[4] Michalik, Marcin, et al. "An evolutionarily conserved glycine-tyrosine motif forms a folding core in outer membrane
proteins." *PLoS One* 12.8 (2017): e0182016.

[5] Lu, Y., Yu, M., Jia, Y. *et al.* Structural basis for the activity regulation of a potassium channel AKT1 from Arabidopsis. *Nat*
*Commun* 13, 5682 (2022).

D.- There is care in the preparation of the figures but it is difficult to follow the nicely described structural
features of the macromolecule.

**Reply:** We appreciate your comments. We have carefully checked and adjusted the order of figures and
figure calls appropriately.

The manuscript might be improved with a revision concerning the above-mentioned aspects and the
following points.

**1.-** The purification protocol seems to yield a large quantity of homogeneous protein. Please provide details
about the amount of protein in mg per volume of culture. Line 354, I understand that the GFP and the twin-
Strep tag were introduced “following” (not “followed”) the PreScission Protease (PPase) recognition site.

**Reply:** We thank the reviewer for raising this point. The protein yield is approximately 0.2 mg/L of HEK293
culture. In the revised manuscript, we have provided the yield and also corrected the typo your pointed
out. In line 440, it reads “The yield of protein was approximately 0.2 mg/L of HEK293F cell culture.”.

**2-** The data processing protocol is biased towards the description of the procedures to obtain the self-
inhibited structure of SOS1. However, I wonder if the authors have explored enough the large
conformational landscape of the observed particles. In particular, they discard around 80% of the particles
(they used 151,649 particles out of 543439) that could contain valuable information about conformational
changes that SOS1 might undergo to develop its function. Indeed, I have got the impression from the
central panel of figure S2 showing four classes, that the discarded classes display different conformation
at the cytosolic moiety of the molecule. The authors should process such particles and analyze the
resulting structures or, alternatively, provide a convincing argument to discard them. In addition, would it
be possible to mask the transmembrane domain of the truncated protein version and produce a structural
model?

**Reply:** We thanks the reviewers for the constructive comments. We re-processed the remaining classes,
especially the class I and class IV. We did not further focus on class III, because it contains 21,309 particles
and 3D refinement only results in a 7.5 Å map. The class I and class IV contain 290,003 and 80,155
particles, respectively. We carried out further 3D classification and refinement for these two particle-sets.
The final resolutions of cryo-EM maps of class I and class IV were 3.5 Å and 3.4 Å, respectively. The
mainchain and sidechains are clearly resolved, which allow us to build the models. We have also
reprocessed the data from the truncated sample and applied a TMD mask as reviewer suggested.
Unfortunately, we were unable to obtain a reasonable map, likely due to poorly aligned particles in the
absence of intracellular domains.

Next, we compared these three structures from class I, II and IV. and found that the SOS1^{CI} structure
assumes a nearly identical conformation to structure mentioned in the original manuscript (SOS1-ClassII,
2.8-Å), supported by R.M.S.D. between these two structures are 0.31 Å (0.27 Å for TMD and 0.32 Å for
intracellular domains). However, the structural comparison between SOS1^{CII} and SOS1^{CIV} structures yields
an RMSD of 1.46 Å, 2.00 Å and 0.70 Å for overall structure, TMD and the intracellular domain, respectively,
indicating that the TMD undergoes conformational change between SOS1^{CII} and SOS1^{CIV} structures.
Taking a closer look at the SOS1, we found that six strip shaped densities are buried within the cavity
between two dimerization domains. Consequently, the cavity in SOS1^{CIV} is larger than that in SOS1-ClassII
and the extracellular part of SOS1^{CIV} TMD is more expanded. We speculate that the lipid molecule may
exert regulatory effect on SOS1 by influence dimerization interface. We have also included related
discussion in the revised manuscript.

In the line 83-94 (main text), it reads “Four classes (class I-IV) are generated from 3D classification

without alignment (Supplementary Fig. 2a). We determined three cryo-EM maps from different classes
(class I, II, and IV) at 3.5 Å, 2.8 Å and 3.4 Å, respectively, according to the gold-standard Fourier shell
correlation (GSFSC) criterion (Supplementary Fig. 2b-j). The high-quality cryo-EM maps allow us to reliably
build the SOS1 models (Supplementary Fig. 3). Next, we compared the SOS1-ClassI and SOS1-ClassIV
with the 2.8-Å resolution SOS1-ClassII. We found that the structures of SOS1-ClassI and SOS1-ClassII are
nearly identical, supported by RMSD of ~0.3 Å for 1856 C α atoms. However, superimposition of SOS1-
ClassIV structure to the SOS1-ClassII structure gives rise to RMSDs of 1.5 Å for overall structure, indicating
that two different SOS1 structures have been resolved. We firstly focused on the 2.8 Å-resolution SOS1-
ClassII to display structural features of SOS1 and name SOS1-ClassII as SOS1 in short.”.

In the line 203-219 (main text), it reads “Structural comparison indicates that the dimerized SOS1-ClassIV
harbors a distinct conformation to the 2.8-Å resolution SOS1 that the TMD undergoes significant
conformational changes and the intracellular domains are nearly identical (Fig3. a-d), supported by the
RMSD values between these two structures are 2.0 Å and 0.7 Å for TMD and CTD, respectively. Taking a
close look, the extracellular ends of TM helices from two protomers expand in opposing directions, except
that the TM8 and TM9 remain unaltered (Fig. 3b), while the cytoplasmic ends of TM helices are
superimposable (Fig. 3c). Therefore, we termed the SOS1-ClassIV structure as SOS1^{expand}. We next
compared structures of one protomer in these two conformations and found that they adopt a similar
conformation, but have some subtle structural discrepancies at TM8 and TM9 helices (Fig. 3e), with RMSD
of 0.6 Å. Within the hydrophobic central cavity between dimerization domains, we found that the lipid
molecules differently interact with the SOS1 in these two conformations. In both structures, four lipid
molecules are determined at similar positions proximal to the 2-fold symmetrical axis (Fig. 3f, g). However,
in the SOS1^{expand} structure, an extra lipid is inserted into space between TM1 and TM3' helices at the
dimerization interface. Meanwhile, the side chain of Y270 on TM8' is extruded in the SOS1^{expand} structure
to accommodate this lipid molecule (Fig. 3h).”.

The data processing method has also been updated (line 482-490), it reads “These particles were
subsequently submitted to another round of skip-align 3D classification in RELION, resulting in four
reconstructions at overall resolutions of 5.5 Å, 3.1 Å, 7.5 Å, and 4.0 Å, respectively. Particles from these
classes, namely, 290,003 particles (58.6%) from class I, 151,646 particles (21.5%) from class II, and
80,155 particles (12.4%) from class IV, were imported into cryoSPARC for further processing.
Heterogeneous refinements were performed on the classes individually to improve map quality. The final
maps were generated by non-uniform refinement in cryoSPARC, which generated a class I map at 3.5 Å,
a class II map at 2.8 Å, and a class IV map at 3.4 Å, all based on the GSFSC criterion. A diagram of data
processing is presented in Supplementary Figure 2.”.

The supplementary figure 2 about data processing has been adjusted and attached here for your
convenience.

Supplementary Figure 2. Single-particle cryo-EM analysis of SOS1.

Structural analysis of the SOS1^{expand} structure has been shown in Figure 3 in the revised version. We
attached here for your convenience.

Figure 3. Lipids induced rearrangement of dimerization interface. **a**, Overall structure of the SOS1^{expand}.
 Two monomers are colored in green and yellow, respectively. Lipid molecules are shown as red spheres.
 **b-d**, Structural comparisons based on the whole dimeric structure between SOS1^{expand} and SOS1,
 illustrated using three slice views (b–d) as indicated by labeled panes; namely, TMD from extracellular side
 (b), TMD from intracellular side (c), and α -CTD from extracellular side (d). Black arrows indicated the helix
 movements of SOS1^{expand} compared to the SOS1. TM8 and TM9 remain almost static and are labeled in
 (b). **e**, Structural comparison between the monomers of SOS1^{expand} and SOS1, using monomer itself as
 the alignment reference. TM8 and TM9 show discernible conformational changes and are highlighted.
 **f**, Four lipid molecules embedded between the dimer interface of SOS1. **g-h**, Six lipid molecules embedded
 between the dimer interface of SOS1^{expand}. The extra lipid in a SOS1^{expand} monomer is highlighted in (g),
 and its interactions with adjacent regions are shown in (h). Y270 is shown as sticks and displayed
 conformational change between SOS1^{expand} and SOS1.

We also deposited PDB and cryo-EM map and the accession numbers have been updated in the Data
 availability section. It reads “The cryo-EM density map of the SOS1 and SOS1^{expand} have been deposited
 in the Electron Microscopy Data Bank (EMDB) under the accession code EMD-33592 and EMD-35085,
 respectively. The coordinate for the SOS1 and SOS1^{expand} have been deposited in the Protein Data Bank
 (PDB) under the PDB ID 7Y3E and 8HYA, respectively.”.

**3.-** The authors employed a construct including a GFP protein at the C-terminal end of the truncated protein.

I wonder if this might be the origin of the inhomogeneity observed for this preparation as, in contrast to the
wild-type protein, the GFP might be too close to the beta-CTD.

**Reply:** We thank the reviewer for pointing this out. However, we don't think the GFP may affect the
structure of the truncated SOS1. First, we predicted an AlphaFold model for the truncated construct. It
turns out that 38 residues between GFP and beta CTD represents a disordered loop. Second, we included
a PPase cleavage site before the GFP-StrepII tag in the truncated SOS1^{Δ998} construct. During sample
preparation, the GFP-StrepII tag can be removed by incubation with PPase, indicating there is enough
space to allow the PPase access. In fact, the sample from SEC and grids preparation does not contain
GFP tag any more. At last, a previous report indicated that the SOS1^{Δ990} with an eGFP tag at the C-terminal
was functional active and showed obvious Na⁺/H⁺ transport activity when reconstituted into liposomes [1],
suggesting the GFP tag does not hamper transporter activity.

Figure* 2. Alphafold prediction of the construct for SOS1^{Δ998} expression and purification. The protein
segment between the last residues 1972 of β-CTD to Q998 is colored in blue. PPase cleavage site located
in front of the GFP-StrepII tag (12 residues) is colored in red.

Reference:

[1] Ullah, Asad, Debajyoti Dutta, and Larry Fliegel. "Expression and characterization of the SOS1 Arabidopsis salt tolerance
protein." *Molecular and cellular biochemistry* 415.1 (2016): 133-143.

**4.-** Lines 221 to 224. Is SOS1 trapped in an Inward facing conformation or in an intermediate state between
inward and outward-facing conformation? Please clarify.

**Reply:** We appreciate the reviewer's comment. The SOS1 is trapped in an intermediate state, not inward-
facing or outward-facing conformation. In the manuscript, we described that the SOS1 is stabilized at
inward-facing occluded state, because it has an intracellular water-access cavity like the NHEs structures
at inward-facing conformation, while the ion-binding residue (D201) is completely buried and inaccessible
from intracellular side due to upwards movement of the TM5b helix. To avoid confusion, we called this
state as occluded state in the revised manuscript.

**5.-** Supplementary Fig. 4. Please color the sticks by element type so the reader can visualize the

differences between SOS1 CNBD and a canonical one.

**Reply:** We appreciate the reviewer's comment. The sticks have been colored according to element type.
We attached the old and revised figure here for your convenience.

Figure* 3. Comparison of the old and revised Supplementary Fig. 6c.

**6.-** Lines 288 to 293. The authors speculate about the influence on the phosphorylation-mediated activation
mechanism of a negatively charged patch around the C-loop. They should use the salt tolerance test of
AXT3K cells to prove this statement.

**Reply:** We appreciated this comment. So far, a report suggested that the S1138 is phosphorylated by
SOS2 and the S1136 is essential for substrate recognition by the protein kinase [1]. Replacing either site
with alanine (S1136A or S1138A) or removing both sites but retaining C-loop and C-helix (SOS1 Δ 1047 or
SOS1 Δ 1072) did not activate SOS1, but disrupted ability of SOS2T168D/ Δ 308 (a constitutively active form
of SOS2) to activate the SOS1 [1]. Therefore, we substituted these two sites to aspartate to mimic negative
charged phosphorylated serine. However, the SOS^{S1136D} and SOS1^{S1138D} are still inhibited with basal
activity (Figure* 4). These experimental data appears to not support our hypothesis. However, there are
probably other phosphorylation sites remain to be identified (See reply on page 20-21). Therefore, we have
toned down the description about the charge electrostatic potential surface in the discussion section of the
revised manuscript. It reads "Considering the critical roles of C-loop/C-helix and phosphorylation of C
terminal tail in activation of SOS1, we further hypothesis that the phosphorylation of the C-terminal tail
would introduce additional negative charge(s) that would repel negatively charged intracellular domains,
reducing binding affinity of the C-loop and C-helix to the intracellular domain and destabilizing dimerization
of the intracellular domain, thereby activating the SOS1. However, considering the phosphorylation sites
have not been clearly identified, further investigations are required to better understand how
phosphorylation activates the SOS1."

Figure* 4. Salt tolerance test of AXT3K cells expressing solely the wild-type SOS1 and SOS1 mutants as
indicated.

Reference:

[1] Quintero, Francisco J., et al. "Activation of the plasma membrane Na/H antiporter Salt-Overly-Sensitive 1 (SOS1) by
phosphorylation of an auto-inhibitory C-terminal domain." Proceedings of the National Academy of Sciences 108.6 (2011):
2611-2616.

Other minor comments

7.- Line 416. "151,646 particles" not "151,646k particles"

**Reply:** We thank the reviewer for pointing out this and have made a correction.

8.- Fig. 1a (line 110) is called later than Fig. 1b. Supplementary fig 2 goes to supplementary fig 5d

**Reply:** We thank the reviewer for pointing this out. We have fixed these two issues you raised and carefully
checked other figure calls as well.

9.- Line 215. "NHE-CHP1" not "NHE-CHP11F"

**Reply:** We thank the reviewer for pointing out this and have corrected this in the revised manuscript.

10.- Please homogenize cation calls. Either Na⁺ or sodium, K⁺ or potassium

**Reply:** We thank the reviewer for pointing out this and we have uniformed cation calls as K⁺ or Na⁺ in the
revised manuscript.

Reviewer #2 (Remarks to the Author):

The Na/H exchanger SOS1 is a critical component of the SOS pathway for salt tolerance that is ubiquitous
in plants. Although the pore domain of SOS1 can be reliably modeled using as template known structures
of phylogenetically similar proteins (e.g. animal NHEs) or by the AlphaFold software, the SOS1 protein is
likely to possess regulatory roles that go beyond the mere ion exchange for sodium extrusion, and which
may involve the unique C-terminal of this protein. Therefore, obtaining the refined cryo-EM structure of
SOS1 could have been a major achievement in plant physiology. However, this report is insufficient to
claim success, as it suffers from limitations that curtail the novelty and the potential impact. Below, I will try
to explain the limitations of this work.

**Reply:** We thank reviewer's suggestions for improving our manuscript.

1. The main contribution of this work is a structural description of the mechanism underlying the
autoinhibition of SOS1 by two segments of the cytosolic domain that repress the activity of SOS1 to keep
the protein in a resting state. However, no biochemical analyses are performed to shed light on the
differences between repressed, derepressed and activated proteins.

**Reply:** We appreciate reviewer's comment. In this study, we employed single particle cryo-EM method to
determine SOS1 at resting state and in attempt to determine its active state by delete the C-terminal

autoinhibitory region (SOS1^{Δ998}). We found that C-loop and C-helix plays important roles in stabilizing
 SOS1 at its resting state. The structural study of SOS1^{Δ998} mutant revealed a highly dynamic intracellular
 domains that is completely blurred on the 2D average image and 3D classes, suggesting that the
 intracellular domains may undergo significant conformational change to activate SOS1. We carried out
 extensive yeast complementary analysis, a classic and widely used experimental approach to study SOS1
 function, which we believe are sufficient to support our speculations.

**2. The SOS1 protein is 1146 residues long. The cryo-EM structure of SOS1 is only partial, comprising the**
 **protein stretches 32–972, 1007–1017 and 1030–1048. The structure of the phosphorylation domain, the**
 **interplay with the auto-inhibitory domain, and a functional description of how SOS1 becomes active are all**
 **missing.**

**Reply:** We appreciate reviewer’s comment. The unresolved regions are supposed to be very flexible
 according to the secondary structure prediction and predicted model by AlphaFold2 (Figure* 5a, b). It is
 very common that the region of highly motility cannot be resolved in the cryo-EM map.

We agree with reviewer in that it is important to discuss how phosphorylation site activate SOS1. However,
 the phosphorylation site(s) have not been clearly identified to date, so we could not precisely discuss how
 the potential phosphorylation site activates SOS1. However, we did propose a hypothesis that the
 phosphorylation of C-terminal tail of SOS1 introduces negative charge(s), which electrostatically repulses
 the negative charged electrostatic surface, thereby interfering with binding of C-loop and C-helix to the
 intracellular domains and activating SOS1. But this hypothesis is needed to be further validated after
 phosphorylation sites are identified (See reply on page 20-21). Please see discussion in the revised
 manuscript. It reads “Interestingly, we found that the surface electrostatic potential of the intracellular
 domains is negatively charged (Supplementary Fig. 12). Considering the critical roles of C-loop/C-helix
 and phosphorylation of C terminal tail in activation of SOS1, we further hypothesis that the phosphorylation
 of the C-terminal tail would introduce additional negative charge(s) that would repel negatively charged
 intracellular domains, reducing binding affinity of the C-loop and C-helix to the intracellular domain and
 destabilizing dimerization of the intracellular domain, thereby activating the SOS1. However, considering
 the phosphorylation sites have not been clearly identified, further investigations are required to better
 understand how phosphorylation activates the SOS1.”.

 **Figure* 5. The flexible C-terminal region of SOS1. a, Secondary structure prediction of C-terminal region**
 **(residues 990–1146) of SOS1 made by using JPred secondary structural prediction software. b, 3D**
 **structure prediction of C-terminal structure (residues 998–1146) by AlphaFold. C-loop and C-helix are**
 **highlighted in the two figures.**

**3. The CNBLD of SOS1 has primary sequence homology with bona fide CNBD of the hHCN1 channel.**

Authors show that critical residues for cAMP binding in the CNBD pocket of hHCN1 are not really
conserved in SOS1, and that some of these important residues are substituted by larger ones imposing a
steric hindrance. Authors conclude that SOS1 cannot possibly bind cAMP, but they do not make any
attempt, not even present a speculative hypothesis, to provide a clue on what the agonist is, if any. The
original report of the SOS1 gene sequence described mutations in the putative CNBLD that behaved as
lost-of-function mutants (Shi et al., 2000, PNAS). Authors could try to provide a structural basis for these
findings.

**Reply:** We appreciate the reviewer's comment. So far, no biochemical evidence is reported that the SOS1
is directly regulated by cyclic nucleotide and our structure clearly elucidates that the binding pocket of
cyclic nucleotide in SOS1 is not functional. Some substitutions directly destroy critical interactions for
cAMP recognition and binding or make the pocket too small to accommodate cAMP. We entirely agree
that screening of potential ligand of CNBLD is important. However, these experiments are very challenging
and time consuming and we believe are simply beyond the present scope of this work. Moreover, we argue
that the present work represents a sufficiently important advance as to stand on its own.

In 2000 PNAS paper, the author found that G777D and G784E mutants abolished transport activity of
SOS1. The reviewer 1 also has a same question about structural basis for these findings. Please check
our reply on page 2-4.

**4.** A large proportion of this report is devoted to give a structural framework to the mechanism of auto-
inhibition that keeps SOS1 in a resting state. This is the only part in which authors took the effort to
corroborate hypotheses by conducting functional studies. There are however several limitations here too.
Firstly, the structure of the auto-inhibitory domain is based not on original cryo-EM data but on the docking
of the AlphaFold2 prediction for this domain into the SOS1 structure. Secondly, the C-loop and C-helix
domains conforming the auto-inhibitory domain reported here largely correspond to the bipartite auto-
inhibitory domain previously identified by Quintero et al., 2011, PNAS, which detracts from novelty.
Dissecting the two halves of the bipartite auto-inhibitory domain into what is called here C-loop and C-helix
was also reported there. The real advance in this report is the identification of W1013 and F1042 as the
critical residues in C-loop and C-helix, respectively.

**Reply:** We appreciate the reviewer's comment. We did utilize AlphaFold2 to facilitate our model building,
but this should not be a reason to undervalue our work. Because our cryo-EM map provides important and
essential information on where to properly fit the β -CTD into and how it interacts with other intracellular
domains. we have also included model-map fitting figure in the revised supplementary figure 3 and
attached here for your convenience. The C-loop and C-helix have been proposed to constitute the
autoinhibitory domain of SOS1 in 2011 PNAS paper. Our work nicely confirmed their speculations. But
most importantly, we provided molecular basis of how these two conserved motifs interact with and
stabilize the dimerization of intracellular domains, thereby blocking transport activity of the SOS1. We
believe these mechanistic insights are fundamentally important and long-awaited.

Figure* 6. Electron density of β -CTD. **a**, Local resolution of the cryo-EM density map of SOS1. The density
 of β -CTD is highlighted in a black box. **b**, Electron density map after density modification is shown in
 transparent surface, and the refined model is shown in cartoon with some large-sidechain residues shown
 in sticks and labeled.

**5. Growth and sodium content data of yeast expressing mutated/truncated SOS1 proteins attributed to**
 **differential transport activity should be completed, at the very least, with the inspection of protein**
 **abundance, and preferably with direct measurements of transport activity. I could not find whether the**
 **mutant proteins named Δ C-loop and Δ C-helix corresponded to internal deletions or protein truncations;**
 **nothing in methods on how these mutations were constructed.**

**Reply:** We agree with the reviewer in that it would be nice to directly measure the purified protein samples
 by in vitro transport assays. However, it proved to be highly challenging due to the low yield of the protein,
 whereas liposome reconstitution would require large amounts of the protein samples. Regarding concerns
 about the expression level of mutated/truncated SOS1 expression, we perform the expression analysis of
 SOS1 and its mutants in yeast strain AXT3K using Western blotting against the C-terminal 6 \times His tag.

For all proteins tested, the expression level is comparable, suggesting the variations in transport activity
 were not caused by fluctuations in expression levels, but rather by mutations that had an impact on the
 protein's functionality. These new data are now included in the Supplementary Figure 10. We have
 attached here for your convenience. The method has also been updated (line 559-573).

The Δ C-loop and Δ C-helix mutations are internal deletions. We have clarified this in the revised manuscript.
 In the line 298, it now reads "To understand the functional roles of the C-loop and C-helix determined in
 the structure, we constructed internal deletions of C-loop (SOS1 Δ C-loop), C-helix (SOS1 Δ C-helix) or both
 (SOS1 Δ C-loop/C-helix) and performed functional complementation assays using the salt-sensitive yeast mutant
 strain AXT3K...".

We have included the strategy how we make these mutations in the methods section. In the line 531-537,
 it reads "Full-length *Arabidopsis thaliana* SOS1 was cloned into the plasmid pYPGE15 with a C-terminal
 6 \times His tag. Nine different constructs of SOS1 were made, containing five mutations (D201A, W1013A,
 F1042A, ⁷⁰²EEK⁷⁰⁴/AAA, ⁴⁸⁴DDEE⁴⁸⁷/AAAA), three internal deletions (SOS1 Δ C-loop, SOS1 Δ C-helix and
 SOS1 Δ C-loop/C-helix) and a truncation construction (SOS1 Δ 998). All the mutations were generated by overlap
 extension PCR and sequenced prior to use. The forward primers and the reverse primers for the
 modifications are summarized in Supplementary Table 2."

**Supplementary Figure 10. Protein expression analysis SOS1 and derived mutants in yeasts.**

Western blot (a) and quantification (b) of wild-type and mutated SOS1 in transgenic yeast against the C-
 terminal His tag. Antibody against β -tubulin was used for the loading control. The levels of mutants were
 normalized to β -tubulin and presented as the relative expression levels to the wildtype SOS1. Data are
 obtained from densitometric scans by ImageJ and presented as means \pm standard deviation of three
 independent experiments.

**6.** The purpose of the lengthy explanation of interactions between HC7-HC8 and TMs is unclear because,
 at the end, it only leads to the suggestion that these interactions are crucial to regulation SOS1 activity.
 However, no solid rationale or experimental evidence is given for this proposal.

**Reply:** We thank the reviewer's comments. We agree with reviewer that to tone down our speculation
 about these interactions. We have moved these structural details into the supplementary figure legend in
 case some audiences are interested.

**7.** No mutagenesis and functional analyses have been performed to substantiate some of the structure-
 based speculations on SOS1 activity (e.g., residues presumably interacting with lipid molecules, to which
 authors refer to as 'essential for the stability of the SOS1 dimer', or the role of D201 in substrate
 coordination).

**Reply:** We are grateful to the reviewer for providing valuable comments. It is a challenge to prove the
 absence of lipids within the cavity, even with the introduction of mutations. Moreover, lipid molecules have
 been observed in various other NHE structures, including NHE1-CHP1, NHE3-CHP1, NHE9, and NHA2.
 The binding of lipids at the dimerization interface of NHE9 has been shown to enhance the stability of the
 dimeric NHE9, as determined by the thermal shift assay [1]. As another reviewer suggested, we have
 tuned down related statement, which now reads as "Considering no additional lipids were supplied during
 sample preparation, we speculate that these lipid molecules observed within the cavity may be derived
 from HEK293 host and are probably involved in stabilizing the dimeric SOS1." in line 181-183.

The complementation assay using salt-sensitive yeast shows that the D201A mutation abolishes transport
activity of SOS1, even in the presence of the SOS2-SOS3 complex, supporting that the D201^{TM6} may be
important for the function of the SOS1. We have added the description of the yeast complementary
analysis in the results section. The results of the plaque experiment were merged into Supplementary
Figure 9.

Figure* 7. SOS1^{D201A} mutant was transformed in strain AXT3K, with and without the coexpression of p414-
SOS2-SOS3, and compared with wide-type SOS1 in AP medium with the indicated concentrations of NaCl.

Reference:

[1] Winkelmann, Iven, et al. "Structure and elevator mechanism of the mammalian sodium/proton exchanger NHE9." *The*
*EMBO journal* 39.24 (2020): 4541-4559.

**8. Description of the residue interactions that authors consider important are sometimes hard to follow**
**because figures do not represent with sufficient clarity (or not shown at all) what the text says (e.g. AH**
**helices, see below).**

**Reply:** We greatly thank the reviewer to raise some specific points to improve our manuscript and we have
made corrections according to your suggestions below.

**Other minor issues are:**

**9. No description of how SOS1 was purified from HEK cells.**

**Reply:** Thanks for your comments. We have provided details of purification of SOS1 in the method section,
including SOS1 expression, cell disruption, cell membrane enrichment, extraction SOS1 from membrane,
streptactin affinity purification, enzyme digestion and size exclusion chromatography. In the revised line
416-443, it reads "Recombinant proteins were expressed in HEK293F cells using the Bac-to-Bac
baculovirus expression system (Invitrogen, USA). Specifically, P2 viruses generated in Sf9 insect cells
were used to infect HEK293F at a density of 2.2–2.5×10⁶ cells ml⁻¹ with 1% (v/v) fetal bovine serum (FBS).
Transfected HEK293F cells were cultured in a shaker with 5% CO₂ at 37°C. After 12 h, 10 mM sodium
butyrate was added to the medium and the cells were incubated for another 48 h before harvesting.
Harvested cells were snap-frozen in liquid nitrogen and resuspended in a lysis buffer (20 mM HEPES pH
7.5, 150 mM NaCl, 5 mM β-mercaptoethanol (β-ME), 2 μM leupeptin, 0.8 μM pepstatin, and 2 μM aprotinin),
followed by lysing with a Dounce homogenizer. The membrane fraction was isolated by ultracentrifugation
at 45,000 g for 40 min and immediately resuspended into a solubilization buffer (20 mM HEPES pH 7.5,
150 mM NaCl, 1% (w/v) digitonin, 5 mM β-ME, 5 mM ATP, 5 mM MgCl₂, 2 μM leupeptin, 0.8 μM pepstatin,
and 2 μM aprotinin) for 3 h with agitation. Insoluble debris was removed by ultracentrifugation at 45,000 g
for 40 min, and the supernatant was subjected to affinity chromatography column with Streptactin Beads
(Smart-Lifesciences, China) at a flow rate of 1 mL/min. The beads were washed with 10 column volumes

of washing buffer (20 mM HEPES pH 7.5, 150 mM NaCl, 5 mM β -ME, 0.03% (w/v) glycol-diosgenin (GDN),
2 μ M leupeptin, 0.8 μ M pepstatin, and 2 μ M aprotinin) to remove nonspecific bound protein and then the
SOS1 protein was eluted with the washing buffer supplemented with 5 mM desthiobiotin. The eluate was
incubated together with PPase for 3 h to improve the sample homogeneity, concentrated with 100-kDa
cut-off Amicon Ultra centrifugal filter (Merck Millipore, Germany) to 1 mL and further purified by size
exclusion chromatography (SEC) using a Superose 6 Increase 10/300 GL column (GE Healthcare, USA)
with a running buffer containing 20 mM HEPES pH 7.5, 150 mM NaCl, 5 mM β -ME and 0.007% (w/v) glyco-
diosgenin (GDN). The peak fractions were collected, concentrated to \sim 6 mg/mL for cryo-EM grid
preparation. The final yield of protein was approximately 0.25 mg/L of HEK293F cell culture. All processes
for protein purification were carried out at 4°C. The purified protein was also analyzed using High
Performance Liquid Chromatography (HPLC) and Coomassie-stained SDS-PAGE.”.

**10. Ln 21. Change text to 'We report the PARTIAL structure of SOS1' to give a realistic view of the progress.**

**Reply:** We thank the reviewer for pointing this out. Although some disordered regions were not visualized
in our structure, the protein we expressed for structural determination is full-length and wild type. The
unresolved regions are a result of their conformational heterogeneity. We prefer not to make changes
because labeling it as a "partial structure" may mislead the audience into thinking we only determined a
small piece of the SOS1 structure. Furthermore, we have already provided a clear and honest description
of which regions are missing in our manuscript. In the line 96, it reads “*Arabidopsis thaliana* SOS1 consists
of 1146 residues. However, our SOS1 model is only composed of residues 32–972, 1007–1017 and 1030–
1048. The missing residues occur at the N- or C-terminus, probably due to high flexibility²⁵.”.

**11. Ln 65-66. First report on CBL10 mediated regulation of SOS1 is Quan et al. (2007), Plant Cell 19:**
**1415-1431.**

**Reply:** Thank you for your suggestions. This reference has been added in the revised manuscript.

**12. Ln 84-85. Cannot understand how authors concluded from 'mushroom-like' 2D pictures that SOS1**
**consisted of a transmembrane domain (TMD) and a large cytoplasmic domain (CTD).**

**Reply:** The 2D class averages revealed the prominent detergent feature around the SOS1 protein, which
is visible as a band in the sideview and as a disc in the tilted top-down view (marked with white arrow in
the figure below). Within the detergent micelle, the rod-shaped feature is derived from transmembrane
helices (yellow dot). The structures protrude from the detergent micelles could be the cytoplasmic domain
(red arrow), since the SOS1 is known to possess a large C-terminal cytoplasmic domain for transport
regulation.

**Figure* 8. Reference-free 2D class average of full-length wild-type SOS1 protein.**

**13. Ln 101-102. Amphipathic helix AH3-4 is not shown neither in Fig 1d nor Suppl 5d.**

**Reply:** We thank your kind reminders and we have revised the figure calls in this sentence. The revised
sentence (line 104-107) now reads: “Intriguingly, in addition to the short amphipathic helix between TM3
and TM4 (AH³⁻⁴) on the intracellular side^{21,26} (Supplementary Fig. 5a-b), SOS1 has an amphipathic helix
on the extracellular side between TM2 and TM3 (AH²⁻³) (Fig. 1d).”.

**14. Ln 117-118. HC7 is not shown in Suppl Fig 3a**

**Reply:** Thank you very much for pointing out this issue. We have added HC7 label in the Suppl Fig 5a and
included Fig. 1e to show the position of HC7 and HC8. It now reads: “However, the other arm formed by
HC7 and HC8 is positioned at opposite sides of the HC1-HC2 helices and parallel to the membrane plane,
participating in interactions with intracellular loops between TM helices (Supplementary Fig. 5a, c-d).”.

**15. Ln 131-132. Text refers to HC7-HC8 but Suppl. Fig 3e shows HC6**

**Reply:** Thank you very much for the pointing out this typo. We have corrected this sentence. In the revised
line 124-126, it now reads: “The surface of the β -roll forms extensive electrostatic interactions with the HC6
of the α -CTD and there is hydrophobic interaction between α C of CNBLD and β -CTD (Supplementary Fig.
5f-g).”.

**16. Ln 160-161. Revise broken sentence.**

**Reply:** Thank you for pointing this out. We have adjusted the sentence in the revised manuscript. The
revised sentence reads: “At the site proximal to the intracellular surface, the N-terminus of TM10 is located
near the 2-fold symmetry axis, and forms hydrophobic interactions with each other via residues F314,
M317 and I321.”.

**17. Ln 208-219 and Fig 3B. To what protein the grey background structure corresponds to, NHE1 or SOS1?
Whereas the displacement of TM5b of SOS1 relative to that of NHE1 is evident, residues D201 of SOS1
and D267 of NHE1 occupy the same position in the inner cavity.**

**Reply:** Thank you for raising this point. The gray background represents the dimerization domain of SOS1,
and we have clarified this in the revised figure legend. It reads “The core domain is shown in cartoon, and
the dimerization domain of SOS1 is shown in grey surface.”.

The structural comparison of core domains of SOS1 and inward-facing NHE1 clearly elucidates the
displacement of TM5b (Fig. 4b, c). Moreover, the D201 of SOS1 and D267 of NHE1 occupy same position,
suggesting that the transition from the occluded state (SOS1) to the inward open state is produced by the
TM5b shift from the alpha helix to the 3_{10} helix, and does not require the displacement of the core domain
relative to the dimerization domain.

**18. Ln 289-291 and Suppl Fig 7. The phosphorylation domain is missing in the SOS1 structure. Hence,
the speculation regarding the effect of the phosphate charge on the interaction between the C-loop and
cytoplasmic domain has no basis.**

**Reply:** We appreciated reviewer’s comment. Our study demonstrated that C-loop and C-helix are directly
involved in inhibiting SOS1. Decrease their affinity to the intracellular domains (SOS1^{W1013A} and SOS1^{F1042A})
are able to significantly improve transport activity of SOS1. We all know that phosphorylation could activate
SOS1. So we believe that it is reasonable to speculate that the activation of SOS1 by phosphorylation is

probably due to phosphorylation affects the binding affinity of C-loop and C-helix to the intracellular
domains.

**19.** In Suppl Fig 7, the C-loop and C-helix domains are not labeled. The blue-color cartoon structure is
hard to see against the electrostatic potential surface representation.

**Reply:** Thank you for pointing this out. We have revised the figure and attached here for your convenience.

**20.** Ln 340-341. Revise sentence, NHE1-CHP1-OF named twice.

**Reply:** We appreciate reviewer for pointing this out and we have adjusted the sentence. The revised
sentence now reads: "Interestingly, the structure of the core domain of SOS1, including TM5b, resembles
that of the NHE1-CHP1^{OF} complex rather than that of the NHE1-CHP1^{IF} complex."

**Reviewer #3 (Remarks to the Author):**

In the manuscript of Wang et al. the authors describe the structure of salt-overly-sensitive 1 (SOS1) protein.
This Na⁺/H⁺ antiporter belongs to the superfamily of cation proton antiporters (CPA) and is essential in
higher plants to cope with salt stress.

While several structures of Na⁺/H⁺ antiporter have been solved over the past decades describing their
overall architecture and mode of transport, this study provides new and interesting insights into unique
regulation mechanisms.

The manuscript is largely composed of a detailed description of the structure and single elements but lacks
a better final proposition of a model and a broader comparison and perspective to the superfamily and the
transporter field. The manuscript could be substantially strengthened if the authors could comment
on/incorporate the following:

**Reply:** We appreciate very much the reviewer's positive comment and his/her suggestions for improving
our manuscript.

- An introduction and discussion on the composition of the cytoplasmic domain (CTD). Which Na⁺/H⁺
antiporter have such a CTD, including for example a CNBLD? The authors argue from the given structure
why the CNBLD presumably cannot bind a cyclic-nucleotide, but has this been confirmed biochemically or
structurally?

**Reply:** We thank the reviewer for the valuable comments. So far, none of determined structures of
 eukaryotic Na⁺/H⁺ antiporters contain such a large CTD or CNBLD. The CTD in most of NHEs are
 disordered, probably with exception of NHE10 (SLC9C1) and NHE11 (SLC9C2). The NHE10 and NHE11
 also contain a CNBD domain. A previous study demonstrated that the NHE10 from *Strongylocentrotus*
 *purpuratus* (spNHE10) is gated by voltage sensor and cyclic-nucleotide binding [1]. However, the
 structures have not been determined.

The structural study of CNBD from some of the well characterized CNBD-containing families reveals a
 conserved mode of cAMP recognition and regulation. The ribose and cyclic phosphate interact exclusively
 with the β-roll, primarily through an electrostatic interaction between an arginine residue located in the loop
 between the αP and the β5 sheet of the CNBD and the phosphate of cAMP and through hydrogen-bonding
 interactions with the oxygens on the phosphoribose (Figure* 9a). The importance of the arginine in cyclic
 nucleotide-binding has been demonstrated by site-directed mutagenesis. Mutation of this arginine to a
 lysine in PKA reduces the affinity for cAMP by nearly ten-fold [2]. In HCN channels, mutations of the
 arginine (R549 in HCN1) to glutamine or glutamate completely abolishes channel activation by cyclic
 nucleotides [3][4]. This conserved arginine residues are replaced by hydrophobic Y820 in the CNBLD of
 SOS1. As mentioned above, the cAMP is able to regulate spNHE10, which also contains this conserved
 arginine (Figure* 9b). Thus, based on these results, we speculate that the CNBD in SOS1 is unable to
 bind cAMP.

Figure* 9. Comparison of the cAMP-binding region in the known structures of human HCN1 with SOS1.
 **a.** The main residues interacting with cAMP in HCN1 (gray) and their correcting residues in the CNBLD
 of SOS1 are shown in stick. Hydrogen-bonding interactions is shown as dotted lines. **b.** Sequence
 alignment of the cAMP-binding region.

Reference:

[1] Windler, Florian, et al. "The solute carrier SLC9C1 is a Na⁺/H⁺-exchanger gated by an S4-type voltage-sensor and cyclic-
 nucleotide binding." *Nature communications* 9.1 (2018): 1-13.

[2] Bubis, J., et al. "A point mutation abolishes binding of cAMP to site A in the regulatory subunit of cAMP-dependent protein
 kinase." *Journal of Biological Chemistry* 263.20 (1988): 9668-9673.

[3] Liu, David T., et al. "Constraining ligand-binding site stoichiometry suggests that a cyclic nucleotide-gated channel is

composed of two functional dimers." *Neuron* 21.1 (1998): 235-248.
[4] Chen, Shan, Jing Wang, and Steven A. Siegelbaum. "Properties of hyperpolarization-activated pacemaker current
defined by coassembly of HCN1 and HCN2 subunits and basal modulation by cyclic nucleotide." *The Journal of general*
*physiology* 117.5 (2001): 491-504.

- Are the exact residues known which are phosphorylated in the autoinhibitory region? This becomes
largely important for the storytelling and conclusion drawn by the authors. If not known than the authors
could have screened the few potential positions through their complementation assay.

**Reply:** We thank reviewer for raising this point and agree with reviewer that it is very important to discuss
phosphorylation site. So far, S1138 has been proposed to be the phosphorylation site and S1136 cannot
be phosphorylated by constitutively active SOS2 mutant (SOS2^{T168D/Δ308}), but the S1136 is essential for
substrate recognition by SOS2^{T168D/Δ308}. Replacing either site with alanine (S1136A or S1138A) or removing
both sites but retaining C-loop and C-helix (SOS1^{Δ1047} or SOS1^{Δ1072}) did not activate SOS1, but disrupted
ability of SOS2^{T168D/Δ308} to activate the SOS1 [1].

As reviewer suggested, to search more phosphorylation sites, we designed a SOS1^{DAPA} double mutants
(S1136A and S1138A) and carried out yeast complementary assay. The SOS1^{DAPA} mutant cannot be
remarkably activated like the WT SOS1 by SOS2^{T168D/Δ308}. Strikingly, it is able to be largely activated by
the SOS2-SOS3 complex. These results suggest that, on the one hand, for some reason the constitutively
active SOS2^{T168D/Δ308} works differently to activate the SOS2 as the SOS2-SOS3 complex. On the other
hand, in addition to the S1138, the SOS2-SOS3 complex probably can recognizes and phosphorylates
other sites to activated the SOS1. Therefore, we mutate each individual Ser to Ala on the SOS1^{DAPA} mutant
to disrupt the potential phosphorylation site. The hypothesis is the mutant could not be fully activated once
the phosphorylation site is replaced by Ala. We next examine activity of these mutants in the presence and
absence of the SOS2-SOS3 complex. But unfortunately, we did not observe any mutant shows obvious
lower activity than the others. We speculate that there may be multiple sites that could be phosphorylated,
and simultaneous mutations at multiple sites are required to significantly disrupt the activation of SOS1.
This is likely essential to clearly distinguish differences in activity using current functional assays. However,
it is a time-consuming work and we believe our work is important enough to stand on its own.

Figure* 10. Phosphorylation Site Screening by yeast complementary assay.

Reference:

[1] Quintero, Francisco J., et al. "Activation of the plasma membrane Na/H antiporter Salt-Overly-Sensitive 1 (SOS1) by
phosphorylation of an auto-inhibitory C-terminal domain." *Proceedings of the National Academy of Sciences* 108.6 (2011):
2611-2616.

- If I understand the authors claims correctly, then the biggest determinant for the inhibition is the
stabilization of the dimeric cytosolic domain, which would sterically hinder the elevator mechanism of the
core domain. This in turn would make the interaction of the TM5 with HC8 less relevant, as also shown by
the functional complementation assays. While I find the description of TM5b as an intracellular lateral gate
interesting and important the current order of manuscript sections feels off. Better to finish with a short
proper conclusion section at the end including all relevant elements.

**Reply:** We appreciate and agree with reviewer’s comment. In the revised manuscript, we have included a
discussion section to summarize our findings. In the line 359-408, it reads “SOS1 is the only known plasma
membrane-localized Na⁺/H⁺ antiporter identified to date in higher plants. It is ubiquitously expressed in
epidermal cells at the root tip and in parenchyma at the xylem-symplast boundary of roots, stems and
leaves and plays pivotal roles in maintaining ion homeostasis and controlling long-distance Na⁺ transport

via the xylem. Soil salinity is becoming a severe environmental stress factor worldwide, and the major toxic
cation present in saline soils is Na^+ , which is seriously detrimental to plant growth and restricts agricultural
production. Therefore, SOS1 is emerging as a promising candidate to develop salt-tolerant transgenic
crops. Despite there have been recent advances in the understanding functional roles of cytosolic C-
terminal tail of eukaryotic Na^+/H^+ antiporters (i.e., NHE1 and NHE3)^{19,27}, the autoinhibitory mechanism of
SOS1 remains elusive.

Here, we determined the 2.8 Å-resolution cryo-EM structure of the full-length wild-type *Arabidopsis*
*thaliana* SOS1, revealing the architecture and subunit arrangement of this Na^+/H^+ antiporter. Unlike the
classic Na^+/H^+ exchanger, SOS1 harbors a large intracellular domain consisting of an “U”-shaped α -helical
disk just below the membrane, a cyclic nucleotide-binding homology domain, and a C-terminal antiparallel
β -sheet roll. We also determined SOS1 at two conformations in which lipid molecules differentially insert
into the extracellular cavity formed by two dimerization domains. In the SOS1^{expand} structure, two extra
lipids were determined to be buried in the dimerization interface, leading to extracellular expansion of the
TMD. However, the differences in bound lipids did not result in significant conformational changes in the
intracellular ends of TM helices and in the intracellular domains. According to the structural comparison,
we found that SOS1 is locked in an occluded state by shifting of TM5b toward the dimerization domain.
We proposed that TM5b may function as a lateral gate to control the accessibility of the ion binding pocket
from the intracellular side.

Two discrete fragments, C-loop (residues 1007-1017) and C-helix (residues 1030-1048), were
determined to interact with intracellular domains from different subunits. We speculate that interactions of
C-loop and C-loop with other domains contribute to stabilize dimerization of the intracellular domains.
Deletion of C-loop and/or C-helix (SOS1 ^{Δ C-helix}, SOS1 ^{Δ C-helix}) and mutation of W1013 or F1042 significantly
stimulate the transport activity of SOS1, demonstrating C-loop and C-helix serve as a crucial molecular
switch to regulate the activity of this antiporter. To gain insights into activation mechanism of SOS1, we
also made a constitutively active construct by truncating C-terminal autoinhibitory tail (SOS1 ^{Δ 998}) and
carried out cryo-EM study. It turns out that intracellular domains are completely blurred during 2D and 3D
classification, preventing us from obtaining a high-resolution map and suggesting that intracellular domains
may become highly mobile upon SOS1 activation. Interestingly, we found that the surface electrostatic
potential of the intracellular domains is negatively charged (Supplementary Fig. 12). Considering the
critical roles of C-loop/C-helix and phosphorylation of C terminal tail in activation of SOS1, we further
speculate that the phosphorylation of the C-terminal tail would introduce additional negative charge(s) that
would repel negatively charged intracellular domains, reducing binding affinity of the C-loop and C-helix to
the intracellular domain and destabilizing dimerization of the intracellular domain, thereby activating the
SOS1. However, further investigations are required to better understand how phosphorylation activates
the SOS1.”.

- Functional assays could be complemented by combination of 702EEK704 and 484DDEE487 alanine
mutations, as the authors claim that both the detachment of TM5b from HC8 and the interactions of HC1
are required to activate the protein, while as single mutations not enough.

**Reply:** We appreciate the reviewer’s comments. Structural analysis revealed two conserved interactions,
which we speculated might contribute to the functional regulation of SOS1. However, we found that
mutating ⁷⁰²EEK⁷⁰⁴ to three alanine residues (SOS1^{3A}) had little effect on SOS1 activity. In contrast, the
SOS1^{4A} (⁴⁸⁴DDEE⁴⁸⁷/AAAA) mutant showed higher activity than the wild-type SOS1. When we combined
the two mutations (SOS1^{3A4A}) as reviewer suggested, we observed a similar level of growth as the SOS1^{4A}

mutant, and the activity of SOS1^{3A4A} did not show any further enhancement. As suggested by the reviewer,
we have restructured the manuscript to avoid redundant descriptions, and we have discussed SOS1^{3A} and
SOS1^{4A} in different sections. Thus, we have not included the functional data of SOS1^{3A4A} in the revised
manuscript.

Figure* 11. Salt tolerance test of AXT3K cells expressing SOS1^{3A}, SOS1^{4A} and SOS1^{3A4A}, and compared
with wide-type SOS1 in AP medium with the indicated concentrations of NaCl.

- The description on the structural attempts for the truncated SOS1 construct is poor and not entirely
convincing. First the SEC profile, shows a broader peak, with a potential right shoulder, which could also
be indicative of presence of monomers (in particular considering it was done on a superose 6). More
importantly however, while I agree with the authors that the cytosolic domain is not visible, I also do not
see any evidence or clear density for the TMD in the 2D classes (there are also no top views shown). The
latter makes one wonder if the entire protein might have been unstable or the data set too small for proper
processing and thus relevant to draw any conclusions from it. Can the authors comment on how many
images were acquired and particles processed? This needs to be properly revisited, as it represents the
basis for an important statement: namely that the entire CTD is flexible, potentially even no longer
dimerized, without the autoinhibitory region. Otherwise, most of the claims throughout the text (e.g. 324-
326) also with regards to stabilization of the dimer interface need to be toned down.

**Reply:** We appreciate reviewer's comment. For the WT SOS1 sample, we collected ~1500 micrographs
and determined high resolution cryo-EM maps of SOS1 at occluded state. For the truncated SOS1^{Δ998}, we
totally collected 1312 micrographs and initially picked about 931,449 particles. The 2D classification
images all confirmed that the intracellular domains are flexible. Consequently, unlike in the WT SOS1
sample, the particles could not be properly aligned without aid of the intracellular domain. So, we could
not achieve a reasonable map for the SOS1^{Δ998} sample. In the revised supplementary figure 11, we provided
more 2D classification images, including tilted top views. We attached the revised supplementary figure
11d here for your convenience.

In the revised manuscript, we have included data processing details about SOS1^{Δ998} sample. In the
line 491-502, it reads "For the truncated SOS1^{Δ998} protein, a total of 931,449 particles were initially picked
from 1,312 micrographs using blob-picker and template-picker in CryoSPARC, followed by particle
cleaning using 2D classification. In contrast to the full-length SOS1, the intracellular part of SOS1^{Δ998} in
2D micrographs is completely blurred. The helical features of the transmembrane region within the micelle
are also ambiguous. Ab initio reconstruction was conducted to generate an initial map, in the presence or
absence of C2 symmetry. 3D classification was carried out against five references, including low-pass
filtered 2.8 Å map of SOS1 and four trash maps, with the application of C2 symmetry, yielding five classes.
However, all of these attempts are failed to generate a reasonable map with either transmembrane features

or intracellular domains features. All procedures of data processing were conducted in cryoSPARC.”.

- Another aspect why the structural attempt of the truncated SOS1 is unsatisfactory, is because even if
the CTD are flexible and does not visible in the cryo-EM data, it would be highly interesting to see which
state the transporter domain adopts under these conditions (and outward or inward facing instead of
occluded state, with an open TM5b gate as expected?, or does it even remain as a dimer?)

**Reply:** We agree with reviewer that it would be great to know conformational state of truncated SOS1.
However, it is tough to solve a membrane structure without a soluble domain to facilitate particle alignment.
For membrane protein with negligible soluble domain, researchers usually employ a specific antibody
recognizing conformational epitope to overcome this issue. Regarding to the oligomeric state truncated
SOS1, we believe a part of sample become monomer indicated by a right shoulder on SEC profile, as
reviewer also mentioned above. However, we did not pick these fractions during grids preparation. The
dimension of the micelle on the 2D classes is about 150 Å or 160 Å, similar to that of SOS1 WT sample,
representing a dimerized state.

Figure* 12. Dimension of the micelle on the 2D classes.

- Figure 4 should be replaced with a schematic overview of the proposed regulation mechanism in
SOS1. This is an example of how the reader is left with scattered detailed descriptions throughout the text,
which could be perfectly summarized in a schematic final figure to accompany a discussion that then could
include a broader perspective an comparison to regulation mechanism in other (Na⁺/H⁺) antiporters.

**Reply:** We appreciate reviewer's comment and have included a discussion section in the revised
manuscript. It reads now "SOS1 is the only known plasma membrane-localized Na⁺/H⁺ antiporter identified
to date in higher plants. It is ubiquitously expressed in epidermal cells at the root tip and in parenchyma at
the xylem-symplast boundary of roots, stems and leaves and plays pivotal roles in maintaining ion
homeostasis and controlling long-distance Na⁺ transport via the xylem. Soil salinity is becoming a severe
environmental stress factor worldwide, and the major toxic cation present in saline soils is Na⁺, which is

seriously detrimental to plant growth and restricts agricultural production. Therefore, SOS1 is emerging as
a promising candidate to develop salt-tolerant transgenic crops. Despite there have been recent advances
in the understanding functional roles of cytosolic C-terminal tail of eukaryotic Na⁺/H⁺ antiporters (i.e., NHE1
and NHE3)^{19,27}, the autoinhibitory mechanism of SOS1 remains elusive.

Here, we determined high resolution cryo-EM structures of the full-length wild-type *Arabidopsis*
*thaliana* SOS1, revealing the architecture and subunit arrangement of this Na⁺/H⁺ antiporter. Unlike the
classic Na⁺/H⁺ exchanger, SOS1 harbors a large intracellular domain consisting of an “U”-shaped α -helical
disk just below the membrane, a cyclic nucleotide-binding homology domain, and a C-terminal antiparallel
β -sheet roll. We also determined SOS1 at two conformations in which lipid molecules differentially insert
into the extracellular cavity formed by two dimerization domains. In the SOS1^{expand} structure, two extra
lipids were determined to be buried in the dimerization interface, leading to extracellular expansion of the
TMD. However, the differences in bound lipids did not result in significant conformational changes in the
intracellular ends of TM helices and in the intracellular domains. According to the structural comparison,
we found that SOS1 is locked in an occluded state by shifting of TM5b toward the dimerization domain.
We proposed that TM5b may function as a lateral gate to control the accessibility of the ion binding pocket
from the intracellular side.

Two discrete fragments, C-loop (residues 1007-1017) and C-helix (residues 1030-1048), were
determined to interact with intracellular domains from different subunits. We speculate that interactions of
C-loop and C-loop with other domains contribute to stabilize dimerization of the intracellular domains.
Deletion of C-loop and/or C-helix (SOS1 ^{Δ C-helix}, SOS1 ^{Δ C-helix}) and mutation of W1013 or F1042 significantly
stimulate the transport activity of SOS1, demonstrating C-loop and C-helix serve as a crucial molecular
switch to regulate the activity of this antiporter. To gain insights into activation mechanism of SOS1, we
also made a constitutively active construct by truncating C-terminal autoinhibitory tail (SOS1 ^{Δ 998}) and
carried out cryo-EM study. It turns out that intracellular domains are completely blurred during 2D and 3D
classification, preventing us from obtaining a high-resolution map and suggesting that intracellular domains
may become highly mobile upon SOS1 activation. Interestingly, we found that the surface electrostatic
potential of the intracellular domains is negatively charged (Supplementary Fig. 12). Considering the
critical roles of C-loop/C-helix and phosphorylation of C terminal tail in activation of SOS1, we further
hypothesis that the phosphorylation of the C-terminal tail would introduce additional negative charge(s)
that would repel negatively charged intracellular domains, reducing binding affinity of the C-loop and C-
helix to the intracellular domain and destabilizing dimerization of the intracellular domain, thereby activating
the SOS1. However, considering the phosphorylation sites have not been clearly identified, further
investigations are required to better understand how phosphorylation activates the SOS1.”.

Moreover, we have also made a schematic figure to display our findings. We have attached here for your
convenience.

**Figure 6. Proposed model for SOS1 activation.** TM5 and TM6 helices from the core domains are
 illustrated and labeled. Side chains of the D201^{TM6} are shown as sticks. Intracellular domains including the
 HCs, CNBLD, and β-CTD are represented according to their structure and labeled. The C-loops and C-
 helices are shown as tubes. **a**, The inhibited state is shown where the intracellular domains are tightly
 packed together. The TM5b occludes D201^{TM6} and makes it inaccessible from the intracellular side. The
 conformational change between the core and dimer domain is hindered by the tightly-packed intracellular
 domains, which are stabilized by the C-loops and C-helices. **b**, The active state is demonstrated, where
 the intracellular domains are unpacked due to the release of the C-loop and C-helices. D201^{TM6} is exposed
 on the intracellular side. The original position of TM5b in the inhibited state is overlaid using dashed
 rectangles.

I would also recommend the authors to consider restructuring, shortening (some things can go into material
 and methods) and subdividing (rather long subsection on autoinhibition) some sections to better guide the
 reader and sell a more concise story. For example, are many structural elements described and their
 function of importance anticipated, while the required functional data is only presented later. Consequently,
 some sections are introduced without prove and later become repetitive. Example: Tm5b is mentioned and
 discussed at three different positions (i.e. around line 224, line 295 and line 327).

**Reply:** We appreciate and agree with reviewer's comment. We combined TM5b related structural and
 functional analysis to avoid redundant description.

Lastly, I would recommend toning down in general some of the strong statements done on stabilization of
 the dimerization, while in many cases the interactions clearly contribute to the dimerization, there is no
 mutagenesis data included to report the impact/contribution on dimerization for single positions.

Yet, it is clear that the CTD likely plays a major role on the stabilization of the dimer, which is different to
 other CTD in other Na⁺/H⁺ antiporters. Considering the increasing knowledge on the importance of the
 dimerization from other Na⁺/H⁺ antiporters, the authors could elaborate more on a boarder comparison to
 other known structures.

**Reply:** We appreciate this comment and agree with reviewer that we should tune down statements on
 stabilization of the dimerization, which are listed as following.

In the line 181-183, it now reads "Considering no additional lipids were supplied during sample preparation,

we speculate that these lipid molecules observed within the cavity may be derived from HEK293 host and
are probably involved in stabilizing the dimeric SOS1.”.

In the line 173-175, it now reads “There is also a polar interaction between the N-terminal K85 of the
extracellularly amphipathic helix AH²⁻³ and E278 of the C-terminal of TM8’, which probably contributes to
the stabilization of the dimerization structure (Supplementary Fig. 7c).”.

In the line 200-202, it now reads “They are located proximal to the cell membrane and are involved in
electrostatic interactions with surrounding positively charged residues, such as K181^{TM5b}, K187^{TM6} and
K460^{HC1}, which may contribute to mediating the dimerization of SOS1.”.

The structures of other NHEs transporters are majorly composed of the transmembrane domain and we
have compared TMD of current SOS1 structure with them. Please also see reply below.

**Other mostly minor:**

- Intro and discussion lacks also a comparison to the other eukaryotic Na⁺/H⁺ antiporters like NHE9
and NHA2 instead of just NHE1

**Reply:** We thank the reviewer for this comment. The reason we extensively discussed NHE1 is due to the
structures of NHE1 are determined in both inward- and outward-facing conformation. Structural
comparison with NHE1 provides important insights about conformational transition of NHEs. However, we
entirely agree with reviewer that we should discuss other NHE antiporters. In the revised manuscript, we
compared SOS1 with NHE9, NHA2 and our recently published NHE3, and included a brief discussion in
the line 100-103, it reads “Similar to other antiporters in the CPA family^{17-19,26}, the arrangement of TM
helices is conserved in SOS1, that is, TMs 4–6 and TMs 11–13 form the core domain, and TMs 1–3 and
TMs 7–10 form the dimerization domain (Fig. 1d and Supplementary Figure 4a-e).”.

Structural comparisons have been shown in supplementary figure 4. We attached here for your
convenience.

**Supplementary Figure 4.** Structural comparison of the transmembrane regions of SOS1 (residue 32-450)
 with inward-facing human NHE1-CHP1^{IF} (PDB ID: 7DSV, residue 98-506) **(a)**, outward-facing human
 NHE1-CHP1^{OF} (PDB ID: 7DSX, residue 98-506) **(b)**, inward-facing human NHE3^{IF} (PDB ID: 7X2U, residue
 40-465) **(c)**, inward-facing horse NHE9 (PDB ID: 6Z3Y, residue 23-489) **(d)** and outward-facing bison
 NHA2 (PDB ID: 7P1I, residue 79-517) **(e)**, using entire TMD as the reference. The helices are displayed
 as cylinders. The RMSD values are 2.4 Å, 3.2 Å, 2.7 Å, 3.0 Å and 3.3 Å, respectively. The structural
 comparison with NHA2 (14TMs) is present in monomeric form for the existence of the extra N-terminal
 helix.

- Line 34: K=K+?

**Reply:** We appreciate your comments. We found “K deficiency symptoms”, “potassium deficiency
 symptoms” and “K⁺ deficiency symptoms” in previous literatures [1-4]. We changed the “K deficiency” to
 “potassium deficiency” to avoid the confusion.

Reference:

[1] Wang, Yi, and Wei-Hua Wu. "Plant sensing and signaling in response to K⁺-deficiency." *Molecular plant* 3.2 (2010): 280-
 287.

[2] Chen, Daoqian, et al. "Silicon moderated the K deficiency by improving the plant-water status in sorghum." *Scientific*
 *reports* 6.1 (2016): 1-14.

[3] Behera, Smrutisanjita, et al. "Two spatially and temporally distinct Ca²⁺ signals convey Arabidopsis thaliana responses
 to K⁺ deficiency." *New Phytologist* 213.2 (2017): 739-750.

[4] Rustioni, Laura, et al. "Iron, magnesium, nitrogen and potassium deficiency symptom discrimination by reflectance

spectroscopy in grapevine leaves." Scientia Horticulturae 241 (2018): 152-159.

**6** Line 37: sodium = sodium ions

**Reply:** We appreciate your comments. We have carefully checked this issue throughout the manuscript
and replaced sodium with Na⁺ in the revised manuscript.

- Lines 51-52: lack of space between the organism names: Pyrococcus abyssi and
Methanocaldococcus jannaschii

**Reply:** We thank the reviewer for pointing out this typo and we have corrected it in our revised manuscript.

- Lines 70-91: can be largely shifted to material and methods

**Reply:** Thanks for your comments and we have merged this part into the method section.

- Line 95: perhaps hint here that the vast majority of the autoinhibitory region (998-1146) is not resolved

**Reply:** Thanks for your comments. We have added this description and the revised sentence now writes:
"However, our SOS1 model is only composed of residues 32–972, 1007–1017 and 1030–1048 and the
vast majority of the autoinhibitory region (998-1146) is not resolved due to high flexibility."

- Line 101: amphipathic helix AH (3-4) is not shown/highlighted in any of the figures. On a similar note.
Helix 3 is not labeled in Figure 1d

**Reply:** Thank you for pointing this out. The AH³⁻⁴ has been highlighted in the Supplementary fig. 5a in the
revision. And we have added the label of TM3 in the Figure 1d.

- e.g Line 119-121: Wouldn't it be easier for the descriptions of the loops and their interactions to use a
similar nomenclature as for the amphipathic helix, so instead of IL4 use IL(TM7-8)

**Reply:** Thank you for pointing out this point and loops have been renamed in the revised manuscript.

- Line 145-149: Hard to validate without map and model but how "bad" is the beta-CTD resolved? Is it
appropriate to use AlphaFold at this point or better to exclude this region from the final model?

**Reply:** The local resolution for β-CTD (888-972) underneath the CNBLD was between 3.0 and 3.7 Å, as
a result, fewer side chains were observed in this region, but strong main-chain density allowed for accurate
model building and placement of AlphaFold structures. The model-map correlation coefficient between β-
CTD and map is 0.6, suggesting that a good agreement between the model and the map. we have also
included model-map fitting figure in the revised supplementary figure 3. We have attached here for your
convenience.

Figure* 13. Electron density of β -CTD. **a**, Local resolution of the cryo-EM density map of SOS1. The density of β -CTD is highlighted in a black box. **b**, Electron density map after density modification is shown in transparent surface, and the refined model is shown in cartoon with some large-sidechain residues shown in sticks and labeled.

- line 172-173: considering that no lipids were added during purification (correct?), it might be fair enough to mention it here and claim that the potential lipid densities observed here likely represent endogenously co-purified lipids from HEK host. A caveat here is that, no functional assays with the from HEK purified protein were performed, which would confirm that this state with the given lipids can be functional.

**Reply:** We appreciate the reviewer's comments. We did not supply extra lipid during purification and agree with reviewer that these lipids may derived from HEK293 host. We have included this statement in the revised manuscript. In the line 181-183, it now reads "Considering no additional lipids were supplied during sample preparation, we speculate that these lipid molecules observed within the cavity may be derived from HEK293 host and are probably involved in stabilizing the dimeric SOS1."

We totally agree with reviewer that it would be nice to study function of the purified protein sample by in vitro transport assays. However, it proved to be highly challenging due to the low yield of the recombinant SOS1, whereas liposome reconstitution would require large amounts of the protein sample. Moreover, lipid molecules have also been visualized in many other NHEs structures (NHE1-CHP1, NHE3-CHP1, NHE9 and NHA2 et al.). Therefore, we believe that the sample copurified with lipids is functional.

- Lines 186-188: based on what is the claim made that K181 and K187 are critical for dimerization of SOS1? Is it based on the inter-protomer interactions established? Still without actual mutagenesis studies you don't know if they are critical or might temporarily simply contribute to dimerization. Besides residues 181/187 need to move during transport cycle and can thus not be determinant for dimerization. Please rephrase.

**Reply:** We appreciate the reviewer's comments and agree with reviewer's statement. We have tune down statements. In the line 200-202, it now reads "They are located proximal to the cell membrane and are involved in electrostatic interactions with surrounding positively charged residues, such as K181^{TM5b}, K187^{TM6} and K460^{HC1}, which may contribute to mediating the dimerization of SOS1."

- I agree with the authors that D201 most likely represent the determinant cation binding site. However, considering that the accessibility of this residue and its importance for the transport is a central point of the

manuscript, the authors make a poor job in conveying this to the reader (in particular for a non-expert).
Preferably, the authors could add a functional complementation assay (as this is not a very laborious
experiment), showing that a mutation of this site renders the transporter inactive. The least the author
should do, is to emphasize stronger (in introduction or in lines starting with 204) that this is a highly
conserved motif of the CPA family, and has been extensively characterized and identified in all Na⁺/H⁺
antiporter studied so far as the central cation binding site.

**Reply:** We appreciate the reviewer's comments. We have carried out experiments as reviewer suggested.
It turns out that the D201A mutation is able to completely abolish the transport activity of the SOS1, even
in the presence of the SOS2 and SOS3. We have added a brief discussion in the revised manuscript. In
the line 234, it now reads "The conserved aspartate residue in "ND" motif on TM6 has been extensively
characterized in CPA1 as the central ion binding site³⁷. In the inward-facing and outward-facing NHE1
structures, residue D267^{TM6} is exposed to the intracellular and extracellular sides for binding with and
exchanging cations. Based on the sequence alignment and structural comparison, we identified the
corresponding residues D201 that potentially responsible for Na⁺ binding in SOS1 (Supplementary Fig.
9a). The complementation assay using salt-sensitive yeast indicates that the growth of AXT3K was
marginally rescued upon SOS1 expression and notably augmented in the presence of SOS2-SOS3 protein
kinase complex³⁶. Nevertheless, the transport activity of SOS1 was eliminated by the D201A mutation,
even in the presence of the SOS3-SOS2 complex, substantiating the significance of the D201^{TM6} in cations
association in both inward-facing and outward-facing conformations (Supplementary Fig. 9b).".

Related experimental results are included in the revised Supplementary Figure 9. Please see below.

- Lines 231-234: while I believe it I think the reader would benefit here from a short statement on how
the authors can be sure to properly model (with correct register) a loose stretch of 10 and 18 aa within an
otherwise large unresolved region. Also how sure can you be with the assignment of which element
belongs to which polypeptide chain?

**Reply:** Thank you for raising these points. The high resolution cryo-EM map provided clear side chain
features for C-loop and C-helix regions, which allows us to unambiguously build the model. The further
functional assay also supports the model is reliable.

In the line 275-278, we added details about model building of these regions. It reads that "Residues on the
C-loop with large sidechain, such as M1011 and W1013 help to determine the peptide sequences and the
sequence of C-helix was confirmed by secondary structure prediction and the large side chain residue of
F1042.".

Basically, it is hard to determine which chain C-loop and C-helix belong to. However, we speculate that

one of them has swapped and interacts with a different subunit. There are 35 (AAs: 973-1006) and 12
(AAs: 1018-1029) missing residues between the β -CTD and C-loop, and between C-loop and C-helix,
respectively. The distance from I972^A (I972 of chain A) to H1007¹ (position 1) or H1007² is approximately
24 Å and 28 Å, respectively. Considering the 35 missing residues between these two positions, it is
possible that the C-loop from both positions belongs to chain A. The distances between I1017¹ and L1030³,
and between I1017¹ and L1030⁴ are 16 Å and 45 Å, respectively. However, only 12 residues are missing
in this region. Therefore, we speculate that I1017¹ should connect with L1030³, which is located on the
same side as the C-loop.

We have included related description in the model building section of revised manuscript. It now reads
“However, the chain ID of the C-loop and C-helix is not clearly determined based on the cryo-EM map, due
to linkers connecting SOS1, C-loop and C-helix are missing. The distances between I972 from A subunit
and H1007 from two C-loops are about 28 Å and 24 Å, respectively. Considering 35 amino acids missing
between these two residues, we speculate that both two C-loop segments possibly belong to chain A. The
C-loop interacting with HC1, HC3 and HC5 from A subunit was set as chain A. The distances between
I1017 on C-loop from chain A and L1030 from two C-helix are 16 Å and 45 Å, respectively. Considering
only 12 residues missing between C-loop and C-helix, we speculate that the C-helix interacting with
CNBLD from chain B belongs to chain A (Supplementary Fig. 13).”.

We also made a figure to display the spatial arrangement and distances. We attached supplementary
figure 13 here for your convenience.

Supplementary Figure 13. C-loop and C-helix are swapped between the two monomers of SOS1. **a**, The
distances between I972^A (the last residue of β -CTD) to H1007¹ (the first residue of C-loop) or H1007² are
about 24 Å and 28 Å, respectively. **b**, The distances between I1017¹ (the last residue of C-loop) and L1030³
(the first residue of C-helix) and between I1017¹ and L1030⁴ are 16 Å and 45 Å, respectively. **c**, The
swapped C-loop and C-helix that viewed from the extracellular side.

- Line 240: Fig 3f is cited before 3e

**Reply:** We thank the reviewer for pointing this out and the two figures have been exchanged in our revised
version.

- Lines 250-254: tone down statement, e.g. “...and likely/presumably stabilize the intracellular domain
into an inhibited state”. In particular because this statement comes before your functional complementation
assays

**Reply:** We appreciate the reviewer's comments and have followed your suggestions to tone down our
statement. It now reads "Taken together, the results show that C-loop and C-helix are involved in extensive
intramolecular or intermolecular interactions with a-CTD and CNBLD and presumably stabilize the
intercellular domain in a specific conformation...".

- Lines 312-315: tone down statement. The results indicate and hint but do not conclusive show your
claim.

**Reply:** We appreciate the reviewer's comments. We have removed this speculation.

- Methods (data processing) considering the processing and the images shown the authors must have
processed large amounts in Relion and not in CryoSPARC as stated.

**Reply:** Thanks for your comment. We have clarified usage of Relion and CryoSPARC in the revised
method section.

- Methods: missing any info/details on the attempt for the structural determination of the truncated
version

**Reply:** We thank the reviewer for pointing this out and we have clarified the details about the data
processing of the truncated SOS1^{Δ998} in the Methods section.

In the line 491-502, it reads "For the truncated SOS1^{Δ998} protein, a total of 931,449 particles were initially
picked from 1,312 micrographs, followed by particle cleaning using 2D classification. In contrast to the full-
length SOS1, the intracellular part of SOS1^{Δ998} in 2D micrographs is completely blurred. The helical
features of the transmembrane region are also ambiguous. Ab initio reconstruction was conducted to
generate an initial map. 3D classification against the low-resolution map and four trash maps was carried
out with the application of C2 symmetry, which yielded five classes. The most populated class was
composed of 26% of total particles but the TM helices cannot be clearly displayed. Particles from this class
were selected and submitted to further 3D classification focusing on TMD domain. However, none of the
six classes displaying resolved transmembrane helices and we did not obtain a map with clear TMD density
because the completely disordered cytoplasmic structure broken the alignment of the particles. All
procedures of data processing were conducted in cryoSPARC."

- Line 432: which lipids were added?

**Reply:** Thanks for your comments. We actually placed hydrophobic tail of lipids in the model, not a
complete lipid molecule, as we do not know the identity of the lipid. To avoid confusion, we revised this
sentence. It now reads "The hydrocarbon chain of lipid was added manually in Coot."

- Methods (complementation assay): might be good to refer here or even in the manuscript that the
expression level or (if expressed at all) of the given constructs were not analyzed and thus the results nor
normalized.

**Reply:** We are grateful for the reviewer's comments. We analyzed the expression levels of the SOS1 and
mutants in a yeast strain using western blotting against the C-terminal 6×His tag. The results show that
their expression levels are comparable. We have included the protein expression data in the revised
manuscript as a supplementary figure 10, and we have mentioned this in the revised main text. In line 302,
it now reads "The expression level of these mutants is comparable to that of the wild-type (WT) SOS1, as
evidenced by the Western blot analysis (Supplementary Fig. 10a, b)."

However, we want to clarify that our functional data has not been normalized by the expression level, which
 is a common practice in previous literature [1-3]. We have addressed this in the revised manuscript in line
 557: "The functional data is not normalized by the expression level."

Reference:

[1] Quintero, Francisco J., et al. "Reconstitution in yeast of the Arabidopsis SOS signaling pathway for Na⁺
 homeostasis." Proceedings of the National Academy of Sciences 99.13 (2002): 9061-9066.

[2] Zhou, Yang, et al. "SpAHA1 and SpSOS1 coordinate in transgenic yeast to improve salt tolerance." PLoS One 10.9
 (2015): e0137447.

[3] Zhou, Yang, et al. "The Sesuvium portulacastrum plasma membrane Na⁺/H⁺ antiporter SpSOS1 complemented the salt
 sensitivity of transgenic Arabidopsis sos1 mutant plants." Plant Molecular Biology Reporter 36 (2018): 553-563.

- Line 459: Would be nice to provide the original data of the atomic absorption spectrometer in the
 supplementary information, not only the final values in the main figure.

**Reply:** Thanks for your suggestion. The original data for determination the sodium content in yeast by the
 proposed atomic absorption spectrometry method has been shown in Table 2. We have also attached here
 for your convenience.

Supplementary Table 3. Determination of Na⁺ content by atomic absorption spectrometer

Na ⁺ content (mg/L)	DW	Control			30 mM		
		1	2	3	1	2	3
Blank	0	1.208	1.132	1.127	1.208	1.132	1.127
AXT3K	0.01g	2.045	1.975	2.075	17.778	17.818	17.909
SOS1-WT	0.01g	1.925	2.05	1.97	11.855	11.465	11.68
SOS1+SOS2+SOS3	0.01g	1.97	2.01	2.035	9.425	11.195	9.971
ΔC-loop	0.01g	2.235	1.925	2.115	5.91	7.07	6.623
W1013A	0.01g	2.035	1.97	1.93	6.8	5.83	6.815
ΔC-helix	0.01g	2.055	2.04	2.14	10.905	12.205	11.074
F1042A	0.01g	2.05	2.06	2.11	12.225	11.76	10.765
ΔC-loop/ΔC-helix	0.01g	1.935	1.95	1.925	5.215	4.44	4.84
Δ998	0.01g	1.89	1.94	1.755	8.435	8.31	8.401
⁷⁰² E ⁷⁰⁴ /AAA	0.01g	2.915	2.697	2.689	15.859	15.03	15.577
⁴⁸⁴ D ⁴⁸⁷ /AAAA	0.01g	2.173	2.76	2.54	15.721	14.371	14.277

Samples volume was 10 ml.
 DW: dry weight.

- Fig 1: Fig1a: I would call it "domain structure" instead of "linear scheme". Also display here the not
 fully resolved autoinhibitory tail". TM 3 and 4 are missing in description. If the authors want to state out the
 difference between dimerization and core domain, coloring might be more helpful, than the use of cylinders
 and helices.

**Reply:** Thanks for your suggestion. We have updated the figure legend and reordered the panels to follow
 the description sequence in the manuscript. Specifically, we have moved panel a to panel c. It now reads:
 "Domain structure and organization of SOS1, numbers of amino acid defining the domains are indicated.

Dashed line indicated the unresolved disorder regions in the structure.” Also, TM3, TM4 and AH³⁻⁴ are
 now labeled in the revised Figure 1a. We attached the old and revised figure here for your convenience.
 Moreover, we would like to maintain the current color scheme to differentiate the dimerization and core
 domains, as it is consistent with panel a and e.

- Figure 3, the labelled boxed in panel a are written with capital, while panels are in small caps. Not all
 boxes and panels are labelled. (similar bug found in other figures as well)

**Reply:** We thank the reviewer for pointing this out. We have carefully checked the all figures and made
 corrections.

- S4: labels or location of alpha-A,B,D partially missing in sequence alignment and structure

**Reply:** We appreciate the reviewer's comments. In the original manuscript, we omitted the alpha-A and
 alpha-D to clearly illustrate the structural discrepancy in the structural elements contributing to the ligand
 binding pocket. In the revised version, we have included the alpha-A,B,D and provided complete sequence
 alignment and structural comparison of the CNBLD in the revised Supplementary Figure 6. We have
 attached the revised Supplementary Figure 6 for your convenience.

- S6: color code in legend wrong. Explain how structures were superimposed to better judge the RMSDs
mentioned in text

**Reply:** We appreciate the reviewer's kind reminding. We have corrected the color code in the figure legend
and added a description of how structures are superposed. It now read: "Structural comparison of the
transmembrane regions of SOS1 (residue 32-450) with inward-facing human NHE1-CHP1^{IF} (PDB ID:
7DSV, residue 98-506) (a), ...using entire TMD as the reference. The helices are displayed as cylinders.
The RMSD values are 2.4 Å, 3.2 Å, 2.7 Å, 3.0 Å and 3.3 Å, respectively. The structural comparison with
NHA2 (14TMs) is present in monomeric form for the existence of the extra N-terminal helix."

REVIEWER COMMENTS

Reviewer #1 (Remarks to the Author):

The authors provided solid arguments and experimental data to respond to my concerns, and the manuscript has been much improved.

Line 81 SOS-Class 1 should be SOS1-Class I

Line 295 intercellular should be Intracellular also in legend if the figure 4i

Reviewer #2 (Remarks to the Author):

Authors took the effort to address most of the requests from reviewers, and the formal presentation of results has been improved. Some of the additional data in the rebuttal letter is only for reviewers and has not been incorporated to the revised Ms.

To sum up the key data in this report: (1) the cryo-EM structure of the pore domain of SOS1 largely matches the arrangement of other cation antiporters of the CPA1 family. (2) The structure of the cytoplasmic region (CTD) between residues 458 and 972 was also determined with confidence; this protein stretch contains a non-canonical CNBLD that is known to be essential for protein activity, cannot bind cyclic nucleotides owing to steric hindrances, and the agonist, if any, remains unknown; these concepts are already in the literature and no new evidence is offered here for why the CNBLD is necessary for SOS1 activity or how it could affect protein regulation. (3) Two additional discrete domains (C-loop and C-helix) corresponding to a previously defined bi-partite auto-inhibitory domain (Quintero et al., 2011) were resolved by cryo-EM as discrete structures detached from the pore and cytoplasmic domains (linker sequences missing), and fitted into the larger protein body; the combined structural and mutagenesis data indicate that the physical interaction C-loop and C-helix with the CTD locks SOS1 in the inhibited state.

My perspective still is that the information gained from this study is meager relative to previous knowledge. Overall this work is not up to the standard expected from Nature Comms. Compared to other reports on the structural biology of ion transporters published by this journal, the novelty and impact of this report is below average. Papers that can be directly compared to this submission have raised the ante. Examples are Tascon et al. (2020) "Structural basis of proton-coupled potassium transport in the KUP family" Nat Commun 11: 626, Lu et al. (2022) "Structural basis for the activity regulation of a potassium channel AKT1 from Arabidopsis" Nature Commun 13: 5682, and even Dong et al. (2021) "Structure and mechanism of the human NHE1-CHP1 complex." Nat Commun 12: 3474 by the Yan Zhao's group.

Moreover, I've got the feeling that some of the data included in the rebuttal with the intent of addressing the reviewers' comments have blurred even more key concepts regarding the mechanism of SOS1 autoinhibition/activation (see below). The numbering of paragraph relates to my original report.

1 & 5. I pointed out the overt absence of biochemical assays supporting the functionality of the structural features deemed essential or critical for SOS1 activity (CNBD, lipid binding, C-loop and C-helix). Authors confirmed that mutant proteins are expressed at similar levels. However, the argument that biochemical analyses could not be done because of the low yield of the proteins is untenable. Ion exchange experiments could have been performed with plasma membrane vesicles prepared from yeast, which incidentally would rule out missorted mutant proteins behaving as inactive in the yeast cells (i.e., no restitution of cellular sodium resistance). According to the publication record, the group of Dr. Jiang has the expertise needed for these experiments. This weakness is further aggravated by the lack of proof that SOS1 was properly folded and active when expressed and purified from

HEK cells. The strong experimental linkage between structure and protein activity/regulation is inexcusable for a paper of this caliber, which should render sound (and novel) conclusions.

2 & 4. I understand that flexible protein domains are difficult to resolve by cryo-EM. However, the structure and mechanism of action of the SOS1 autoinhibitory domain is at the heart of this study and consequently it should go far beyond the fitting of discrete structures resolved in a patchy manner. This resorting to in silico aids increases the level of experimental proof that is needed to validate the conclusions. Besides mutating W1013 in the C-loop and F1042 in the C-helix, other residues in the complementary surfaces should have been mutated to test the correct fitting of W1013 and F1042 residues in the whole structure.

Sorry, I am missing the point of why in their rebuttal to my comment #4 regarding the unlinked structures of C-loop and C-helix authors respond with the structure of the beta-CTD and Fig *6.

I do not share the authors' argument that 'the phosphorylation site(s) have not been clearly identified to date'. The site of SOS2 phosphorylation has been mapped to Ser1138 (Quintero et al., PNAS, 2011). One shortcoming here is that the structure of the phosphorylation domain could not be resolved. The explanation that 'the phosphorylation of the C-terminal tail would introduce additional negative charge(s) that would repel negatively charged intracellular domains, reducing binding affinity of the C-loop and C-helix to the intracellular domain ... thereby activating the SOS1' is not supported by the result that phosphomimicking mutations S1136D and S1138D failed to activate SOS1 (Fig 4 in the rebuttal and Quintero et al., 2011).

In their response to Reviewer #3 authors state that, in line with previous reports, the SOS1-DAPA mutant cannot be activated by SOS2T168D/ Δ 308. However, Fig 10 in the rebuttal shows that SOS2T168D/ Δ 308 partly activated the mutant protein SOS1-DAPA. To explain this contradictory result, authors speculated that multiple sites could be phosphorylated besides Ser1138. To probe the point they systematically mutated 12 serine residues in the C-terminus of SOS1, alone or combined with the DAPA mutation in the SOS2 target site. However, none of the tested mutations abrogated activation by the SOS2-SOS3 complex. Together, these inconsistent results evidence that the presumed interplay between the autoinhibitory domain and activation by phosphorylation is far from clear and that the structural basis for protein activation has not been resolved. They also strengthen the need for conducting biochemical assays instead of relying solely on mutant complementation of yeast cells.

Last, it is worth noting that Figure 10A included in the rebuttal contains at least one panel (400 mM) that shows signs of image cropping and pasting.

Reviewer #3 (Remarks to the Author):

Dear author,

I think the author have done a good job in revising the manuscript, and have sufficiently addressed my comments. In my opinion the revised version is suitable for publication.

Minor points:

1. To fit the citation in the main text, Supplementary Figure 9 and 10 need to be swapped.
2. I would consider adjusting the right panel of the new Figure 6, to indicate that
 - a) the active state is reached when the cytosolic domain is phosphorylated (region of phosphorylation could be hinted at with a dotted circle around the inhibitory tail)
 - b) an active state will mean that the core domain will be able to undergo its elevator movement and alternatingly expose the ion binding site (how exactly TM5c will move is less

accurate to predict). These things could be indicated by arrows or even two states of the protein (outward facing and inward facing being in equilibrium)

Point-by-point response for
Architecture and autoinhibitory mechanism of the plasma membrane Na⁺/H⁺ antiporter SOS1 in
*Arabidopsis*

**Reviewer #1 (Remarks to the Author):**

The authors provided solid arguments and experimental data to respond to my concerns, and the
manuscript has been much improved.

**Reply:** Thank you very much for your positive feedback on our revised manuscript. We greatly appreciate
your efforts in reviewing our work and providing us with constructive criticism to strengthen it. We are glad
that our revisions and experimental data have addressed your concerns.

Line 81 SOS-Class 1 should be SOS1-Class I

**Reply:** We thank the reviewer for pointing out this and have corrected this in the revised manuscript.

Line 295 intercellular should be Intracellular also in legend if the figure 4i

**Reply:** We thank the reviewer for pointing out this and have corrected this in the revised manuscript and
figure legend.

**Reviewer #2 (Remarks to the Author):**

Authors took the effort to address most of the requests from reviewers, and the formal presentation of
results has been improved. Some of the additional data in the rebuttal letter is only for reviewers and has
not been incorporated to the revised Ms.

**Reply:** We appreciate the reviewer's comments. Due to space limitations, we were unable to incorporate
all of the additional data in the manuscript. However, both valuable comments of reviewers and our
responses will be available as a supplementary "peer review file" on the Nature Communication website,
which is accessible for all readers.

To sum up the key data in this report: (1) the cryo-EM structure of the pore domain of SOS1 largely
matches the arrangement of other cation antiporters of the CPA1 family. (2) The structure of the
cytoplasmic region (CTD) between residues 458 and 972 was also determined with confidence; this protein
stretch contains a non-canonical CNBLD that is known to be essential for protein activity, cannot bind
cyclic nucleotides owing to steric hindrances, and the agonist, if any, remains unknown; these concepts
are already in the literature and no new evidence is offered here for why the CNBLD is necessary for SOS1
activity or how it could affect protein regulation. (3) Two additional discrete domains (C-loop and C-helix)
corresponding to a previously defined bi-partite auto-inhibitory domain (Quintero et al., 2011) were
resolved by cryo-EM as discrete structures detached from the pore and cytoplasmic domains (linker
sequences missing), and fitted into the larger protein body; the combined structural and mutagenesis data
indicate that the physical interaction C-loop and C-helix with the CTD locks SOS1 in the inhibited state.

My perspective still is that the information gained from this study is meager relative to previous knowledge.
Overall this work is not up to the standard expected from Nature Comms. Compared to other reports on
the structural biology of ion transporters published by this journal, the novelty and impact of this report is
below average. Papers that can be directly compared to this submission have raised the ante. Examples
are Tascon et al. (2020) "Structural basis of proton-coupled potassium transport in the KUP family" Nat
Commun 11: 626, Lu et al. (2022) "Structural basis for the activity regulation of a potassium channel AKT1
from Arabidopsis" Nature Commun 13: 5682, and even Dong et al. (2021) "Structure and mechanism of
the human NHE1-CHP1 complex." Nat Commun 12: 3474 by the Yan Zhao's group.

Moreover, I've got the feeling that some of the data included in the rebuttal with the intent of addressing
the reviewers' comments have blurred even more key concepts regarding the mechanism of SOS1
autoinhibition/activation (see below). The numbering of paragraph relates to my original report.

1 & 5. I pointed out the overt absence of biochemical assays supporting the functionality of the structural
features deemed essential or critical for SOS1 activity (CNBD, lipid binding, C-loop and C-helix). Authors
confirmed that mutant proteins are expressed at similar levels. However, the argument that biochemical
analyses could not be done because of the low yield of the proteins is untenable. Ion exchange
experiments could have been performed with plasma membrane vesicles prepared from yeast, which
incidentally would rule out missorted mutant proteins behaving as inactive in the yeast cells (i.e., no
restitution of cellular sodium resistance). According to the publication record, the group of Dr. Jiang has
the expertise needed for these experiments. This weakness is further aggravated by the lack of proof that
SOS1 was properly folded and active when expressed and purified from HEK cells. The strong
experimental linkage between structure and protein activity/regulation is inexcusable for a paper of this
caliber, which should render sound (and novel) conclusions.

**Reply:** We appreciate the reviewer's comments and suggestions. The yeast complementation assay and
Na⁺ content determination assay used in our study are both the classic and widely used methods to
measure the transport activity of plant sodium-proton antiporters [1-6]. We employed these two different
approaches to assess the transport activity differences between WT SOS1 and its mutants, which we
believed that is sufficient to support our speculations.

The reviewer suggested that using the plasma membrane vesicles transport assay could rule out the
missorted mutant proteins. However, in our experiments, most of the tested mutations in SOS1 actually
enhanced the transport activity by disrupting the auto-inhibitory interaction, indicating that misfolding may
not be a concern. The only exception is the D201A mutant, which resulted in a complete loss of transport
activity of SOS1, even in the presence of the SOS2-SOS3 complex. Notably, the residue D201 is highly
conserved among the Na⁺/H⁺ antiporters. Previous immunofluorescence experiment conducted on the
homologous protein NHE1 demonstrated that substituting the aspartate with glutamate or asparagine at
the corresponding position did not prevent its targeting to the plasma membrane [7]. Moreover, based on
the sequence alignment and structural comparison, the aspartate would be alternately exposed to the
intracellular and extracellular sides of the membrane during the transport cycle, indicating its crucial role
in ion binding [8].

Upon submitting our previous response, we carefully reconsidered the reviewer's suggestion in the
 initial version of comments regarding the incorporation of other biochemical experiments to strengthen our
 conclusion. Consequently, we conducted the non-invasive micro-test technology (NMT) assay to provide
 additional support for our findings. The NMT assay is a widely-used technique in vitro for detecting the flux
 rate of ions or molecules into or out of cells. It is extensively employed in the analysis of SOS1 function
 and it allows real-time monitoring of Na⁺ transport in the transgenic yeasts, similar to the vesicle transport
 assay suggested by the reviewer [9-15]. In our study, we directly monitor the Na⁺ efflux from the transgenic
 yeast cells treated with 200 mM NaCl. Notably, the transgenic yeasts expressing SOS1^{ΔC-loop} and SOS1^{ΔC-}
 83 ^{loop/C-helix} mutants had a faster net efflux than that expressing WT SOS1 under saline conditions. We have
 84 included the description in the revised manuscript. In line 315-319, it reads "Non-invasive micro-test
 technology (NMT) showed a significant enhancement of Na⁺ efflux in yeast expressing SOS1^{ΔC-loop} and
 SOS1^{ΔC-loop/C-helix} compared to WT SOS1 under salt conditions (Supplementary Fig. 11). Notably, the mean
 rate of Na⁺ efflux in the SOS1^{ΔC-loop/C-helix} transformants exhibited 2-fold faster than the SOS1^{ΔC-loop}
 transformants."

Related experimental results are included in the revised Supplementary Figure 11. Please see below.

 Figure** 1. Net Na⁺ effluxes in yeast strains measured by NMT assay.

Yeast strains expressing empty pYPGE15 vector, SOS1-WT, SOS1^{ΔC-loop}, SOS1^{ΔC-helix}, and SOS1^{ΔC-loop/C-}
 93 ^{helix} were subjected to ion fluxes measurements using NMT assay for 5 min. Each point represents the
 94 mean of three individual samples and the bars denote the standard deviation. The column chart shows the
 95 mean efflux rates of Na⁺ within the measuring period. Asterisks indicate significant differences compared
 with SOS1-WT (**** p < 0.0001).

The method has also been updated (line 568-580), "The real-time Na⁺ efflux from the yeast cells under
 salt stress was measured using NMT^{55,56} (NMT Physiolyzer, YoungerUSA, MA, USA). Yeast cells were
 first cultivated in 5 mL AP medium (8 mM phosphoric acid, 2% glucose, 2 mM MgSO₄, 1 mM KCl, 0.2 mM
 CaCl₂, plus trace elements and vitamins, adjusted to pH 6.5 with arginine) at 28 °C for 24 h with shaking

(200 rpm), and then transferred into 10 mL fresh medium with a final concentration of 100 mM NaCl and
allowed to grow at 28 °C for 2 h with shaking (200 rpm) , after which 0.5 mL nutrient solution with yeast
cells were harvested by centrifuge (3000 g, 5 min), the pellet was resuspended in 0.2 mL measuring buffer
(100 mM mannitol, 1.0 mM NaCl, 0.2 mM MES, pH 6.3). Na⁺-flux microsensor were positioned 5 μm away
from the cells enriched by the conical filter membrane, and a continuous flux recording was taken for 5
106 min for each sample using imFluxes V3.0 software (Xuyue Company, Beijing, China). The experiment was
107 repeated three times for each category of transformed cell.”

The reviewer also doubts that the protein is probably poorly folded in HEK293 cell. But we believe that
it is correctly folded based on two aspects. Firstly, HEK293F expression system possesses significant
advantages for the heterologous expression of eukaryotic proteins for biochemical analysis and protein-
protein interaction studies. Especially, it also was widely used for the structural and electrophysiological
analysis of plant membrane proteins, such as ion channels, transporters and photoactive plant proteins
[16-21]. Secondly, SOS1 belongs to the monovalent cation proton antiporter-1 (CPA1) family. The
structure of the transmembrane domain of SOS1 is similar to other CPA1 homologues. The overall folding
of each subunit shows a pseudo 2-fold symmetry, with the symmetry axis parallel to the membrane plane,
which relates TMs 1–6 to TMs 8–13. In particular, both TM5 and TM12 are characteristically unwound in
the middle and the two unwound peptides cross each other in the vicinity of the pseudo 2-fold axis.
Therefore, we believe that the purified sample from HEK293F cell used for the cryo-EM study was properly
folded and functional.

Reference:

- [1] Quintero, Francisco J., Michael R. Blatt, and José M. Pardo. “Functional conservation between yeast and plant
endosomal Na⁺/H⁺ antiporters.” *FEBS letters* 471.2-3 (2000): 224-228.
- [2] Quintero, Francisco J., et al. “Reconstitution in yeast of the Arabidopsis SOS signaling pathway for Na⁺
homeostasis.” *Proceedings of the National Academy of Sciences* 99.13 (2002): 9061-9066.
- [3] Shi, Huazhong, et al. “The putative plasma membrane Na⁺/H⁺ antiporter SOS1 controls long-distance Na⁺ transport in
plants.” *The plant cell* 14.2 (2002): 465-477.
- [4] Zhao, Fengyun, et al. “Expression of yeast SOD2 in transgenic rice results in increased salt tolerance.” *Plant
Science* 170.2 (2006): 216-224.
- [5] Zhao, Xiufang, et al. “Soybean Na⁺/H⁺ antiporter GmsSOS1 enhances antioxidant enzyme activity and reduces Na⁺
accumulation in Arabidopsis and yeast cells under salt stress.” *Acta Physiologiae Plantarum* 39 (2017): 1-11.
- [6] Xu, Haixia, et al. “Functional characterization of a wheat plasma membrane Na⁺/H⁺ antiporter in yeast.” *Archives of
biochemistry and biophysics* 473.1 (2008): 8-15.
- [7] Murtazina, Rakhilya, et al. “Functional analysis of polar amino - acid residues in membrane associated regions of the
NHE1 isoform of the mammalian Na⁺/H⁺ exchanger.” *European Journal of Biochemistry* 268.17 (2001): 4674-4685.
- [8] Dong, Yanli, et al. "Structure and mechanism of the human NHE1-CHP1 complex." *Nature Communications* 12.1 (2021):
3474.
- [9] Fan, Yafei, et al. "Co-expression of SpSOS1 and SpAHA1 in transgenic Arabidopsis plants improves salinity
tolerance." *BMC plant biology* 19.1 (2019): 1-13.
- [10] Yang, Yang, et al. "Overexpression of the P t SOS 2 gene improves tolerance to salt stress in transgenic poplar
plants." *Plant Biotechnology Journal* 13.7 (2015): 962-973.

- [11] Ma, Dong-Mei, et al. "Co-expression of the Arabidopsis SOS genes enhances salt tolerance in transgenic tall fescue
(*Festuca arundinacea* Schreb.)." *Protoplasma* 251 (2014): 219-231.
- [12] Yue, Yuesen, et al. "SOS1 gene overexpression increased salt tolerance in transgenic tobacco by maintaining a higher
K⁺/Na⁺ ratio." *Journal of Plant Physiology* 169.3 (2012): 255-261.
- [13] Zhu, Min, et al. "Nax loci affect SOS1-like Na⁺/H⁺ exchanger expression and activity in wheat." *Journal of experimental
botany* 67.3 (2016): 835-844.
- [14] Bose, Jayakumar, et al. "Haem oxygenase modifies salinity tolerance in Arabidopsis by controlling K⁺ retention via
regulation of the plasma membrane H⁺-ATPase and by altering SOS1 transcript levels in roots." *Journal of experimental
botany* 64.2 (2013): 471-481.
- [15] Guo, Kun - Mei, Olga Babourina, and Zed Rengel. "Na⁺/H⁺ antiporter activity of the SOS1 gene: lifetime imaging
analysis and electrophysiological studies on Arabidopsis seedlings." *Physiologia Plantarum* 137.2 (2009): 155-165.
- [16] Wang, Jiangqin, et al. "Structural basis of ALMT1-mediated aluminum resistance in Arabidopsis." *Cell Research* 32.1
(2022): 89-98.
- [17] Li, Yawen, et al. "Structural insights into a plant mechanosensitive ion channel MSL1." *Cell Reports* 30.13 (2020): 4518-
4527.
- [18] Liu, Peng, et al. "Mechanism of sphingolipid homeostasis revealed by structural analysis of Arabidopsis SPT-ORM1
complex." *Science Advances* 9.13 (2023): eadg0728.
- [19] Su, Nannan, et al. "Structures and mechanisms of the Arabidopsis auxin transporter PIN3." *Nature* 609.7927 (2022):
616-621.
- [20] Green, Mariah N., et al. "Structure of the Arabidopsis thaliana glutamate receptor-like channel GLR3. 4." *Molecular
cell* 81.15 (2021): 3216-3226.
- [21] Yang, Liang, et al. "Using HEK293T expression system to study photoactive plant cryptochromes." *Frontiers in plant
science* 7 (2016): 940.

2 & 4. I understand that flexible protein domains are difficult to resolve by cryo-EM. However, the structure
and mechanism of action of the SOS1 autoinhibitory domain is at the heart of this study and consequently
it should go far beyond the fitting of discrete structures resolved in a patchy manner. This resorting to in
silico aids increases the level of experimental proof that is needed to validate the conclusions. Besides
mutating W1013 in the C-loop and F1042 in the C-helix, other residues in the complementary surfaces
should have been mutated to test the correct fitting of W1013 and F1042 residues in the whole structure.

Sorry, I am missing the point of why in their rebuttal to my comment #4 regarding the unlinked structures
of C-loop and C-helix authors respond with the structure of the beta-CTD and Fig *6.

**Reply:** We appreciate the reviewer's comments. We should clarify that the structures of C-loop and C-
helix were built based on the actual cryo-electron microscopy electron density, rather than predicted by
AlphaFold2. According to the local resolution map shown in Supplementary Figure 3, the resolution for the
C-loop and C-helix regions ranges from approximately 2.8 to 3.5 Å. The high-resolution features enable
176 us to build de novo and assign the side chain of these residues on the C-loop and C-helix unambiguously.
To support the accurate fitting of their positions, we have provided the additional views of the density map
for the C-loop and C-helix here. Based on the high-quality density map and the precise fitting of the C-loop
and C-helix structures, we believe that conducting further functional validation to test the correct fitting of

W1013 and F1042 residues is unnecessary.

C-loop: 1007-HRGLMSWPENI-1017 C-helix: 1030- LSLSERAMQLSIFGSMVNV-1048

Figure** 2. Density maps of the C-loop and C-helix.

I do not share the authors' argument that 'the phosphorylation site(s) have not been clearly identified to
date'. The site of SOS2 phosphorylation has been mapped to Ser1138 (Quintero et al., PNAS, 2011). One
shortcoming here is that the structure of the phosphorylation domain could not be resolved. The
explanation that 'the phosphorylation of the C-terminal tail would introduce additional negative charge(s)
that would repel negatively charged intracellular domains, reducing binding affinity of the C-loop and C-
helix to the intracellular domain ... thereby activating the SOS1' is not supported by the result that
phosphomimicking mutations S1136D and S1138D failed to activate SOS1 (Fig 4 in the rebuttal and
Quintero et al., 2011).

In their response to Reviewer #3 authors state that, in line with previous reports, the SOS1-DAPA mutant
cannot be activated by SOS2T168D/Δ308. However, Fig 10 in the rebuttal shows that SOS2T168D/Δ308
partly activated the mutant protein SOS1-DAPA. To explain this contradictory result, authors speculated
that multiple sites could be phosphorylated besides Ser1138. To probe the point, they systematically
mutated 12 serine residues in the C-terminus of SOS1, alone or combined with the DAPA mutation in the
SOS2 target site. However, none of the tested mutations abrogated activation by the SOS2-SOS3
complex. Together, these inconsistent results evidence that the presumed interplay between the
autoinhibitory domain and activation by phosphorylation is far from clear and that the structural basis for
protein activation has not been resolved. They also strengthen the need for conducting biochemical assays
instead of relying solely on mutant complementation of yeast cells.

**Reply:** Thanks for your comments. We acknowledged that S1138 has previously been identified as the
SOS2 phosphorylation site. However, in our experiments, we observed that the SOS2-SOS3 protein
kinase complex was still able to activate the S1136A/S1138A mutant of SOS1 (SOS1^{S1136A/S1138A}) (Figure**
3). This finding suggested that, on the one hand, the phosphorylation at S1138 is not necessary for

activation of SOS1 by the SOS2-SOS3 complex. On the other hand, there may be other site(s) could be
recognized and phosphorylated by SOS2-SOS3 that are involved in the transporter activation. Therefore,
we have clarified the statement about the phosphorylation sites have not been clearly identified in the
revised manuscript. Therefore, S1136D/S1138D mutant of SOS1 could not be activated through the
introduction of negative charges, but this still could not rule out the possibility that other phosphorylation
site(s) induce the conformational change through electrostatic repulsion.

We have added the description in the Discussion section of the manuscript, line 410-418 reads,"
Previous studies have identified S1138 as the phosphorylation site and highlighted the role of S1136 in
localizing SOS2 to the phosphorylation site⁷. However, we found that the serine-to-alanine mutant
SOS1^{S1136A/S1138A} conferred the same degree of salt tolerance in yeast as the wild-type SOS1 in the
presence of the SOS2-SOS3 complex (Supplementary Fig. 14). This finding strongly suggests the
existence of additional potential phosphorylation site(s) involved in SOS1 activation by the SOS2-SOS3
complex that have not yet been clearly identified. Further investigations are required to map the specific
locations of these additional phosphorylation sites and gain valuable insights into the activation
mechanism.", and the corresponding figure is attached here for your convenience.

Figure** 3. Activation of the serine-to-alanine mutant SOS1^{S1136A/S1138A} by SOS2-SOS3 complex.
Transgenic yeasts expressing SOS1 and its mutant SOS1^{S1136A/S1138A} were grown in AP medium
supplemented with different concentrations of NaCl as indicated, with and without the presence of the
SOS2-SOS3 complex.

The reviewer pointed out that the SOS1^{DAPA} mutant was unable to be activated by SOS2^{T168D/Δ308} in
the previous reports, but could partially activated in our study. We found that the previous report only tested
the growth of the yeast that co-expression with SOS1^{DAPA} and SOS2^{T168D/Δ308} in the presence of 150 mM
NaCl, and then concluded that SOS1^{DAPA} mutant cannot be activated by SOS2^{T168D/Δ308} [1]. In contrast,
our study employed a NaCl concentration gradient ranging from 0 mM to 400 mM, which enabled us to
assess the salt tolerance of the transgenic yeast in a more precise manner. As a result, we observed that
the yeast cells co-transformed with SOS1^{DAPA} and SOS2^{T168D/Δ308} grew better than with SOS1^{DAPA} alone
in the presence of 50 mM NaCl (Figure** 4). However, these yeast cells were nearly unable to grow under
conditions exceeding 100 mM NaCl, which was basically consistent with the published results that the co-
transformed yeast with SOS1^{DAPA} and SOS2^{T168D/Δ308} cannot grow in the presence of 150 mM NaCl. For
easier comparison, we have attached the relevant figures here.

a

b

Figure** 4. Growth comparison of yeast cells expressing SOS1^{DAPA} and SOS2^{T168D/Δ308} under salt stress
 conditions between the previous study (a) and our experiments (b). Panel a is adapted from Figure 3B in
 the paper titled “Activation of the plasma membrane Na/H antiporter Salt-Overly-Sensitive 1 (SOS1) by
 phosphorylation of an auto-inhibitory C-terminal domain” published in 2011[1].

We would like to clarify that our hypothesis concerning multiple phosphorylation sites is primarily
 based on the observation of SOS1^{DAPA} can still activated by the SOS2-SOS3 complex comparable to WT
 SOS1, however, it is not based on the functional difference in the activation of SOS1^{DAPA} with
 SOS2^{T168D/Δ308} between previous reports and our study. To search other phosphorylation sites, we mutated
 several individual Ser to Ala on the SOS1^{DAPA} mutant, but did not observe any mutant shows obvious lower
 activity than the others in the presence of the SOS2-SOS3 complex. These results suggested that the
 phosphorylation and activation of SOS1 is a complicated mechanism, and the simultaneous disruption of
 multiple sites may be required to completely abolish the activation of SOS1. However, it is a time-
 consuming task and we believe our work is important enough to stand on its own.

Regulation by phosphorylation is a widely acknowledged mechanism for controlling biological activity
 of proteins. Many previous studies using different approaches have proposed a common model in which
 phosphorylation-introduced negative charge(s) cause electrostatic repulsion with adjacent negative
 charged regions, ultimately influencing the ligand binding, intramolecular interactions and protein-protein
 associations [2-10]. In our study, the C-loop and C-helix were identified as critical elements in inhibiting
 the transport activity of SOS1, supported by a range of biochemical assays, including the complementary
 assay, ion content measurement assay, as well as Na⁺ efflux measurement assay. Specially, these
 inhibitory elements contact the intracellular α-CTD and CNBLD regions, which are characterized by a
 negatively charged surface. Therefore, we believe it is reasonable to hypothesize that the negative
 charge(s) introduced by phosphorylation could induce electrostatic repulsion and contribute to the
 dissociation of the C-loop and C-helix from the intracellular domains, thereby activating SOS1.

Reference:

- [1] Quintero, Francisco J., et al. "Activation of the plasma membrane Na/H antiporter Salt-Overly-Sensitive 1 (SOS1) by
 phosphorylation of an auto-inhibitory C-terminal domain." *Proceedings of the National Academy of Sciences* 108.6 (2011):
 2611-2616.
- [2] Wang, Lingyun, and Ji - Bin Peng. "Phosphorylation of KLHL3 at serine 433 impairs its interaction with the acidic motif
 of WNK4: a molecular dynamics study." *Protein Science* 26.2 (2017): 163-173.
- [3] Cai, Na, et al. "Mass spectrometric analysis of TRPM6 and TRPM7 phosphorylation reveals regulatory mechanisms of
 the channel-kinases." *Scientific reports* 7.1 (2017): 42739.
- [4] Jarmuła, Adam, et al. "Mechanism of influence of phosphorylation on serine 124 on a decrease of catalytic activity of
 human thymidylate synthase." *Bioorganic & medicinal chemistry* 18.10 (2010): 3361-3370.
- [5] Schneider, Michael L., and Carol Beth Post. "Solution structure of a band 3 peptide inhibitor bound to aldolase: a
 proposed mechanism for regulating binding by tyrosine phosphorylation." *Biochemistry* 34.51 (1995): 16574-16584.
- [6] Piazza, Michael, et al. "Solution structure of calmodulin bound to the target peptide of endothelial nitric oxide synthase
 phosphorylated at Thr495." *Biochemistry* 53.8 (2014): 1241-1249.
- [7] Hurley, J. H., et al. "Regulation of isocitrate dehydrogenase by phosphorylation involves no long-range conformational
 change in the free enzyme." *Journal of Biological Chemistry* 265.7 (1990): 3599-3602.
- [8] Tang, Chenxiang, et al. "Impaired dNTPase Activity of SAMHD1 by Phosphomimetic Mutation of Thr-592* \diamond ." *Journal of*
 *Biological Chemistry* 290.44 (2015): 26352-26359.
- [9] Chen, Yuwen, Meng Gao, and Yongqi Huang. "Deciphering the Mechanism of 14 - 3 - 3 ζ - Mediated Inhibition of
 Phosducin Binding to the G Protein Transducin Complex." *The FASEB Journal* 33.S1 (2019): 631-10.
- [10] Kuang, Yao, et al. "Structural basis for the phosphorylation of FUNDC1 LIR as a molecular switch of
 mitophagy." *Autophagy* 12.12 (2016): 2363-2373.

Last, it is worth noting that Figure 10A included in the rebuttal contains at least one panel (400 mM) that
 shows signs of image cropping and pasting.

**Reply:** Thanks for your comments. The growth images of yeast cells co-expressing three SOS genes
 under various NaCl conditions were pasted to the last row. This is because these transformants were
 initially grown on separate plates and were not originally intended to be included together. To avoid
 confusion, we have spotted these transformants on a single plate in each panel and the results have been
 attached here for your reference (Figure** 5).

Figure** 5. The reproduced results of the yeast complementary assay.

**Reviewer #3 (Remarks to the Author):**

Dear author,

I think the author have done a good job in revising the manuscript, and have sufficiently addressed my
comments. In my opinion the revised version is suitable for publication.

**Minor points:**

1. To fit the citation in the main text, Supplementary Figure 9 and 10 need to be swapped.

**Reply:** Thank you very much for pointing out this issue. Supplementary Figure 9 and 10 have been
swapped in the revised version.

2. I would consider adjusting the right panel of the new Figure 6, to indicate that

a) the active state is reached when the cytosolic domain is phosphorylated (region of phosphorylation
could be hinted at with a dotted circle around the inhibitory tail)

b) an active state will mean that the core domain will be able to undergo its elevator movement and
alternately expose the ion binding site (how exactly TM5c will move is less accurate to predict). These
things could be indicated by arrows or even two states of the protein (outward facing and inward facing
being in equilibrium)

**Reply:** We appreciate the reviewer's comments. We agree that additional clarification can be provided in
the right panel of Figure 6 to better illustrate the active state and the transport mechanism of SOS1. We
have made the following adjustments:

a) The putative phosphorylated region was indicated with a red dotted circle on the unresolved C-terminal
tail of the active state.

b) In order to indicate the elevator-like alternating access mechanism during cation exchange, we overlaid
our inward-facing model (opaque) with a proposed outward-facing model (transparent). The movement
of the core domain from the inward-facing state to the outward-facing state is indicated with bent
arrows. The ion binding site is alternatively exposed to the intracellular side and extracellular side of
the membrane during the transport process.

Figure 6. Proposed model for SOS1 activation.

TM5 and TM6 helices from the core domains are illustrated and labeled. Side chains of the D201^{TM6} are
shown as sticks. Intracellular domains including the HCs, CNBLD, and β-CTD are represented according

to their structure and labeled. The C-loop and C-helix are shown as tubes. The unresolved structure of the
C-terminal tail is shown in dashed lines. **a**, The inhibited state is shown where the intracellular domains
are tightly packed together. The TM5b occludes D201^{TM6} and makes it inaccessible from the intracellular
side. **b**, The active state is demonstrated, where the intracellular domains are unpacked due to the release
of the C-loop and C-helix. The phosphorylated region is denoted by a red dotted circle. The core domain
undergoes elevator-like movement between the inward-facing conformation and outward-facing
conformation, depicted using opaque and transparent cartoons, respectively. Arrows indicate the
movement of the core domain from the inward-facing state to the outward-facing state.